# Efficient and Learnable Transformed Tensor Nuclear Norm with Exact Recoverable Theory

## Abstract

The tensor nuclear norm represents the low-rank property of tensor slices under a transformation. Finding a good transformation is crucial for the tensor nuclear norm. However, existing transformations are either fixed and not adaptable to the data, leading to ineffective results, or they are nonlinear and non-invertible, which prevents theoretical guarantees for the transformed tensor nuclear norm. Besides, some transformations are too complex and computationally expensive. To address these issues, this paper first proposes a fast data-adaptive and learnable column-orthogonal transformation learning framework with an exact recoverable theoretical guarantee. Extensive experiments have validated the effectiveness of the proposed models and theories.

## 1 Introduction

In real-life scenarios, many high-dimensional tensor data, such as hyperspectral images (HSIs), multispectral images (MSIs), and multi-frame videos, exhibit strong low-rank properties. Leveraging such low-rank structures of tensor data is crucial for solving tensor data restoration tasks, including but not limited to tensor completion (TC) [1, 2] and tensor robust principal component analysis (TRPCA) [3, 4]. Numerous methods have achieved outstanding results in practical applications by exploiting the low-rank property of tensors, such as video processing [5, 6], hyperspectral denoising [7, 8, 9], classification [10, 11].

There are various definitions of tensor rank, which differ from the rank used for matrices [12, 1]. Two well-known types of tensor decomposition are based on the CANDECOMP/PARAFAC (CP) and Tucker decompositions, which define the CP rank and Tucker rank, respectively [12]. These decompositions have been widely studied and have demonstrated competitive performance in low-rank tensor recovery. Computing the CP rank is known to be NP-hard, and a clear convex surrogate for this rank has not been established. On the other hand, computing the Tucker rank involves unfolding tensors along each mode into matrices, which may result in the loss of intrinsic high-order interactive information. In addition to these two ranks, the tensor tubal rank is also commonly used for tensor decomposition [13]. This rank is computed via tensor singular value decomposition (t-SVD), which was initially derived from a novel definition of the tensor-tensor (t-t) product [14]. Unlike other methods, t-SVD can operate on an integral third-order tensor without reshaping it into matrices, by using the discrete Fourier transform (DFT). For a third-order tensor $\mathcal{A} \in \mathbb{R}^{n_1 \times n_2 \times n_3}$, assuming that its third mode has a low-rank property, the transformed tensor $\overline{\mathcal{A}}$ can be obtained as follows:

$$\overline{\mathcal{A}} = \mathcal{A} \times_3 \mathbf{L}, \tag{1}$$

where $\times_3$ denotes mode-3 tensor product [12], and $\mathbf{L} \in \mathbb{R}^{n_3 \times n_3}$ is corresponding DFT matrix which satisfies $\mathbf{L}\mathbf{L}^T = \mathbf{L}^T\mathbf{L} = n_3\mathbf{I}$. Then the definition of the tensor tubal rank of $\mathcal{A}$ is $\mathrm{rank}_t(\mathcal{A}) =$

Table 1: The characteristics of different transformed TNN.

| Methods | TNN [2] | DCTNN [28] | UTNN [30] | WTNN [29] | CTNN [32] | FTNN [31] | S2NTNN [23] | Q-rank [24] | SALTS [25] | Ours |
|---|---|---|---|---|---|---|---|---|---|---|
| Transform | FFT | DCT | Unitary | Wavelet | Couple | Framelet | DNN | Unitary | Unitary | COM |
| Learnable? | ✗ | ✗ | ✗ | ✗ | ✗ | ✗ | ✓ | ✓ | ✓ | ✓ |
| Theory? | ✓ | ✓ | ✓ | ✓ | ✗ | ✗ | ✗ | ✓ | ✗ | ✓ |
| Speed | Moderate | Moderate | Moderate | Moderate | Slow | Slow | Fast | Very slow | Very slow | **Fast** |

$\sum_{i=1}^{n_3} \text{rank}(\overline{\boldsymbol{\mathcal{A}}}(:,:,i))$, where $\overline{\boldsymbol{\mathcal{A}}}(:,:,i)$ is the frontal slice of $\overline{\boldsymbol{\mathcal{A}}}$. Since the minimization of the tubal rank is an NP-hard problem. Zhang et al. [15] built a convex surrogate of the tensor tubal rank, named the tensor nuclear norm (**TNN**) by summing the matrix nuclear norm of each frontal slice under DFT. Thus the DFT-transformed TNN is defined as:

$$\|\boldsymbol{\mathcal{A}}\|_* = \sum_{i=1}^{n_3} \|\overline{\boldsymbol{\mathcal{A}}}(:,:,i)\|_* = \sum_{i=1}^{n_3} \|\overline{\boldsymbol{A}}^{(k)}\|_*. \tag{2}$$

Based on the DFT transformed TNN, Zhang and Aeron [2] and Lu et al. [3] give the exact recovery theorem for TC and TRPCA task by minimizing the TNN norm, respectively. Since then, many variants of DFT transformed TNN are proposed, such as weight TNN [16], partial sum of TNN (PSTNN) [17], Schatten-p norm TNN [18], p-shrinkage TNN [19], and many others [20, 21, 22].

Referring to Eq. (1), if we substitute the DFT matrix with another transform matrix/operator $\mathbf{L}$, we can obtain a transformed tensor and corresponding induced TNN norms that differ from those obtained using DFT. Hence, a crucial question arises: what type of transform matrix/operator is appropriate? Intuitively, a suitable transform operator should satisfy the following three criteria:

1) **Data adaptation.** The design of transform operators must depend on the data to better utilize its characteristics, which is a recent viewpoint. Works such as S2NTNN [23], Q-rank [24], and SALTS [25] have employed various methods to learn transform matrices from data. S2NTNN uses deep neural networks, Q-rank introduces a new algebraic definition, and SALTS uses SVD decomposition. Although only Q-rank has theoretical guarantees, updating the transform matrix and tensor recovery are independent processes that take a long time, making it impractical for real-world tasks.

2) **Theoretical guarantee** Theoretical guarantees are crucial for both models and algorithms. Currently, the exact recoverable guarantees are based on fixed linear invertible transforms, such as DFT, discrete cosine transform (DCT) [26, 27, 28], wavelet transformation [29], and unitary transformation [30], but they lack adaptability to data. In addition, there are fixed complex transforms that do not have recoverable theoretical guarantees, such as framelet transform [31], and coupe transform [32].

3) **Good Performance** Good transforms should improve restoration performance.

To achieve these objectives, this paper leverages the tensor structure and exploits the low-rank property of the third mode of the tensor to learn an adaptive column-orthogonal matrix (COM) transform for each data instance. Specifically, we model the low-rank tensor to be restored as the product of a smaller-sized factor tensor and a COM. This modeling approach effectively captures the low-rank structure of the tensor and facilitates the learning of the COM transform. Moreover, due to the reduced size of the factor tensor compared to the original tensor, our proposed model achieves accelerated computation. Additionally, we provide theoretical guarantees for the recoverability of our proposed model. To facilitate comparison, we present some classical transform-based tensor nuclear norm (TNN) approaches in Table 1. It can be observed from the table that only our modeling approach can stand out by simultaneously considering data adaptability, theoretical guarantees, and computational efficiency. In summary, this article first presents an efficient learnable transformed tensor nuclear norm (TNN) model with recoverable theoretical guarantees.

## 2 Notations and Preliminaries

### 2.1 Notations

In this paper, we denote tensors by boldface Euler script letters, e.g., $\boldsymbol{\mathcal{A}}$. Matrices are denoted by boldface capital letters, e.g., $\boldsymbol{A}$; vectors are denoted by boldface lowercase letters, e.g., $\boldsymbol{a}$, and scalars are denoted by lowercase letters, e.g., a. We denote $\boldsymbol{I}_n$ as the $n \times n$ identity matrix. For a 3-order tensor $\boldsymbol{\mathcal{A}} \in \mathbb{R}^{n_1 \times n_2 \times n_3}$, the frontal slice $\boldsymbol{\mathcal{A}}(:,:,i)$ is denoted compactly as $\boldsymbol{A}^{(i)}$. The tube

is denoted as $\mathcal{A}(i, j, :)$. The mode-n unfolding matrix of $\mathcal{A}$ is denoted as $\mathbf{A}_{(n)} = \text{unfold}_n(\mathcal{A})$, and $\text{fold}_n(\mathbf{A}_{(n)}) = \mathcal{A}$, where $\text{fold}_n$ is the inverse of unfolding operator. The mode-$n$ product of a tensor $\mathcal{X} \in \mathbb{R}^{I_1 \times I_2 \times I_3}$ and a matrix $\mathbf{A} \in \mathbb{R}^{J_n \times I_n}$ is denoted as $\mathcal{Y} := \mathcal{X} \times_n \mathbf{A}$ (see definition in [12]). Some norms of vector, matrix and tensor are used. We denote the $\|\mathcal{A}\|_1 = \sum_{ijk} |a_{ijk}|$, the infinity norm as $\|\mathcal{A}\|_\infty = \max_{ijk} |a_{ijk}|$ and the Frobenius norm as $\|\mathcal{A}\|_F = \sqrt{\sum_{ijk} |a_{ijk}|^2}$, respectively.

## 2.2 Adaptive Transformation

For a third-order tensor $\mathcal{A} \in \mathbb{R}^{n_1 \times n_2 \times n_3}$, assuming that its third mode has low-rank property, it can be factorized as

$$\mathcal{A} = \mathcal{U} \times_3 \mathbf{V}, \tag{3}$$

where $\times_3$ denotes mode-3 tensor product, $\mathcal{U} \in \mathbb{R}^{n_1 \times n_2 \times r_3}$, $\mathbf{V} \in \mathbb{R}^{n_3 \times r_3}(r_3 \leq n_3)$ satisfying $\mathbf{V}^T \mathbf{V} = \mathbf{I}$ and $r_3 = \text{Rank}(\mathbf{A}_{(3)})$. According to low-rank tensor decomposition (3), we have.

$$\mathcal{U} = \mathcal{A} \times_3 \mathbf{V}^T \iff \mathbf{U}_{(3)} = \mathbf{U}_{(3)} \mathbf{V}^T \mathbf{V} = \mathbf{A}_{(3)} \mathbf{V}. \tag{4}$$

Therefore, if we regard $\mathcal{U}$ as a transformed tensor $\overline{\mathcal{A}}$, then $\mathbf{V}^T$ can be regarded as the transform matrix $\mathbf{L}$, and $\mathbf{V}$ is the inverse transform of $\mathbf{V}^T$. Then we denote the TNN under the COM learned from the data as the Adaptive TNN (**ATNN**), which can be reformulated as:

$$\|\overline{\mathcal{A}}\|_* = \sum_{k=1}^{r_3} \|\overline{\mathbf{A}}^{(k)}\|_* = \sum_{k=1}^{R} \|(\mathcal{A} \times_3 \mathbf{L}^T)^{(k)}\|_*, \text{ s.t. } \mathcal{A} = \mathcal{A} \times_3 \mathbf{L}^T \times_3 \mathbf{L}. \tag{5}$$

**Remark 1** *It should be noted that comparing Eq. (5) and Eq. (2), it can be seen that ATNN has faster solution efficiency than DFT-transformed TNN since the transformed tensor under COM transform has fewer slices. The stronger the low rank of the tensor, that is, the lower the $r_3/n_3$ value, the higher the solution efficiency of ATNN can be obtained. However, since we want to ensure that the information of $\mathcal{A}$ with a rank of $Rank(\mathbf{A}_{(3)})$ before and after the transform will not be lost, i.e., $\mathcal{A} = \mathcal{A} \times_3 \mathbf{L}^T \times_3 \mathbf{L}$ is established, the condition $r_3 \geq Rank(\mathbf{A}_{(3)})$ must hold.*

## 2.3 T-product and T-SVD

Here, we give the definitions of t-product and t-SVD based on COM transform.

For $\mathcal{A} \in \mathbb{R}^{n_1 \times n_2 \times n_3}, \mathcal{B} \in \mathbb{R}^{n_2 \times n_4 \times n_3}$, the COM $\mathbf{L}^T$ transformed tensor of $\mathcal{A}, \mathcal{B}$ are $\overline{\mathcal{A}} = \mathcal{A} \times \mathbf{L}^T \in \mathbb{R}^{n_1 \times n_2 \times R}, \overline{\mathcal{B}} = \mathcal{B} \times \mathbf{L}^T \in \mathbb{R}^{n_2 \times n_4 \times R}$, respectively, via Eq. (1), then we define

$$\overline{\mathbf{A}} = \text{bdiag}(\overline{\mathcal{A}}) = \begin{bmatrix} \overline{\mathbf{A}}^{(1)} & & & \\ & \overline{\mathbf{A}}^{(2)} & & \\ & & \ddots & \\ & & & \overline{\mathbf{A}}^{(R)} \end{bmatrix}, \overline{\mathcal{A}} = \text{bfold}\left(\overline{\mathbf{A}}\right). \tag{6}$$

**Definition 1 (T-product)** *Let $\mathcal{A} \in \mathbb{R}^{n_1 \times n_2 \times n_3}, \mathcal{B} \in \mathbb{R}^{n_2 \times n_4 \times n_3}$ and COM $\mathbf{L}^T \in \mathbb{R}^{r_3 \times n_3}, (r_3 \leq n_3)$ satisfying $\mathbf{L}^T \mathbf{L} = \mathbf{I}_R$, then the t-product under transform $\mathbf{L}^T$ is defined as*

$$\mathcal{C} = \mathcal{A} *_L \mathcal{B} = bfold(bdiag(\overline{\mathcal{A}})bdiag(\overline{\mathcal{B}})) \times_3 \mathbf{L} = bfold(\overline{\mathbf{A}}\,\overline{\mathbf{B}}) \times_3 \mathbf{L} \in \mathbb{R}^{n1 \times n_4 \times n_3}, \tag{7}$$

*where $\overline{\mathcal{A}} = \mathcal{A} \times_3 \mathbf{L}^T \in \mathbb{R}^{n_1 \times n_2 \times r_3}$ and $\overline{\mathcal{B}} = \mathcal{B} \times_3 \mathbf{L}^T \in \mathbb{R}^{n_2 \times n_4 \times r_3}$.*

According to the Definition 1, we have $\mathcal{C} = \mathcal{A} *_L \mathcal{B} \iff \overline{\mathbf{C}} = \overline{\mathbf{A}}\,\overline{\mathbf{B}}$ since $\text{bfold}(\overline{\mathbf{C}}) = \overline{\mathcal{C}} = \mathcal{C} \times_3 \mathbf{L}^T = \text{bfold}(\overline{\mathbf{A}}\,\overline{\mathbf{B}}) \times_3 \mathbf{L} \times_3 \mathbf{L}^T = \text{bfold}(\overline{\mathbf{A}}\,\overline{\mathbf{B}}) \times_3 (\mathbf{L}^T\mathbf{L}) = \text{bfold}(\overline{\mathbf{A}}\,\overline{\mathbf{B}})$.

The t-product enjoys many similar properties to the matrix-matrix product. For example, the t-product is associate, i.e., $\mathcal{A} * (\mathcal{B} * \mathcal{C}) = (\mathcal{A} * \mathcal{B}) * \mathcal{C}$. We also need some other concepts on tensors.

**Definition 2 (Transpose)** *The transpose of a tensor $\mathcal{A} \in \mathbb{R}^{n_1 \times n_2 \times n_3}$ is the tensor $\mathcal{A}^T \in \mathbb{R}^{n_2 \times n_1 \times n_3}$ obtained by transposing each of the frontal slices.*

112 **Definition 3 (Identity tensor)** *A third-order tensor $\mathcal{A} \in \mathbb{R}^{n \times n \times n_3}$ is called identity tensor if it*
113 *satisfies that each frontal slice is identity matrix, i.e., $\mathbf{A}^{(i)} = \mathbf{I}$ for all $i = 1, \cdots, n_3$.*

114 **Definition 4 (Orthogonal tensor)** *A third-order tensor $\mathcal{Q} \in \mathbb{R}^{n \times n \times n_3}$ is called orthogonal tensor*
115 *if it satisfies that $\mathcal{Q}^T *_L \mathcal{Q} = \mathcal{Q} *_L \mathcal{Q}^T = \mathcal{I}$.*

116 **Definition 5 (F-diagonal tensor)** *A tensor is called f-diagonal if each of its frontal slices is a diago-*
117 *nal matrix.*

118 **Theorem 1 (T-SVD)** *Let $\mathcal{A} \in \mathbb{R}^{n_1 \times n_2 \times n_3}$. Then it can be factorized as*

$$\mathcal{A} = \mathcal{U} *_L \mathcal{S} *_L \mathcal{V}^T, \tag{8}$$

119 *where $\mathcal{U} \in \mathcal{R}^{n_1 \times n_1 \times n_3}$, $\mathcal{V} \in \mathcal{R}^{n_2 \times n_2 \times n_3}$ are orthogonal, and $\mathcal{S} \in \mathcal{R}^{n_1 \times n_2 \times n_3}$ is f-diagonal.*

120 By replacing DFT transform with COM transform $\mathbf{L}^T$, we can prove the above Theorem [3].

121 **Definition 6 (Tensor tubal rank [14] & TNN [3])** *For $\mathcal{A} \in \mathbb{R}^{n_1 \times n_2 \times n_3}$, the tensor tubal rank,*
122 *denoted as $rank_t(\mathcal{A})$, is defined as the number of nonzero singular tubes of $\mathcal{S}$, where $\mathcal{S}$ is from the*
123 *t-SVD of $\mathcal{A} = \mathcal{U} *_L \mathcal{S} *_L \mathcal{V}^T$. We can write*

$$rank_t(\mathcal{A}) = \#\{i, \mathcal{S}(i, i, :) \neq \mathbf{0}\}. \tag{9}$$

124 *And its tensor nuclear norm (TNN) is defined as*

$$\|\mathcal{A}\|_* = \sum_i \|\mathcal{S}(i, i, :)\|_1 = \|\mathcal{S}\|_1. \tag{10}$$

125 Using the t-product definition, we can get $\mathcal{A} = \mathcal{U} *_L \mathcal{S} *_L \mathcal{V}^T \iff \overline{A} = \overline{\mathbf{U}}\,\overline{\mathbf{S}}\,\overline{\mathbf{V}^T}$, thus we have

$$\|\mathcal{A}\|_* = \|\mathcal{S}\|_1 = \|\overline{\mathcal{S}}\|_* = \|\overline{\mathcal{A}}\|_* = \|\overline{\mathcal{A}}\|_* \tag{11}$$

126 by combing Eq. (5), Eq. (6) and Eq. (10).

## 3 Tensor Recovery via ATNN Minimization

### 3.1 Models

129 The observed tensor and the tensor that needs to be recovered are denoted as $\mathcal{Y}$ and $\mathcal{X}_0$, respectively.
130 For the tensor completion (TC), the observation $\mathcal{Y}$ has the support set $\Omega \sim \text{Ber}(\rho)$, i.e., $\mathcal{P}_\Omega(\mathcal{Y}) = $
131 $\mathcal{P}_\Omega(\mathcal{X}_0)$. For the tensor robust principal component analysis (TRPCA), the observation $\mathcal{Y}$ is
132 corrupted with a sparse component $\mathcal{E}_0$ (which may represent foreground and sparse noise), denoted
133 as $\mathcal{Y} = \mathcal{X}_0 + \mathcal{E}_0$.

134 If the COM $\mathbf{L}^T$ satisfying Eq. (5) is known, we can obtain the following two models:

$$(\text{TRPCA}) : \max_{\mathcal{X}, \mathcal{S}} \ \|\mathcal{X} \times_3 \mathbf{L}^T\|_* + \lambda\|\mathcal{S}\|_1, \ s.t. \ \mathcal{Y} = \mathcal{X} + \mathcal{E},$$

$$(\text{TC}) : \max_{\mathcal{X}} \ \|\mathcal{X} \times_3 \mathbf{L}^T\|_*, \ s.t. \ \mathcal{P}_\Omega(\mathcal{Y}) = \mathcal{P}_\Omega(\mathcal{X}). \tag{12}$$

135 Actually, it is often not possible to obtain $\mathbf{L}^T$ that satisfies Eq. (5) in advance. Recall Eq. (5), where
136 the constraint $\mathcal{A} = \mathcal{A} \times_3 \mathbf{L}^T \times_3 \mathbf{L}$ shows that the information of $\mathcal{A}$ after the change and inverse
137 change will not be lost, as long as $\mathbf{L}$ is obtained from the SVD decomposition of $\mathcal{X}$, Eq. (5) can be
138 satisfied. Hence, we can learn a suitable COM $\mathbf{L}$ from the data. By decomposing $\mathcal{X}$ as $\mathcal{X} = \overline{\mathcal{M}} \times_3 \mathbf{L}$
139 and setting $\mathcal{M} = \mathcal{X} \times_3 \mathbf{L}^T$, we can obtain the following alternative model to Eq. (12):

$$(\text{TRPCA}) : \max_{\overline{\mathcal{M}}, \mathcal{S}, \mathbf{L}} \ \|\overline{\mathcal{M}}\|_* + \lambda\|\mathcal{E}\|_1, \ s.t. \ \mathcal{Y} = \overline{\mathcal{M}} \times_3 \mathbf{L} + \mathcal{E}, \mathbf{L}^T\mathbf{L} = \mathbf{I},$$

$$(\text{TC}) : \max_{\overline{\mathcal{M}}, \mathbf{L}} \ \|\overline{\mathcal{M}}\|_*, \ s.t. \ \mathcal{P}_\Omega(\mathcal{Y}) = \mathcal{P}_\Omega(\overline{\mathcal{M}} \times_3 \mathbf{L}), \mathbf{L}^T\mathbf{L} = \mathbf{I}. \tag{13}$$

### 3.2 Incoherence Conditions

141 The incoherence condition is one of the most vital theoretical tools in low-rank recovery [33, 3, 4].
142 Below, we define $\mathring{\mathfrak{e}}_i$ as the tensor column basis and the tensor incoherence conditions similar to [3].

143 **Definition 7 (Tensor Incoherence Conditions)** *For $\mathcal{X}_0 \in \mathbb{R}^{n_1 \times n_2 \times n_3}$ with t-SVD rank R, it has*
144 *the skinny t-SVD $\mathcal{X}_0 = \mathcal{U} *_L \mathcal{S} *_L \mathcal{V}^T$. Then $\mathcal{X}_0$ is said to satisfy the tensor incoherence conditions*
145 *with parameter $\mu$ if*

$$\max_{i \in [1, n_1]} \|\mathcal{U}^T *_L \mathring{\mathfrak{e}}_i\|_F \leq \sqrt{\frac{\mu R}{n_1}}, \ \max_{j \in [1, n_2]} \|\mathcal{V}^T *_L \mathring{\mathfrak{e}}_j\|_F \leq \sqrt{\frac{\mu R}{n_2}}, \|\mathcal{U} *_L \mathcal{V}^T\|_F \leq \sqrt{\frac{\mu R}{n_1 n_2}}. \tag{14}$$

**Algorithm 1** ADMM for solving ATNN-RPCA model (13)

---

**Input**: Observation $\mathcal{Y} \in \mathbb{R}^{n_1 \times n_2 \times n_3}$, $\lambda = 1/\sqrt{\max(n_1, n_2)}$, $\mu = 1/\|\mathcal{Y}\|_*$, $\rho = 1.25$, $\mu_m = 1e^7\mu$, and the column number of learnable COM matrix $r_3$.

1: Initialize $\boldsymbol{\Lambda} = \mathcal{E} = \mathcal{O}$, $\overline{\mathcal{M}} = \text{bdiag}(\mathcal{U})$ and $\mathbf{L} = \mathbf{V}$, where $\mathcal{U}, \mathbf{V}$ is the low-rank tensor decomposition of among mode-3, i.e., $\text{unfold}_3(\mathcal{Y}) = (\mathcal{U}) \times_3 \mathbf{L}$

2: **while** not convergence **do**

3: Update $\overline{\mathcal{M}} := \text{SVD}_{1/\mu}((\mathcal{Y} - \mathcal{E} + \boldsymbol{\Lambda}/\mu) \times_3 \mathbf{L}^T)$.

4: Update $\mathbf{L} := \mathbf{B}\mathbf{D}^T$, where $[\mathbf{B}, \mathbf{C}, \mathbf{D}] = \text{svd}(\text{unfold}_3(\mathcal{Y} - \mathcal{E} + \boldsymbol{\Lambda}/\mu)^T \text{unfold}_3(\text{bfold}(\overline{\mathcal{U}})))$.

5: Update $\mathcal{X} := \overline{\mathcal{M}} \times_3 \mathbf{L}$

6: Update $\mathcal{E} := \mathcal{S}_{\lambda/\mu}(\mathcal{Y} - \mathcal{X} + \boldsymbol{\Lambda}/\mu)$.

7: Update multipliers $\boldsymbol{\Lambda} := \boldsymbol{\Lambda} + \mu(\mathcal{Y} - \mathcal{X} - \mathcal{E})$ ;

8: Let $\mu = \min\{\rho\mu, \mu_m\}$.

9: **end while**

**Output**: recovered tensors $\mathcal{X} = \overline{\mathcal{M}} \times_3 \mathbf{L}$ and $\mathcal{E}$.

---

## 3.3 Main results

We now demonstrate that both the model (12) and (13) possess exact recovery capability.

**Theorem 2 (TRPCA Theorem)** *Consider ATNN-based TRPCA model (12) and (13). Suppose that $\mathcal{X}_0 \in \mathbb{R}^{n \times n \times n_3}$ obeys the tensor incoherence conditions (14) and $\mathcal{E}_0$'s support set, denoted as $\Omega_0$, is uniformly distributed among all sets of cardinality $m$. Then, there exist universal constants $c_1, c_2 > 0$ such that $(\mathcal{X}_0, \mathcal{E}_0)$ is the unique solution to model (12) and (13) when $\lambda = 1/\sqrt{n}$ with probability at least $1 - c_1(nn_3)^{-c_2}$, provided that*

$$rank_t(\mathcal{X}_0) \leq \rho_r \mu^{-1} n \log^{-2}(n) \ \ and \ \ m \leq \rho_s n^2 n_3, \tag{15}$$

*where $\rho_r, \rho_s > 0$ are some numerical constants.*

**Theorem 3 (TC Theorem)** *Consider ATNN-based TC model (12) and (13). Suppose that $\mathcal{X}_0 \in \mathbb{R}^{n \times n \times n_3}$ obeys the tensor incoherence conditions (14) and $\Omega \sim Ber(p)$. Then, there exist universal constants $c_0, c_1, c_2 > 0$ such that $\mathcal{X}_0$ is the unique solution to model model (12) and (13) with probability at least $1 - c_1(nn_3)^{-c_2}$, provided that*

$$p \geq c_0 \mu R n^{-1} \log^2(n). \tag{16}$$

**Remark 2** *It should be noted that although the model (12) and (13) are slightly different, they are the same in the proof of the exact recoverable theory. Assume that the optimal values of models (12) and (13) are $(\hat{\mathcal{X}}, \hat{\mathcal{E}})$ and $(\hat{\overline{\mathcal{M}}}, \hat{\mathbf{L}}, \hat{\mathcal{E}})$, respectively. A recoverable theory of model (12) requires proving $(\hat{\mathcal{X}}, \hat{\mathcal{E}}) = (\mathcal{X}_0, \mathcal{E}_0)$ under the given $\mathbf{L}$ in advance. A recoverable theory of model (13) requires proving $(\hat{\overline{\mathcal{M}}} \times_3 \hat{\mathbf{L}}, \hat{\mathcal{E}}) = (\mathcal{X}_0, \mathcal{E}_0)$ under the final learned $\hat{\mathbf{L}}$.*

## 3.4 Solving Algorithm

This subsection derives efficient algorithms for solving the ATNN-based TRPCA and TC problem via the Alternating Direction Method of Multipliers (ADMM) framework [34].

We first write the augmented Lagrangian function of the TRPCA problem in Eq. (13) as:

$$\min_{\overline{\mathcal{M}}, \mathcal{E}, \boldsymbol{\Lambda}, \mathbf{L}^T \mathbf{L} = \boldsymbol{I}} \|\overline{\mathcal{M}}\|_* + \lambda\|\mathcal{E}\|_1 + \frac{\mu}{2}\|\mathcal{Y} - \overline{\mathcal{M}} \times_3 \mathbf{L} - \mathcal{E} + \boldsymbol{\Lambda}/\mu\|_F^2, \tag{17}$$

where $\mu$ is the penalty parameter and $\boldsymbol{\Lambda}$ is the lagrange multiplier.

Due to page limitation, we provide Algorithm 1 for solving Eq. (17) using the soft-thresholding operator $\mathcal{S}\tau(\cdot)$ [35] and the singular value soft-thresholding operator $\text{SVD}\tau(\cdot)$ [36]. Additionally, for the ATNN-TC model (13), we provide Algorithm 2 directly. For more detailed information, please refer to the supplementary material.

**Algorithm 2** ADMM for solving ATNN-TC model (13)

---

**Input**: Observation $\mathcal{Y} \in \mathbb{R}^{n_1 \times n_2 \times n_3}$ with support set $\mathbf{\Omega}$, $\mu = 0.1$, $\rho = 1.05$, $\mu_m = 1e^7\mu$, and the column number of learnable COM matrix $r_3$.
 1: Similar initialization with Algorithm 1.
 2: **while** not convergence **do**
 3: Update $\overline{\mathcal{M}}, \mathbf{L}, \mathcal{X}, \mathbf{\Lambda}$ via the similar way in Algorithm 1.
 4: Update $\mathcal{E} := \mathcal{P}_{\Omega}(\mathcal{Y} - \mathcal{X} + \mathbf{\Lambda}/\mu)$, where $\mathcal{P}_{\mathbf{\Omega}}$ is projection operator.
 5: Let $\mu = \min\{\rho\mu, \mu_m\}$.
 6: **end while**
**Output**: recovered tensors $\mathcal{X} = \overline{\mathcal{M}} \times_3 \mathbf{L}$.

---

## 3.5 Computational Complexity Analysis

As depicted in Algorithm 1 and 2, each iteration of the algorithm involves updating $\overline{\mathcal{M}}$ through small-scale SVD computations, updating $\mathbf{L}$ through small-scale SVD computation, updating $\mathcal{E}$ through soft thresholding operations, and some matrix multiplications. For a third-order tensor $\mathcal{X} \in \mathbb{R}^{n_1 \times n_2 \times n_3}$, the time complexity of the soft threshold operator is $\mathcal{O}(n_1 n_2 n_3)$, the time complexity of solving $\mathbf{L}$ is $\mathcal{O}(n_3 r_3^2)$, and the time complexity of solving $\overline{\mathcal{M}}$ is $\mathcal{O}(r_3 n_1 n_2^2)$. Thus, the overall time complexity of Algorithm 1 and 2 is $\mathcal{O}(r_3 n_1 n_2^2 + n_3 r_3^2 + n_1 n_2 n_3)$. Similarly, for the DFT-transformed TRPCA and TC models, the time complexity is $\mathcal{O}(n_3 n_1 n_2^2 + n_1 n_2 n_3)$. By comparing the two time complexities mentioned above, it can be observed that their ratio is positively correlated with $r_3/n_3$. Therefore, as the low-rank property of the tensor in the third dimension becomes stronger, the acceleration capability of the proposed algorithm in this paper also becomes stronger.

# 4 Experiments

In this section, we present numerical experiments to validate the main results stated in Theorems 2 and 3. Following the suggestion of Theorem 2, we set $\lambda = 1/\sqrt{\max\{n_1, n_2\}}$ for the TRPCA task in all experiments. However, it should be noted that further performance improvements can be achieved by carefully tuning the value of $\lambda$. The suggested value in the theory provides a useful guideline in practical applications. All simulations were conducted on a PC equipped with an Intel(R) Core(TM) i5-10600KF 4.10GHz CPU, 32 GB memory, and a GeForce RTX 3080 GPU with 10 GB memory.

## 4.1 Simulated Experiments

In this section, we will verify the correct recovery guarantee of Theorem 2 and 3 on randomly generated problems. We generate a tensor with tubal rank $R$ as a product $\mathcal{X}_0 = \mathcal{P} *_L \mathcal{Q}^T$, where $\mathcal{P}$ and $\mathcal{Q}$ are $n \times R \times n$ tensors with entries independently sampled from $\mathcal{N}(0, 1/n)$ distribution and the COM $\mathbf{L} \in \mathbb{R}^{r_3 \times n}$ is generated by orthogonalizing the random matrix with entries independently sampled from $\mathcal{N}(0, 1)$. For the TRPCA task, the support set $\mathbf{\Omega}$ (with size $m$) of $\mathcal{E}_0$ with independent Bernoulli $\pm 1$ entries is chosen uniformly at random, and the observation tensor is set as: $\mathcal{Y} = \mathcal{X}_0 + \mathcal{E}_0$. For the TC tasks, the observation $\mathcal{Y}$ is set as $\mathcal{Y} = \mathcal{P}_{\mathbf{\Omega}}(\mathcal{X}_0)$.

Next, we investigate how the tubal rank of $\mathcal{X}_0$ and the sparsity of $\mathcal{E}_0$ (and missing ratio of $\mathcal{X}_0$) affect the performance of model (12) and (13). We consider $n = 50$ and two values of $r_3$, i.e., $r_3 = 5, 20$. We vary the sparsity $\rho_s$ of $\mathcal{E}_0$ as $[0.01 : 0.01 : 0.5]$, the missing ratio $\rho$ of $\mathcal{X}_0$ as $[0.01 : 0.02 : 0.99]$, and tubal rank of $\mathcal{X}_0$ as $[1 : 1 : 50]$, respectively. For each combination of $(R, \rho_s)$ and $(R, \rho)$, we perform 10 test instances and declare a trial successful if the recovered tensor $\hat{\mathcal{X}}$ satisfies $\|\hat{\mathcal{X}} - \mathcal{X}_0\|_F / \|\mathcal{X}_0\|_F \le 0.01$. The fraction of successful recoveries are plotted in Figure 1. From Figure 1, we observe that there is a significant region where the recovery is correct for both models. Furthermore, two notable phenomena can be observed from the figure:

  1) The phase transition diagram in the first row of Figure 1 closely resembles the second row, indicating that even if we don't know the correct COM $\mathbf{L}$ in the model (12), we can learn the COM $\mathbf{L}$ through model (13).

  2) The phase transition diagram of $r_3 = 5$ is much better than that of $r_3 = 20$ for both TRPCA and TC tasks, which shows that it is necessary to consider the low-rank property of mode 3.

Table 2: Quantitative comparison of all RPCA-based competing methods under salt-and-pepper noise with the variance of **0.6**. The best and second results are highlighted in bold italics and underline.

| Methods | WDC | | | PaviaU | | | Beans | | | Cloth | | |
|---|---|---|---|---|---|---|---|---|---|---|---|---|
| | PSNR | SSIM | Times | PSNR | SSIM | Times | PSNR | SSIM | Times | PSNR | SSIM | Times |
| RPCA | 32.08 | 0.5223 | 28.99 | 24.98 | 0.8264 | 6.59 | 17.88 | 0.5920 | **17.92** | 18.47 | 0.5418 | **18.28** |
| SNN | 26.02 | 0.7178 | 136.2 | 31.34 | 0.9492 | 121.1 | 16.14 | 0.5238 | 176.2 | 16.77 | 0.5297 | 176.7 |
| KBR | 22.64 | 0.6438 | 167.2 | 20.91 | 0.4477 | 58.63 | 20.26 | 0.4162 | 252.1 | 20.91 | 0.5454 | 162.9 |
| TNN | 19.619 | 0.3728 | 419.2 | 17.09 | 0.2345 | 120.2 | 20.39 | 0.2572 | 322.4 | 15.51 | 0.1744 | 324.8 |
| CTNN | 17.21 | 0.2036 | 485.7 | 15.38 | 0.1163 | 130.7 | 15.64 | 0.1218 | 363.4 | 14.55 | 0.1162 | 353.9 |
| CTV | 33.85 | 0.9454 | 170.2 | 31.91 | 0.8872 | 41.85 | 29.35 | 0.7770 | 103.8 | 27.33 | 0.7721 | 102.2 |
| TCTV | 32.12 | 0.9090 | 815.2 | 29.62 | 0.8554 | 172.5 | **32.85** | **0.9204** | 641.3 | 27.36 | 0.7534 | 627.9 |
| Ours | **39.82** | **0.9913** | **21.34** | **35.31** | **0.9721** | **5.32** | 29.46 | 0.9108 | 29.22 | **27.53** | **0.8563** | 19.30 |

Table 3: Quantitative comparison of all competing methods under missing ratio with **0.95**. The best and second results are highlighted in bold italics and underline, respectively.

| Methods | WDC | | | PaviaU | | | Beans | | | Cloth | | |
|---|---|---|---|---|---|---|---|---|---|---|---|---|
| | PSNR | SSIM | Times | PSNR | SSIM | Times | PSNR | SSIM | Times | PSNR | SSIM | Times |
| LRMC | 18.53 | 0.4623 | **24.38** | 15.17 | 0.2834 | **2.93** | 15.96 | 0.3972 | **7.61** | 13.11 | 0.1902 | **10.95** |
| HaLRTC | 22.09 | 0.6676 | 54.37 | 18.87 | 0.3912 | 30.34 | 20.62 | 0.4542 | 64.48 | 19.01 | 0.3570 | 92.65 |
| KBR | 31.42 | 0.9022 | 1589 | 29.92 | 0.8591 | 725.7 | 26.06 | 0.7208 | 1253 | 24.14 | 0.6422 | 1292 |
| TNN | 30.01 | 0.8824 | 1019 | 26.43 | 0.7126 | 207.9 | 26.10 | 0.6712 | 419.2 | 23.46 | 0.6012 | 441.2 |
| CTNN | 33.36 | 0.9432 | 378.9 | 31.69 | 0.9172 | 114.4 | 27.61 | 0.8041 | 129.6 | 25.71 | 0.7362 | 136.2 |
| UTNN | 27.89 | 0.8652 | 487.6 | 21.80 | 0.5982 | 156.3 | 17.28 | 0.4131 | 116.6 | 16.27 | 0.3183 | 117.9 |
| FTNN | 34.87 | 0.5320 | 4376 | 32.56 | 0.9092 | 1263 | 28.48 | 0.8143 | 1587 | 25.25 | 0.7253 | 2054 |
| OITNN | 32.92 | 0.9396 | 838.2 | 28.46 | 0.8142 | 292.4 | 27.28 | 0.7442 | 448.6 | 24.06 | 0.6516 | 391.8 |
| TCTV | 33.33 | 0.9391 | 2116 | 31.81 | 0.8960 | 861.4 | **31.77** | **0.9143** | 1570 | 28.38 | 0.8442 | 1488 |
| S2NTNN | 37.36 | 0.9749 | 168.7 | **35.15** | **0.9431** | 40.78 | 27.44 | 0.7589 | 104.2 | **31.28** | **0.8679** | 113.2 |
| Ours | **38.06** | **0.9793** | 232.4 | 33.94 | 0.9293 | 58.34 | 28.83 | 0.8164 | 156.3 | 25.81 | 0.7146 | 142.4 |

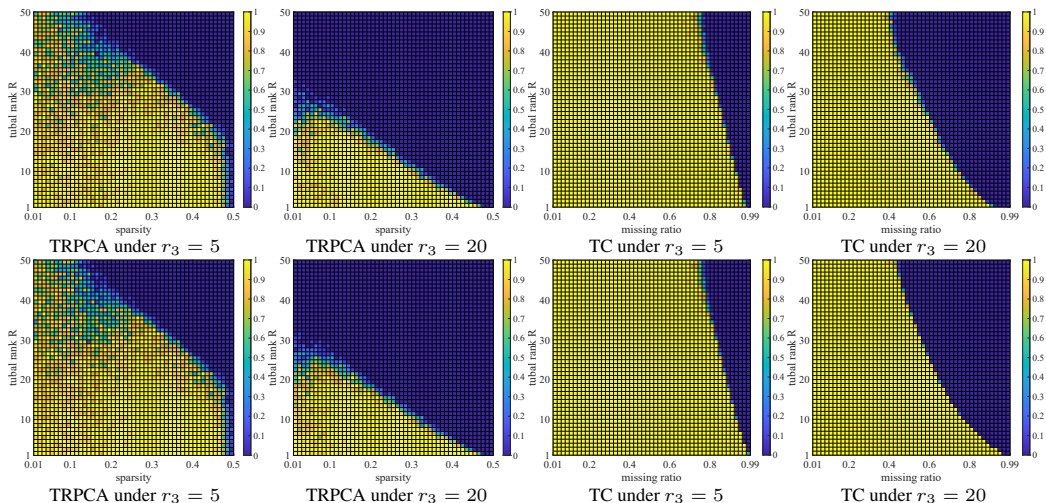

Figure 1: TRPCA and TC phase transition diagrams for varying tubal ranks of $\mathcal{X}_0$ and sparsities of $\mathcal{E}_0$ or missing ratio of $\mathcal{X}_0$. The first and second rows show the phase transition diagrams based on models (13) and (12), respectively, under different $r_3$ settings.

## 4.2 Real Experiments

To validate the effectiveness of the proposed ATNN model in tensor recovery task, we conducted experiments on various datasets, including hyperspectral images (HSI), multispectral images (MSI), color video images, and surveillance videos. Due to page limitations, we have included the results of robustness analysis, parameter settings for robustness, convergence verification, and more detailed experimental outcomes in the Supplementary Material.

For comprehensive comparison, we have included additional state-of-the-art methods except those listed in Table 1. These methods include CTV [42] and TCTV [4] for the TRPCA task, LRMC [33], HaLRTC [1], UTNN [29], and OITNN [43] for the TC task, and GODEC [37], DECOLOR [38], OMoGMF [39], RegL1 [40], and PRMF [41] for background modeling. Before conducting this experiment, the gray value of each band was normalized into [0, 1] via the max-min formula.

Table 4: AUC comparison of all competing methods on all video sequences in the Li dataset. The best and second results in each video sequence are highlighted in bold italics and underline, respectively.

| Methods | data | | | | | | | | | | Time /s |
|---|---|---|---|---|---|---|---|---|---|---|---|
| | airp. | boot. | shop. | lobb. | esca. | curt. | camp. | wate. | foun. | Average | |
| RPCA [33] | 0.8721 | 0.9168 | 0.9445 | 0.9130 | 0.9050 | 0.8722 | 0.8917 | 0.8345 | 0.9418 | 0.8991 | 2.37 |
| GODEC [37] | 0.9001 | 0.9046 | 0.9187 | 0.8556 | 0.9125 | 0.9131 | 0.8693 | 0.9370 | 0.9099 | 0.9023 | 0.64 |
| DECOLOR [38] | 0.8627 | 0.8910 | 0.9462 | 0.9241 | 0.9077 | 0.8864 | 0.8945 | 0.8000 | 0.9443 | 0.8952 | 8.29 |
| OMoGMF [39] | 0.9143 | 0.9238 | 0.9478 | 0.9252 | 0.9112 | 0.9049 | 0.8877 | 0.8958 | 0.9419 | 0.9170 | 3.92 |
| RegL1 [40] | 0.8977 | 0.9249 | 0.9423 | 0.8819 | 0.4159 | 0.8899 | 0.8871 | 0.8920 | 0.9194 | 0.8501 | 10.74 |
| PRMF [41] | 0.8905 | 0.9218 | 0.9415 | 0.8818 | 0.9065 | 0.8806 | 0.8865 | 0.8799 | 0.9166 | 0.9006 | 13.68 |
| CTV [42] | 0.9178 | 0.9107 | 0.9541 | 0.9337 | 0.9148 | 0.8710 | 0.8814 | 0.9386 | 0.9383 | 0.9180 | 10.28 |
| TNN [2] | 0.5218 | 0.5694 | 0.6605 | 0.6311 | 0.5981 | 0.5823 | 0.5464 | 0.6642 | 0.5781 | 0.5947 | 16.87 |
| CTNN [28] | 0.6859 | 0.6176 | 0.6835 | 0.6613 | 0.6582 | 0.6988 | 0.5881 | 0.5272 | 0.5450 | 0.6295 | 17.39 |
| ATNN | 0.9185 | 0.9227 | 0.9484 | 0.9362 | 0.9158 | 0.9162 | 0.8912 | 0.9152 | 0.9456 | 0.9233 | 2.32 |

Table 5: Quantitative comparison of all competing methods on color video under missing ratio with 0.95. The best and second results are highlighted in bold italics and underline, respectively.

| Methods | Akiyo | | | Foreman | | | Carphone | | | News | | |
|---|---|---|---|---|---|---|---|---|---|---|---|---|
| | PSNR | SSIM | Times | PSNR | SSIM | Times | PSNR | SSIM | Times | PSNR | SSIM | Times |
| LRMC | 10.81 | 0.2626 | 8.06 | 8.79 | 0.1192 | 7.21 | 11.57 | 0.2713 | 6.92 | 13.27 | 0.3660 | 13.41 |
| HaLRTC | 17.66 | 0.5327 | 61.04 | 15.55 | 0.3336 | 44.87 | 14.20 | 0.3448 | 42.46 | 16.43 | 0.4890 | 87.63 |
| KBR | 29.76 | 0.9118 | 689.2 | 23.97 | 0.7193 | 668.2 | 26.49 | 0.8164 | 798.2 | 26.42 | 0.8480 | 1043 |
| TNN | 31.94 | 0.9343 | 217.5 | 23.15 | 0.6052 | 181.5 | 26.27 | 0.7658 | 493.6 | 28.56 | 0.8660 | 249.6 |
| CTNN | 28.63 | 0.8463 | 192.0 | 22.13 | 0.5779 | 152.7 | 25.06 | 0.7263 | 196.2 | 25.59 | 0.7740 | 174.7 |
| UTNN | 21.72 | 0.7237 | 172.4 | 16.51 | 0.2587 | 167.6 | 20.24 | 0.5394 | 202.7 | 21.21 | 0.7060 | 162.6 |
| FTNN | 30.74 | 0.9252 | 1258 | 22.97 | 0.6781 | 1123 | 25.43 | 0.7778 | 1335 | 28.77 | 0.8770 | 1494 |
| OITNN | 32.68 | 0.9533 | 397.5 | 23.89 | 0.7206 | 296.7 | 27.14 | 0.8340 | 472.3 | 29.43 | 0.9010 | 322.3 |
| TCTV | 33.41 | 0.9542 | 874.8 | 26.69 | 0.8071 | 821.4 | 29.10 | 0.8747 | 1103 | 30.65 | 0.9170 | 772.2 |
| S2NTNN | 33.16 | 0.9520 | 168.7 | 23.57 | 0.6091 | 83.98 | 27.33 | 0.8093 | 100.7 | 29.11 | 0.8872 | 90.61 |
| Ours | 33.74 | 0.9574 | 95.89 | 24.16 | 0.6252 | 78.21 | 27.44 | 0.7773 | 80.11 | 29.72 | 0.9021 | 78.94 |

#### 4.2.1 Hyperspectral and Multispectral Image Recovery

Two HSI images, i.e., WDC [1] and PaviaU [2] datasets are used. The sizes of the two data are $256 \times 256 \times 191$ and $256 \times 256 \times 93$, respectively. Two MSI images in CAVE dataset [3], i.e., Cloth and Beans are used. The size of the two data is $512 \times 512 \times 31$.

For the TRPCA task, we conducted experiments with six different levels of salt and pepper noise variance: 0.1, 0.2, 0.3, 0.4, 0.5, and 0.6. Table 2 reports the performance metrics of each method under a variance of 0.6, demonstrating that our ATNN outperforms all competing methods. Notably, our method achieves superior performance despite only utilizing the low-rank property of tensors, surpassing the performance of CTV and TCTV, which additionally exploit the local smoothness and low-rank property of images. Furthermore, our method exhibits comparable computational efficiency to RPCA, indicating that the introduction of the learnable COM matrix effectively reduces the time complexity of the model. To better visualize the comparison, we choose three bands of HSI to form a pseudo-color image to show four representative competing methods' visual restoration performance, as shown in Figure 2. From the images, it is evident that our proposed ATNN model can effectively remove noise and preserve more detailed information.

For the TC task, since all the methods achieve very accurate recovery results when the sample ratio (SR) is high, we test four different SRs: 0.01, 0.05, 0.1 and 0.2. The metric of each tested algorithm under an SR of 0.05 is placed in Table 3. As can be seen from the metrics in the table, our proposed method excels in recovery performance and running time.

#### 4.2.2 Background Modeling from Surveillance Video

The aim of this task is to separate the background and foreground from Surveillance Video. We choose nine video sequences in Li dataset [4] with the known foreground of size $144 \times 176 \times 20$ for testing, as shown in Table 4. It can be seen from the table that our proposed model is far ahead in

---

[1] https://engineering.purdue.edu/~biehl/MultiSpec/

[2] https://www.ehu.eus/ccwintco/index.php/

[3] https://www.cs.columbia.edu/CAVE/databases/multispectral/

[4] http://perception.i2r.a-star.edu.sg/bkmodel/bkindex.html

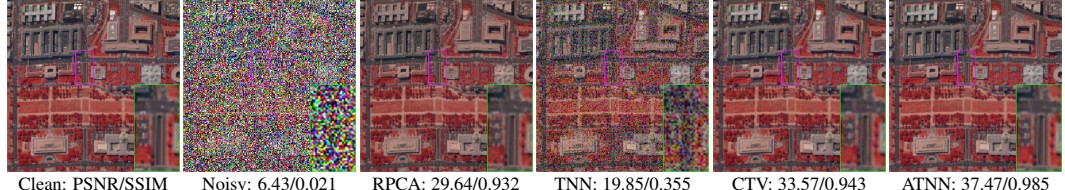

| Clean: PSNR/SSIM | Noisy: 6.43/0.021 | RPCA: 29.64/0.932 | TNN: 19.85/0.355 | CTV: 33.57/0.943 | ATNN: 37.47/0.985 |

Figure 2: Denoised images of all competing methods with bands 58-27-9 as R-G-B under sparse noise with missing percent is 0.6 on simulated WDC dataset.

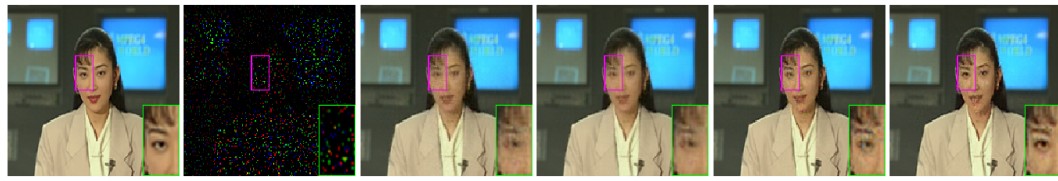

| Clean: PSNR/SSIM | Observed: 6.24/0.014 | TNN: 31.66/0.935 | OITNN: 32.60/0.958 | S2NTNN: 35.27/0.966 | ATNN: 35.52/0.971 |

Figure 3: Recovered images of all competing methods under sample ratio of 0.05 on the 10th frame of Akiyo data.

terms of evaluation metrics and running time. Even compared to the CTV model that simultaneously utilizes local smoothness and low-rank priors, our method outperforms it. It is worth noting that although tensor-based models have a higher performance ceiling than matrix-based models due to their ability to capture more complex structures, for TNN regularization, if the variation matrix is not well defined, the results can even be worse than matrix-based methods. This further highlights the necessity of learning the transform matrix.

### 4.2.3 Color Video Completion

We selected four color video sequences, namely Akiyo, Foreman, Carphone, and Mobile, from the open-source YUV video dataset[5]. To ensure efficient comparison, we considered the first 100 frames of each color video sequence. As the color video is represented as a fourth-order tensor in RGB format with dimensions $144 \times 176 \times 3 \times 100$, we reshaped it into a tensor of size $144 \times 176 \times 300$. We adopted similar sample ratio (SR) settings as mentioned in Subsection 4.2.1. The performance metrics of all competing methods are presented in Table 5. It is evident that our proposed model consistently ranks within the top three, outperforming TCTV even under the Akiyo dataset. In comparison to other TNN models with fixed transform matrices, our model exhibits superior performance and remarkable computational efficiency. Furthermore, we provided the recovered images of some competing methods in Figure 3 for better visual comparison. For the convenience of observation, we have enlarged a part of the picture and placed the repair indicator below the picture. It can be seen that our proposed ATNN model has a strong ability to preserve the local information of the data.

## 5 Conclusion

In this paper, we introduce an efficient and learnable transformed tensor nuclear norm (TNN) model with a provable recovery guarantee. Our approach leverages the low-rank property of the third mode of the tensor to represent the tensor to be repaired as a combination of a small-sized tensor and a column-orthogonal matrix. The column-orthogonal matrix serves as an adaptively learned transform matrix derived from the data. By employing the nuclear norm on the small-sized tensor, our model achieves higher computational efficiency compared to existing methods. Additionally, we provide a theoretical framework that guarantees exact recovery for our proposed model with a column-orthogonal transform matrix. Extensive experimental results demonstrate the effectiveness of our approach and the validity of our theoretical findings.

**Limitations** There are two shortcomings in our work. Firstly, the recoverable theory does not explain how the low-rank property of the third dimension of the tensor affects the model's restoration performance. Secondly, the ATNN model only learns the low-rank property of the tensor, without incorporating image priors. These two points will be the focus of our future research.

---

[5]http://trace.eas.asu.edu/yuv/

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
