# Supplementary Material for "Efficient and Learnable Transformed Tensor Nuclear Norm with Exact Recoverable Theory"

## Abstract

In this document, we first introduce the notations, preliminaries, and models in Section 1. Next, we provide the proofs of the exact recoverability theories for Tensor Robust Principal Component Analysis (TRPCA) (i.e., Theorem 2) and Tensor Completion (TC) (i.e., Theorem 3) in Sections 2 and 3, respectively. Section 4 presents detailed information about Algorithm 1 and Algorithm 2 mentioned in the manuscript. Finally, in Section 5, we provide additional experimental evidence to further validate the effectiveness of our proposed models.

## Contents

# 1   Notations and Preliminaries

## 1.1   Notations

Before completing the proofs, it is necessary to introduce some symbols that will be used throughout the document. In this paper, we denote tensors by boldface Euler script letters, e.g., $\mathcal{A}$. Matrices are denoted by boldface capital letters, e.g., $\mathbf{A}$; vectors are denoted by boldface lowercase letters, e.g., $\mathbf{a}$, and scalars are denoted by lowercase letters, e.g., a. We denote $\boldsymbol{I}_n$ as the $n \times n$ identity matrix. For a 3-order tensor $\mathcal{A} \in \mathbb{R}^{n_1 \times n_2 \times n_3}$, we denote its $(i, j, k)$-th entry as $\mathcal{A}_{ijk}$ or $a_{ijk}$ and use $\mathcal{A}(i, :, :)$, $\mathcal{A}(:, i, :)$ and $\mathcal{A}(:, :, i)$ to denote respectively the $i$-th horizontal, lateral and frontal slice (see definition in [1]). More often, the frontal slice $\mathcal{A}(:, :, i)$ is denoted compactly as $\mathbf{A}^{(i)}$. The tube is denoted as $\mathcal{A}(i, j, :)$. The mode-n unfolding matrix of $\mathcal{A}$ is denoted as $\mathbf{A}_{(n)} = \text{unfold}_n(\mathcal{A})$, and $\text{fold}_n(\mathbf{A}_{(n)}) = \mathcal{A}$, where $\text{fold}_n$ is the inverse of unfolding operator. The mode-$n$ product of a tensor $\mathcal{X} \in \mathbb{R}^{I_1 \times I_2 \times I_3}$ and a matrix $\mathbf{A} \in \mathbb{R}^{J_n \times I_n}$ is denoted as $\mathcal{Y} := \mathcal{X} \times_n \mathbf{A}$ (see definition in [2]). The inner product of between $\mathbf{A}$ and $\mathbf{B}$ is denoted as $\langle \mathbf{A}, \mathbf{B} \rangle = \text{Tr}(\mathbf{A}^T \mathbf{B})$. The inner product between $\mathcal{A}$ and $\mathcal{B}$ is denoted as $\langle \mathcal{A}, \mathcal{B} \rangle = \sum_{i=1}^{n_3} \langle \mathbf{A}^{(i)}, \mathbf{B}^{(i)} \rangle$.

Some norms of vector, matrix and tensor are used. We denote the $\|\mathcal{A}\|_1 = \sum_{ijk} |a_{ijk}|$, the infinity norm as $\|\mathcal{A}\|_\infty = \max_{ijk} |a_{ijk}|$ and the Frobenius norm as $\|\mathcal{A}\|_F = \sqrt{\sum_{ijk} |a_{ijk}|^2}$, respectively. The spectral norm of a matrix $\mathbf{A}$ is denoted as $\|\mathbf{A}\| = \max_i \sigma_i(\mathbf{A})$, where $\sigma_i(\mathbf{A})$ is the $i$-th largest singular values of $\mathbf{A}$. The matrix nuclear norm is $\|\mathbf{A}\|_* = \sum_i \sigma_i(\mathbf{A})$.

For a given scalar $x$, we denote by $\text{sgn}(x)$ the sign of $x$, which we take to be zero if $x = 0$. By extension, $\text{sgn}(\mathcal{E})$ is the matrix whose entries are the signs of those of $\mathcal{E}$. We recall that any subgradient of the $\ell_1$ norm at $\mathcal{E}$ supported on $\Omega$, is of the form

$$\text{sgn}(\mathcal{E}_0) + \mathcal{F}, \tag{1}$$

where $\mathcal{F}$ vanishes on $\Omega$, i.e. $\mathcal{P}_\Omega \mathcal{F} = 0$, and obeys $\|\mathcal{F}\|_\infty \leq 1$.

Let $\mathbf{A} = \mathbf{U}\mathbf{S}\mathbf{V}^T$ be the skinny SVD of $\mathbf{A}$. It is known that any subgradient of the nuclear norm at $\mathbf{A}$ is of the form $\mathbf{U}\mathbf{V}^T + \mathbf{W}$, where $\mathbf{U}^T \mathbf{W} = \mathbf{0}$, $\mathbf{W}\mathbf{V} = \mathbf{0}$ and $\|\mathbf{W}\| \leq 1$ [3].

Similarly, for $\mathcal{A} \in \mathbb{R}^{n_1 \times n_2 \times n_3}$ with tubal rank $R$, we also have the skinny t-SVD, i.e., $\mathcal{A} = \mathcal{U} *_L \mathcal{S} *_L \mathcal{V}^T$, where $\mathcal{U} \in \mathbb{R}^{n_1 \times R \times r_3}$, $\mathcal{S} \in \mathbb{R}^{R \times R \times r_3}$ and $\mathcal{V} \in \mathbb{R}^{n_2 \times R \times r_3}$, in which $\mathcal{U}^T *_L \mathcal{U} = \mathcal{I}$ and $\mathcal{V}^T *_L \mathcal{V} = \mathcal{I}$, where $\mathbf{L} \in \mathbb{R}^{n_3 \times r_3}$. The skinny t-SVD will be used throughout this paper. With skinny t-SVD, we introduce the subgradient of the tensor nuclear norm, which plays an important role in the proofs.

## 1.2   Subgradient of Tensor Nuclear Norm

**Theorem 1 (Subgradient of tensor nuclear norm)** *Let $\mathcal{A} \in \mathbb{R}^{n_1 \times n_2 \times n_3}$ with $\text{rank}_t(\mathcal{A}) = R$ and its skinny t-SVD be $\mathcal{A} = \mathcal{U} *_L \mathcal{S} *_L \mathcal{V}^T$ under the COM $\mathbf{L} \in \mathbb{R}^{n_3 \times r_3}$. The subdifferential (the set of*

 *subgradients) of $\|\mathcal{A}\|_*$ is:*

$$\partial\|\mathcal{A}\|_* = \{\mathcal{U} *_L \mathcal{V}^T + \mathcal{W} | \mathcal{U}^T *_L \mathcal{W} = \mathbf{0}, \mathcal{W} *_L \mathcal{V} = \mathbf{0}, \|\mathcal{W}\| \leq 1\}. \tag{2}$$

68 **Proof** *The proof is by construction. According to t-product definition in the manuscript, we have*

$$\mathcal{A} = \mathcal{U} *_L \mathcal{S} *_L \mathcal{V}^T \iff \overline{\mathbf{A}} = \overline{\mathbf{U}}\,\overline{\mathbf{S}}\,\overline{\mathbf{V}}^T, \tag{3}$$

69 *where $\overline{\mathbf{U}} = bdiag(\overline{\mathcal{U}})$, $\overline{\mathbf{V}}^T = bdiag(\overline{\mathcal{V}}^T)$ and $\overline{\mathbf{S}} = bdiag(\overline{\mathcal{S}})$. According to Eq. (11) in the*
70 *manuscript, i.e., the following equation:*

$$\|\mathcal{A}\|_* = \|\mathcal{S}\|_1 = \|\overline{\mathcal{S}}\|_* = \|\overline{\mathbf{A}}\|_* = \|\overline{\mathcal{A}}\|_*, \tag{4}$$

71 *we have $\partial\|\mathcal{A}\|_* = \partial\|\overline{\mathbf{A}}\|_*$. Since $\|\overline{\mathbf{A}}\|_*$ is diagonal block matrix, we have $\|\overline{\mathbf{A}}\|_* = \sum_{i=1}^{r_3} \|\overline{\mathbf{A}}^{(i)}\|_*$.*
72 *Performing matrix singular vector decomposition (SVD) operation on each frontal slice $\overline{\mathbf{A}}^{(i)}$, we*
73 *have $\overline{\mathbf{A}}^{(i)} = \mathbf{U}_{(i)}\mathbf{S}_{(i)}\mathbf{V}_{(i)}^T$, where $\mathbf{U}_{(i)}, \mathbf{V}_{(i)}^T$ are orthogonal matrix and $\mathbf{S}_{(i)}$ is a diagonal matrix.*
74 *Merge the SVD of each frontal slice together, we can set*

$$\overline{\mathbf{U}} = \begin{bmatrix} \mathbf{U}_{(i)} & & \\ & \ddots & \\ & & \mathbf{U}_{(r_3)} \end{bmatrix}, \overline{\mathbf{S}} = \begin{bmatrix} \mathbf{S}_{(1)} & & \\ & \ddots & \\ & & \mathbf{S}_{(r_3)} \end{bmatrix}, \overline{\mathbf{V}} = \begin{bmatrix} \mathbf{V}_{(i)} & & \\ & \ddots & \\ & & \mathbf{V}_{(r_3)} \end{bmatrix}, \tag{5}$$

75 *this gives the proof of the Theorem 1 in the manuscript.*

76 *Next, we prove the form of subgradient of $\partial\|\mathcal{A}\|_*$.*

77 *For each frontal slice $\|\overline{\mathbf{A}}^{(i)}\|_*$, its subgradient is: $\partial\|\overline{\mathbf{A}}^{(i)}\|_* = \mathbf{U}_{(i)}\mathbf{V}_{(i)}^T + \mathbf{W}_{(i)}$, where $\mathbf{W}_{(i)}$*
78 *satisfies $\mathbf{U}_{(i)}^T \mathbf{W}_{(i)} = \mathbf{0}, \mathbf{W}_{(i)}\mathbf{V}_{(i)} = \mathbf{0}$ and $\|\mathbf{W}_{(i)}\| \leq 1$. Defining*

$$\overline{\mathbf{W}} = \begin{bmatrix} \mathbf{W}_{(i)} & & \\ & \ddots & \\ & & \mathbf{W}_{(r_3)} \end{bmatrix}, \tag{6}$$

79 *we can easily obtain that $\overline{\mathbf{W}}$ satisfies $\overline{\mathbf{U}}^T \overline{\mathbf{W}} = \mathbf{0}, \overline{\mathbf{W}}\,\overline{\mathbf{V}} = \mathbf{0}$ and $\|\overline{\mathbf{W}}\| \leq 1$. Then, we have*

$$\partial\|\mathcal{A}\|_* = \partial\|\overline{\mathbf{A}}\|_* = \sum_{i=1}^{r_3}\{\mathbf{U}_{(i)}\mathbf{V}_{(i)}^T + \mathbf{W}_{(i)}\} = \overline{\mathbf{U}}\,\overline{\mathbf{V}}^T + \overline{\mathbf{W}} = \mathcal{U} *_L \mathcal{V}^T + \mathcal{W}, \tag{7}$$

80 *with $\mathcal{U}^T *_L \mathcal{W} = \mathbf{0}, \mathcal{W} *_L \mathcal{V} = \mathbf{0}, \|\mathcal{W}\| \leq 1$, where $\mathcal{U} = bfold(\overline{\mathbf{U}}) \in \mathbb{R}^{n_1 \times R \times n_3}, \mathcal{V} = bfold(\overline{\mathbf{V}}) \in$*
81 *$\mathbb{R}^{n_2 \times R \times n_3}$ and $\mathcal{W} = bfold(\overline{\mathbf{W}}) \in \mathbb{R}^{n_1 \times n_2 \times n_3}$.*

82 *This completes the proof.* ∎

83 Furthermore, we define the $\ell_{\infty,2}$-norm of the tensor $\mathcal{A}$ as

$$\|\mathcal{A}\|_{\infty,2} = \max\{\max_i \|\mathcal{A}(i,:,:)\|_F, \max_j \|\mathcal{A}(:,j,:)\|_F\}. \tag{8}$$

84 Define the projection $\mathcal{P}_\Omega(\mathcal{Z}) = \sum_{i,j,k} \delta_{ijk} z_{ijk} \mathfrak{e}_{ijk}$, where $\delta_{ijk} = 1_{(i,j,k)\in\Omega}$, where $1_{(.)}$ is the
85 indicator function. Also $\Omega^c$ denotes the complement of $\Omega$ and $\mathcal{P}_{\Omega^\perp}$ is the projection onto $\Omega^c$.
86 Denote $T$ by the set

$$T = \{\mathcal{U} *_L \mathcal{Y}^T + \mathcal{W} *_L \mathcal{V}^T, \mathcal{Y}, \mathcal{W} \in \mathbb{R}^{n \times r \times n_3}\}, \tag{9}$$

87 and by $\mathbf{T}^\perp$ its orthogonal complement. Then the projections onto $\mathbf{T}$ and $\mathbf{T}^\perp$ are respectively

$$\mathcal{P}_T(\mathcal{Z}) = \mathcal{U} *_L \mathcal{U}^T *_L \mathcal{Z} + \mathcal{Z} *_L \mathcal{U} *_L \mathcal{U}^T - \mathcal{U} *_L \mathcal{U}^T *_L \mathcal{Z} *_L \mathcal{U} *_L \mathcal{U}^T,$$
$$\mathcal{P}_{T^\perp}(\mathcal{Z}) = \mathcal{Z} - \mathcal{P}_T(\mathcal{Z}) = (\mathcal{I}_{n_1} - \mathcal{U} *_L \mathcal{U}^T) *_L \mathcal{Z} *_L (\mathcal{I}_{n_2} - \mathcal{V} *_L \mathcal{V}^T). \tag{10}$$

88 We denote $\mathring{\mathfrak{e}}_i$ as the tensor column basis, which is a tensor of size $n_1 \times 1 \times n_3$ with its $(i,1,1)$-th
89 entry equaling 1 and the rest equaling 0 [1, 4]. We also define the tensor tube basis $\dot{\mathfrak{e}}_j$, which is a
90 tensor of size $1 \times 1 \times n_3$ with its $(1,1,k)$-th entry equaling 1 and the rest equaling 0.

For $i = 1, \cdots, n_1$, $j = 1, \cdots, n_2$ and $k = 1, \cdots, n_3$, we define the random variable $\delta_{ijk} = 1_{(i,j,k)\in\Omega}$. Then the projection $\mathcal{R}_{\Omega}$ is given by

$$\mathcal{R}_{\Omega} := \frac{1}{p}\mathcal{P}_{\Omega}(\mathcal{Z}) = \sum_{i,j,k}\frac{1}{p}\delta_{ijk}z_{ijk}\mathfrak{e}_{ijk}, \tag{11}$$

where $\mathfrak{e}_{ijk} = \mathring{\mathfrak{e}}_i\mathring{\mathfrak{e}}_j\mathring{\mathfrak{e}}_k$ is an $n \times n \times n_3$ sized tensor with $(i,j,k)$-th entry equaling 1 and the rest equaling 0, . Also $\Omega^c$ denotes the complement of $\Omega$ and $\mathcal{P}_{\Omega^{\perp}}$ is the projection onto $\Omega^c$. Then we can get

$$\|\mathcal{P}_{\mathbf{T}}(\mathfrak{e}_{ijk})\|_F^2 \leq \frac{\mu R(n_1 + n_2)}{n_1 n_2} = \frac{2\mu R}{n}, \text{if } n_1 = n_2 = n, \tag{12}$$

by using the Definition 1, i.e., the following tensor incoherence condition (13).

**Definition 1 (Tensor Incoherence Conditions)** *For $\mathcal{X}_0 \in \mathbb{R}^{n_1 \times n_2 \times n_3}$ with t-SVD rank R, it has the skinny t-SVD $\mathcal{X}_0 = \mathcal{U} *_L \mathcal{S} *_L \mathcal{V}^T$. Then $\mathcal{X}_0$ is said to satisfy the tensor incoherence conditions with parameter $\mu$ if*

$$\max_{i\in[1,n_1]}\|\mathcal{U}^T *_L \mathring{\mathfrak{e}}_i\|_F \leq \sqrt{\frac{\mu R}{n_1}}, \max_{j\in[1,n_2]}\|\mathcal{V}^T *_L \mathring{\mathfrak{e}}_j\|_F \leq \sqrt{\frac{\mu R}{n_2}}, \|\mathcal{U} *_L \mathcal{V}^T\|_F \leq \sqrt{\frac{\mu R}{n_1 n_2}}. \tag{13}$$

## 1.3 Models

Two types of models are given in this paper, i.e.,

$$\begin{aligned}
(\text{TRPCA}) : \max_{\mathcal{X},\mathcal{E}} \|\mathcal{X} \times_3 \mathbf{L}^T\|_* + \lambda\|\mathcal{E}\|_1, \ s.t. \ \mathcal{Y} = \mathcal{X} + \mathcal{E}, \\
(\text{TC}) : \max_{\mathcal{X}} \|\mathcal{X} \times_3 \mathbf{L}^T\|_*, \ s.t. \ \mathcal{P}_{\Omega}(\mathcal{Y}) = \mathcal{P}_{\Omega}(\mathcal{X}),
\end{aligned} \tag{14}$$

$$\begin{aligned}
(\text{TRPCA}) : \max_{\overline{\mathcal{M}},\mathcal{S},\mathbf{L}} \|\overline{\mathcal{M}}\|_* + \lambda\|\mathcal{E}\|_1, \ s.t. \ \mathcal{Y} = \overline{\mathcal{M}} \times_3 \mathbf{L} + \mathcal{E}, \mathbf{L}^T\mathbf{L} = \mathbf{I}, \\
(\text{TC}) : \max_{\overline{\mathcal{M}},\mathbf{L}} \|\overline{\mathcal{M}}\|_*, \ s.t. \ \mathcal{P}_{\Omega}(\mathcal{Y}) = \mathcal{P}_{\Omega}(\overline{\mathcal{M}} \times_3 \mathbf{L}), \mathbf{L}^T\mathbf{L} = \mathbf{I}.
\end{aligned} \tag{15}$$

The former model (i.e., model (14)) represents the case where the COM $\mathbf{L}$ is known, while the latter model (i.e., model (15)) represents the case where the COM $\mathbf{L}$ is unknown. For model (15), we assume that the optimal solution of the TRPCA model and the TC model are given by $(\mathcal{X}^* = \overline{\mathcal{M}}^* \times_3 \mathbf{L}^*, \mathcal{E}^*)$ and $\mathcal{X}^* = \overline{\mathcal{M}}^* \times_3 \mathbf{L}^*$, respectively. The following theorem demonstrates that the representation of the ground-truth tensor $\mathcal{X}_0$ under the learned COM $\mathbf{L}^*$ preserves information.

**Theorem 2** *Suppose $\mathcal{X}_0 \in \mathbb{R}^{n_1 \times n_2 \times n_3}$ is the ground-truth tensor, it can be decomposed as $\mathcal{X}_0 = \overline{\mathcal{M}}_0 \times_3 \mathbf{L}_0$, where $\mathbf{L}_0 \in \mathbb{R}^{n_3 \times r_3}(r_3 \leq n_3)$ is the column-orthogonal matrix. Then, for any column-orthogonal matrix $\mathbf{L}$ of the same size as $\mathbf{L}_0$, $\mathcal{X}_0$ can be represented exactly.*

**Proof** *Since both $\mathbf{L}_0 \in \mathbb{R}^{n_3 \times r_3}$ and $\mathbf{L} \in \mathbb{R}^{n_3 \times r_3}(r_3 \leq n_3)$ are column-orthogonal matrices, there exists an orthogonal matrix $\mathbf{Q} \in \mathbb{R}^{r_3 \times r_3}$ that satisfies $\mathbf{L}_0 = \mathbf{LQ}$. Then we have*

$$\mathcal{X}_0 = \overline{\mathcal{M}}_0 \times_3 \mathbf{L}_0 = \overline{\mathcal{M}}_0 \times_3 (\mathbf{LQ}) = \underbrace{\overline{\mathcal{M}}_0 \times_3 \mathbf{Q}}_{\overline{\mathcal{M}}} \times_3\mathbf{L}. \tag{16}$$

*This completes the proof.* ∎

Once we get the optimal COM $\mathbf{L}^*$, the model (15) becomes model (14), so next, we prove the exact recoverability theory of model (15).

## 2 The Proof of Exact Recovery Theorem about TRPCA Model

In this section, we first introduce conditions for $(\mathcal{X}_0, \mathcal{E}_0)$ to be the unique solution to TRPCA model (14). Then we construct a dual certificate in subsection 2.2 which satisfies the conditions in subsection 2.1, and thus our main results in Theorem 2 in our paper are proved.

## 2.1 Dual Certificates

**Lemma 1** *Assume that $\|\mathcal{P}_\Omega \mathcal{P}_T\| \leq \frac{1}{2}$ and $\lambda < \frac{1}{\sqrt{n_3}}$. Then $(\mathcal{L}_0, \mathcal{S}_0)$ is the unique solution to the TRPCA problem if there is a pair $(\mathcal{W}, \mathcal{F})$ obeying*

$$\mathcal{U} *_L \mathcal{V}^T + \mathcal{W} = \lambda(\mathrm{sgn}(\mathcal{S}_0) + \mathcal{F} + \mathcal{P}_\Omega \mathcal{D}), \tag{17}$$

*with $\mathcal{P}_T \mathcal{W} = \mathbf{0}$, $\|\mathcal{W}\| \leq \frac{1}{2}$, $\mathcal{P}_\Omega \mathcal{F} = \mathbf{0}$ and $\|\mathcal{F}\|_\infty \leq \frac{1}{2}$ and $\|\mathcal{P}_\Omega \mathcal{D}\|_F \leq \frac{1}{4}$.*

**Proof** *For any $\mathcal{H} \neq 0$, $(\mathcal{X}_0 + \mathcal{H}, \mathcal{E}_0 - \mathcal{H})$ is also a feasible solution. We show that its objective is larger than that at $(\mathcal{X}_0, \mathcal{E}_0)$, hence proving that $(\mathcal{X}_0, \mathcal{E}_0)$ is the unique solution. To do this, let $\mathcal{U} *_L \mathcal{V}^T + \mathcal{W}_0$ be an arbitrary subgradient of the tensor nuclear norm at $\mathcal{X}_0$ under the COM $\mathbf{L}$, and $\mathrm{sgn}(\mathcal{E}_0) + \mathcal{F}_0$ be an arbitrary subgradient of the $\ell_1$-norm at $\mathcal{E}_0$. Then we have*

$$\|\mathcal{X}_0 + \mathcal{H}\|_* + \lambda\|\mathcal{E}_0 - \mathcal{H}\|_1 \geq \|\mathcal{X}_0\|_* + \lambda\|\mathcal{E}_0\|_1 + \left\langle \mathcal{U} *_L \mathcal{V}^T + \mathcal{W}_0, \mathcal{H} \right\rangle - \lambda \left\langle \mathrm{sgn}(\mathcal{E}_0) + \mathcal{F}_0, \mathcal{H} \right\rangle$$

*Now pick $\mathcal{W}_0$ such that $\langle \mathcal{W}_0, \mathcal{H} \rangle = \|\mathcal{P}_{T^\perp} \mathcal{H}\|_*$ and $\langle \mathcal{F}_0, \mathcal{H} \rangle = \|\mathcal{P}_{\Omega^\perp} \mathcal{H}\|$. We have*

$$\|\mathcal{X}_0 + \mathcal{H}\|_* + \lambda\|\mathcal{E}_0 - \mathcal{H}\|_1 \geq \|\mathcal{X}_0\|_* + \lambda\|\mathcal{E}_0\|_1 + \|\mathcal{P}_{T^\perp} \mathcal{H}\|_* + \lambda\|\mathcal{P}_{\Omega^\perp} \mathcal{H}\|_1$$
$$+ \left\langle \mathcal{U} *_L \mathcal{V}^T - \mathrm{sgn}(\mathcal{E}_0), \mathcal{H} \right\rangle.$$

*By assumption, we have*

$$\left| \left\langle \mathcal{U} *_L \mathcal{V}^T - \mathrm{sgn}(\mathcal{E}_0), \mathcal{H} \right\rangle \right| \leq |\langle \mathcal{W}, \mathcal{H} \rangle| + \lambda |\langle \mathcal{F}, \mathcal{H} \rangle| + \lambda |\langle \mathcal{P}_\Omega \mathcal{D}, \mathcal{H} \rangle|$$
$$\leq \beta \left( \|\mathcal{P}_{T^\perp} \mathcal{H}\|_* + \lambda\|\mathcal{P}_{\Omega^\perp} \mathcal{H}\|_1 \right) + \frac{\lambda}{4}\|\mathcal{P}_\Omega \mathcal{H}\|_F, \tag{18}$$

*where $\beta = \max(\|\mathcal{W}\|, \|\mathcal{F}\|_\infty) < \frac{1}{2}$. Thus we have*

$$\|\mathcal{X}_0 + \mathcal{H}\|_* + \lambda\|\mathcal{E}_0 - \mathcal{H}\|_1 \geq \|\mathcal{X}_0\|_* + \lambda\|\mathcal{E}_0\|_1 + \frac{1}{2} \left( \|\mathcal{P}_{T^\perp} \mathcal{H}\|_* + \lambda\|\mathcal{P}_{\Omega^\perp} \mathcal{H}\|_1 \right) - \frac{\lambda}{4}\|\mathcal{P}_\Omega \mathcal{H}\|_F.$$

*On the other hand,*

$$\|\mathcal{P}_\Omega \mathbf{H}\|_F \leq \|\mathcal{P}_\Omega \mathcal{P}_T \mathbf{H}\|_F + \|\mathcal{P}_\Omega \mathcal{P}_{T^\perp} \mathbf{H}\|_F \leq \frac{1}{2}\|\mathcal{H}\|_F + \|\mathcal{P}_{T^\perp} \mathbf{H}\|_F$$
$$\leq \frac{1}{2}\|\mathcal{P}_\Omega \mathcal{H}\|_F + \frac{1}{2}\|\mathcal{P}_{\Omega^\perp} \mathcal{H}\|_F + \|\mathcal{P}_{T^\perp} \mathbf{H}\|_F.$$

*Thus we can obtain*

$$\|\mathcal{P}_\Omega \mathbf{H}\|_F \leq \|\mathcal{P}_{\Omega^\perp} \mathcal{H}\|_F + 2\|\mathcal{P}_{T^\perp} \mathbf{H}\|_F \leq \|\mathcal{P}_{\Omega^\perp} \mathcal{H}\|_1 + 2\sqrt{n_3}\|\mathcal{P}_{T^\perp} \mathbf{H}\|_*.$$

*In conclusion,*

$$\|\mathcal{X}_0 + \mathcal{H}\|_* + \lambda\|\mathcal{E}_0 - \mathcal{H}\|_1 \geq \|\mathcal{X}_0\|_* + \lambda\|\mathcal{E}_0\|_1 + \frac{1}{2}(1 - \lambda\sqrt{n_3})\|\mathcal{P}_{T^\perp} \mathcal{H}\|_* + \frac{\lambda}{4}\|\mathcal{P}_\Omega \mathcal{H}\|_1,$$

*and the last two terms are strictly positive when $\mathbf{H} \neq \mathbf{0}$. Thus, the proof is completed.* ∎

Lemma 1 implies that it is suffices to produce a dual certificate $\mathcal{W}$ obeying

$$\begin{cases} \mathcal{W} \in T^\perp, \\ \|\mathcal{W}\| \leq \frac{1}{2}, \\ \|\mathcal{P}_\Omega(\mathcal{U} *_L \mathcal{V}^T + \mathcal{W} - \lambda\mathrm{sgn}(\mathcal{S}_0)\|_F \leq \frac{\lambda}{4}, \\ \|\mathcal{P}_{\Omega^\perp}(\mathcal{U} *_L \mathcal{V}^T + \mathcal{W})\|_\infty \leq \frac{\lambda}{2}. \end{cases} \tag{19}$$

## 2.2 Dual Certification via The Golfing Scheme

The remaining work is to construct the aforementioned dual certificates. Before introducing our construction, we first assume that $\Omega \sim \mathrm{Ber}(\rho)$, or equivalently that $\Omega^c \sim \mathrm{Ber}(1 - \rho)$. Now the distribution of $\Omega^c$ is the same as that of $\Omega^c = \Omega_1 \cup \Omega_2 \cup \cdots \cup \Omega_{j_0}$, where each $\Omega_j$ follows the Bernoulli model with parameter $q$, that is,

$$\mathbb{P}\left((i, j, k) \in \Omega\right) = \mathbb{P}(\mathrm{Bin}(j^0, q) = 0) = (1 - q)^{j_0},$$

so that the two models are the same if $\rho = (1 - q)^{j_0}$. Note that because of overlaps between the $\Omega_j$'s, $q \geq (1 - \rho)/j_0$. Now, we construct a dual certificate

$$\mathcal{W} = \mathcal{W}^{\mathcal{L}} + \mathcal{W}^{\mathcal{S}},$$

where each component is as follows:

1) Construction of $\mathcal{W}^{\mathcal{L}}$ via the Golfing scheme. Let $j_0 \geq 1$, and let $\Omega_j, 1 \leq j \leq j_0$, be defined as aforementioned so that $\Omega^c = \cup_{1 \leq j \leq j_0} \Omega_j$. Then define

$$\mathcal{W}^{\mathcal{L}} = \mathcal{P}_{T^\perp} \mathbf{Y}_{j_0}, \tag{20}$$

where

$$\mathcal{Y}_j = \mathcal{Y}_{j-1} + q^{-1} \mathcal{P}_{\Omega_j} \mathcal{P}_T \left( \mathcal{U} *_L \mathcal{V}^T - \mathbf{Y}_{j-1} \right), \mathcal{Y}_0 = \mathbf{0}. \tag{21}$$

2) Construction of $\mathbf{W}^{\mathcal{S}}$ via the Method of Least Squares. Assume that $\|\mathcal{P}_\Omega \mathcal{P}_T\| \leq \frac{1}{2}$. Then, $\|\mathcal{P}_\Omega \mathcal{P}_T \mathcal{P}_\Omega\| < \frac{1}{4}$ and thus, the operator $\mathcal{P}_\Omega - \mathcal{P}_\Omega \mathcal{P}_T \mathcal{P}_\Omega$ mapping $\Omega$ onto itself is invertible, and its inverse is denoted by $(\mathcal{P}_\Omega - \mathcal{P}_\Omega \mathcal{P}_T \mathcal{P}_\Omega)^{-1}$. We then set

$$\mathcal{W}^{\mathcal{S}} = \lambda \mathcal{P}_{T^\perp} (\mathcal{P}_\Omega - \mathcal{P}_\Omega \mathcal{P}_T \mathcal{P}_\Omega)^{-1} \text{sgn}(\mathcal{E}_0). \tag{22}$$

This is equivalent to

$$\mathcal{W}^{\mathcal{S}} = \lambda \mathcal{P}_{T^\perp} \sum_{k \geq 0} (\mathcal{P}_\Omega \mathcal{P}_T \mathcal{P}_\Omega)^k \text{sgn}(\mathcal{E}_0). \tag{23}$$

Since both $\mathcal{W}^{\mathcal{L}}, \mathcal{W}^{\mathcal{S}} \in T^\perp$ and $\mathcal{P}_\Omega \mathcal{W}^{\mathcal{S}} = \lambda \mathcal{P}_\Omega (\mathcal{I} - \mathcal{P}_T) (\mathcal{P}_\Omega - \mathcal{P}_\Omega \mathcal{P}_{T_1} \mathcal{P}_\Omega)^{-1} \text{sgn}(\mathcal{E}_0) = \lambda \text{sgn}(\mathcal{E}_0)$, we shall establish that $\mathcal{W}^{\mathcal{L}} + \mathcal{W}^{\mathcal{S}}$ is a valid dual certificate if it obeys

$$\begin{cases} \|\mathcal{W}^{\mathcal{L}} + \mathcal{W}^{\mathcal{S}}\| < \frac{1}{2}, \\ \left\| \mathcal{P}_\Omega \left( \mathcal{U} *_L \mathcal{V}^T + \mathcal{W}^{\mathcal{L}} \right) \right\|_F \leq \frac{\lambda}{4}, \\ \left\| \mathcal{P}_{\Omega^\perp} \left( \mathcal{U} *_L \mathcal{V}^T + \mathcal{W}^{\mathcal{L}} + \mathcal{W}^{\mathcal{S}} \right) \right\|_\infty \leq \frac{\lambda}{2}. \end{cases} \tag{24}$$

This can be done by using the following two lemmas.

**Lemma 2** *Assume that $\Omega \sim Ber(\rho)$ with $\rho \leq \rho_s$ for some $\rho_s > 0$. Set $j_0 = 2\lceil \log n \rceil$ (use $\log n_{(1)}$ for rectangular matrices ). Then, the $\mathcal{W}^{\mathcal{L}}$ in Eq. (20) obeys*

*(a)* $\|\mathcal{W}^{\mathcal{L}}\| < 1/4$,

*(b)* $\left\| \mathcal{P}_\Omega \left( \mathcal{U} *_L \mathcal{V}^T + \mathcal{W}^{\mathcal{L}} \right) \right\|_F \leq \frac{\lambda}{4}$,

*(c)* $\left\| \mathcal{P}_{\Omega^\perp} \left( \mathcal{U} *_L \mathcal{V}^T + \mathcal{W}^{\mathcal{L}} \right) \right\|_\infty \leq \frac{\lambda}{4}$.

**Lemma 3** *Assume $\Omega \sim Ber(\rho_s)$, and the sign of $\mathcal{S}_0$ are independent and identically distributed symmetric (and independent of $\Omega$). Then, the tensor $\mathcal{W}^{\mathcal{S}}$ with Eq. (22) obeys*

*(a)* $\|\mathcal{W}^{\mathcal{S}}\| < 1/4$,

*(b)* $\|\mathcal{P}_{\Omega^\perp} \mathcal{W}^{\mathcal{S}}\|_\infty < \lambda/4$.

### 2.3 Proofs of Dual Certification

Before proving Lemma 2 and 3, we shall list the following five useful lemmas. The proofs of these lemmas are presented in the next chapter.

**Lemma 4** *For the Bernoulli sign variable $\mathcal{M} \in \mathbb{R}^{n \times n \times n_3}$ defined as*

$$\mathcal{W}_{ijk} = \begin{cases} 1, & w.p. \quad \rho/2, \\ 0, & w.p. \quad 1 - \rho, \\ -1, & w.p. \quad \rho/2, \end{cases} \tag{25}$$

*where $\rho > 0$, there exists a function $\phi(\rho)$ satisfying $\lim_{\rho \to 0^+} \phi(\rho) = 0$, such that the following statement holds with large probability*

$$\|\mathcal{M}\| \leq \phi(\rho)\sqrt{nn_3}. \tag{26}$$

**Lemma 5** *Suppose* $\boldsymbol{\Omega} \sim Ber(\rho)$. *Then with high probability,*

$$\|\boldsymbol{\mathcal{P}_T} - \rho^{-1}\boldsymbol{\mathcal{P}_T}\boldsymbol{\mathcal{P}_\Omega}\boldsymbol{\mathcal{P}_T}\| \leq \epsilon, \tag{27}$$

*provided that* $\rho \geq C_0\epsilon^{-2}\beta\mu R \log(n)/(n)$ *for some numerical constant* $C_0 > 0$. *For the tensor of rectangular frontal slice, we need* $\rho \geq C_0\epsilon^{-2}\beta\mu R \log(n_{(1)})/(n_{(2)})$, *where* $n_{(1)} = \max\{n_1, n_2\}, n_{(2)} = \min\{n_1, n_2\}$.

**Lemma 6** *Assume that* $\boldsymbol{\Omega} \sim Ber(\rho)$, *then* $\|\boldsymbol{\mathcal{P}_\Omega}\boldsymbol{\mathcal{P}_T}\|^2 \leq \rho + \epsilon$, *provided that* $1 - \rho \geq C\epsilon^{-2}(\mu R \log(n)/n)$, *where* $C$ *is as in Lemma 5. For the tensor with frontal slice, the modification is as in Lemma 5.*

**Lemma 7** *Suppose* $\boldsymbol{\mathcal{Z}} \in \boldsymbol{T}$ *is a fixed tensor, and* $\boldsymbol{\Omega} \sim Ber(\rho_0)$. *Then with high probability,*

$$\|\boldsymbol{\mathcal{Z}} - \rho^{-1}\boldsymbol{\mathcal{P}_T}\boldsymbol{\mathcal{P}_\Omega}\boldsymbol{\mathcal{Z}}\|_\infty \leq \epsilon\|\boldsymbol{\mathcal{Z}}\|_\infty, \tag{28}$$

*provided that* $\rho \geq C_0\epsilon^{-2}\beta\mu R \log(n)/(n)$ *for some numerical constant* $C_0 > 0$. *For the tensor of rectangular frontal slice, we need* $\rho \geq C_0\epsilon^{-2}\beta\mu R \log(n_{(1)})/(n_{(2)})$.

**Lemma 8** *Suppose* $\boldsymbol{\mathcal{Z}}$ *is fixed, and* $\boldsymbol{\Omega} \sim Ber(\rho_0)$. *Then with high probability,*

$$\left\|\left(\boldsymbol{\mathcal{I}} - \rho^{-1}\boldsymbol{\mathcal{P}_\Omega}\right)\boldsymbol{\mathcal{Z}}\right\| \leq \sqrt{\frac{C_0 n \log(n)}{\rho}}\|\boldsymbol{\mathcal{Z}}\|_\infty, \tag{29}$$

*provided that* $\rho \geq C_0 \log(n)/(n)$ *for some small numerical constant* $C_0 > 0$. *For the tensor of rectangular frontal slice, we need* $\rho \geq C_0 \log(n_{(1)})/(n_{(2)})$.

### 2.3.1   Proof of Lemma 2

**Proof** *We first introduce some notations. Setting*

$$\boldsymbol{\mathcal{Z}}_j = \boldsymbol{\mathcal{U}} *_L \boldsymbol{\mathcal{V}}^T - \boldsymbol{\mathcal{P}_T}\boldsymbol{\mathcal{Y}}_j,$$

*thus* $\boldsymbol{\mathcal{Z}}_j \in \boldsymbol{T}$ *for all* $j \geq 0$. *From the definition of* $\boldsymbol{\mathcal{Y}}_j$ *(21), and* $\boldsymbol{\mathcal{Y}}_j \in \boldsymbol{\Omega}^\perp$, *we have*

$$\boldsymbol{\mathcal{Z}}_j = (\boldsymbol{\mathcal{P}_T} - q^{-1}\boldsymbol{\mathcal{P}_T}\boldsymbol{\mathcal{P}_{\Omega_j}}\boldsymbol{\mathcal{P}_T})\boldsymbol{\mathcal{Z}}_{j-1},$$
$$\boldsymbol{\mathcal{Y}}_j = \boldsymbol{\mathcal{Y}}_{j-1} + q^{-1}\boldsymbol{\mathcal{P}_{\Omega_j}}\boldsymbol{\mathcal{Z}}_{j-1}.$$

*Therefore, when*

$$q \geq C_0\epsilon^{-2}\mu R \log(n_{(1)})/(n_{(2)}), \tag{30}$$

*we have*

$$\|\boldsymbol{\mathcal{Z}}_j\|_\infty \leq \epsilon\|\boldsymbol{\mathcal{Z}}_{j-1}\|_\infty \leq \epsilon^j\|\boldsymbol{\mathcal{U}} *_L \boldsymbol{\mathcal{V}}^T\|_\infty \tag{31}$$

*by Lemma 7. When q obeys Eq. (30), we have*

$$\|\boldsymbol{\mathcal{Z}}_j\|_F \leq \epsilon\|\boldsymbol{\mathcal{Z}}_{j-1}\|_F \leq \epsilon^j\|\boldsymbol{\mathcal{U}} *_L \boldsymbol{\mathcal{V}}^T\|_F \leq \epsilon^j\sqrt{R} \tag{32}$$

*by Lemma 5. We assume* $\epsilon \leq e^{-1}$.

**proof of (a).** *Since* $\boldsymbol{\mathcal{Y}}_{j_0} = \sum_j q^{-1}\boldsymbol{\mathcal{P}_{\Omega_j}}\boldsymbol{\mathcal{Z}}_{j-1}$, *we have*

$$
\begin{aligned}
\|\boldsymbol{\mathcal{W}}^{\mathcal{L}}\| = \|\boldsymbol{\mathcal{P}_{T^\perp}}\boldsymbol{\mathcal{Y}}_{j_0}\|_\infty &\leq \sum_j \|q^{-1}\boldsymbol{\mathcal{P}_{T^\perp}}\boldsymbol{\mathcal{P}_{\Omega_j}}\boldsymbol{\mathcal{Z}}_{j-1}\| \\
&\leq \sum_j \|\boldsymbol{\mathcal{P}_{T^\perp}}(q^{-1}\boldsymbol{\mathcal{P}_{\Omega_j}}\boldsymbol{\mathcal{Z}}_{j-1} - \boldsymbol{\mathcal{Z}}_{j-1})\| \leq \sum_j \|q^{-1}\boldsymbol{\mathcal{P}_{\Omega_j}}\boldsymbol{\mathcal{Z}}_{j-1} - \boldsymbol{\mathcal{Z}}_{j-1}\| \\
&\leq C_1\sqrt{\frac{n_{(1)}\log(n_{(1)})}{q}} \sum_j \|\boldsymbol{\mathcal{Z}}_{j-1}\|_\infty \\
&\leq C_1\sqrt{\frac{n_{(1)}\log(n_{(1)})}{q}} \sum_j \epsilon^j\|\boldsymbol{\mathcal{U}} *_L \boldsymbol{\mathcal{V}}^T\|_\infty \\
&\leq \frac{C_1}{(1-\epsilon)}\sqrt{\frac{n_{(1)}\log(n_{(1)})}{q}} \|\boldsymbol{\mathcal{U}} *_L \boldsymbol{\mathcal{V}}^T\|_\infty.
\end{aligned}
\tag{33}
$$

The fourth step is according to Lemma [8] and the fifth step can be directly obtained from Eq. [(31)]. Now by using Eq. [(30)] and tensor incoherence condition, we get

$$\|\mathcal{W}^{\mathcal{L}}\| \le C_2 \epsilon$$

for some numerical constant $C_2$.

**proof of (b).** Since $\mathcal{P}_{\Omega}\mathcal{Y}_{j_0} = 0$,

$$\mathcal{P}_{\Omega}(\mathcal{U} *_L \mathcal{V}^T + \mathcal{W}^{\mathcal{L}}) = \mathcal{P}_{\Omega}(\mathcal{U} *_L \mathcal{V}^T + \mathcal{P}_{T^{\perp}}\mathcal{Y}_{j_0})$$
$$= \mathcal{P}_{\Omega}(\mathcal{U} *_L \mathcal{V}^T + \mathcal{P}_{T}\mathcal{Y}_{j_0}) = \mathcal{P}_{\Omega}(\mathcal{Z}_{j_0}).$$

By using Eqs. [(30)], we can get

$$\|\mathcal{P}_{\Omega}(\mathcal{Z}_{j_0})\|_F \le \|\mathcal{Z}_{j_0}\|_F \le \epsilon^{j_0}\sqrt{R}.$$

Since $\epsilon \le e^{-1}$, $j_0 \ge 2\log(n_{(1)})$ and $\epsilon^{j_0} \le 1/(n_{(1)})^2$, and this proves the claim.

**proof of (c).** We have $\mathcal{U} *_L \mathcal{V}^T + \mathcal{W}^{\mathcal{L}} = \mathcal{Z}_{j_0} + \mathcal{Y}_{j_0}$ and know that $\mathcal{Y}_{j_0}$ is supported on $\Omega^c$. Therefore, since $\|\mathcal{Z}_{j_0}\|_{\infty} \le \|\mathcal{Z}_{j_0}\|_F \le \lambda/8$, it suffices to show that $\|\mathcal{Y}_{j_0}\|_{\infty} \le \frac{\lambda}{8}$. To this end, we deduce

$$\|\mathcal{Y}_{j_0}\|_{\infty} \le q^{-1}\sum_j \|\mathcal{P}_{\Omega_j}\mathcal{Z}_{j_0}\|_{\infty} \le q^{-1}\sum_j \|\mathcal{Z}_{j_0}\|_{\infty} \le q^{-1}\sum_j \epsilon^j\|\mathcal{U} *_L \mathcal{V}^T\|_{\infty}.$$

Since $\|\mathcal{U} *_L \mathcal{V}^T\|_{\infty} \le \sqrt{\mu n^{-2}r}$, this gives

$$\|\mathcal{Y}_{j_0}\|_{\infty} \le C' \frac{\epsilon^2}{\sqrt{\mu r (\log(n))^2}} \tag{34}$$

for some numerical constant $C'$ whenever $q$ obeys Eq. [(30)]. By setting $\lambda = 1/\sqrt{n_{(1)}}$, $\|\mathcal{Y}_{j_0}\|_{\infty} \le \lambda/8$ if

$$\epsilon \le C \left(\frac{\mu r(\log(n_{(1)}))^2}{n_{(2)}}\right)^{\frac{1}{4}}.$$

We have seen that (a) and (b) are satisfied if $\epsilon$ is sufficiently small and $j_0 \ge 2\log(n_{(1)})$. For (c), we can take $\epsilon$ on the order of $\left(\mu r(\log(n_{(1)}))^2/(n_{(2)})\right)^{\frac{1}{4}}$, which could be sufficiently small as well provided that $\rho_r$ in Eq. [(30)] in the manuscript is sufficiently small. Note that everything is consistent, since $C_0\epsilon^{-2}\mu r \log(n_{(1)})/(n_{(2)}) < 1$. ∎

### 2.3.2   Proof of Lemma [3]

**Proof** We denote $\mathcal{M} = sgn(\mathcal{E}_0)$ distributed as

$$\mathcal{M}_{ijk} = \begin{cases} 1, & w.p. \quad \rho/2, \\ 0, & w.p. \quad 1 - \rho, \\ -1, & w.p. \quad \rho/2. \end{cases} \tag{35}$$

Note that for any $\sigma > 0$, $\{\|\mathcal{P}_{\Omega}\mathcal{P}_T\| \le \sigma\}$ holds with high probability provided that $\rho$ is sufficiently small, see Lemma [5].

**1. Proof of (a).** By construction,

$$\mathcal{W}^{\mathcal{S}} = \lambda\mathcal{P}_{T^{\perp}}\mathcal{M} + \lambda\mathcal{P}_{T^{\perp}}\sum_{k \ge 1}(\mathcal{P}_{\Omega}\mathcal{P}_T\mathcal{P}_{\Omega})^k\mathcal{M} := \mathcal{P}_{T^{\perp}}\mathcal{W}_0^{\mathcal{S}} + \mathcal{P}_{T^{\perp}}\mathcal{W}_1^{\mathcal{S}}. \tag{36}$$

Note that $\|\mathcal{P}_{T^{\perp}}\mathcal{W}_0^{\mathcal{S}}\| \le \|\mathcal{W}_0^{\mathcal{S}}\| = \lambda\|\mathcal{M}\|$ and $\|\mathcal{P}_{T^{\perp}}\mathcal{W}_1^{\mathcal{S}}\| \le \|\mathcal{W}_1^{\mathcal{S}}\| = \lambda\|\mathcal{R}(\mathcal{M})\|$, where $\mathcal{R} = \sum_{k \ge 1}(\mathcal{P}_{\Omega}\mathcal{P}_T\mathcal{P}_{\Omega})^k$. Now, we will respectively show that $\lambda\|\mathcal{M}\|$ and $\lambda\|\mathcal{R}(\mathcal{M})\|$ are small enough when $\rho$ is sufficiently small for $\lambda = 1/\sqrt{n}$. Therefore, $\|\mathcal{W}^{\mathcal{S}}\| \le 1/4$.

**1) Bound $\lambda\|\mathcal{M}\|$.** By using Lemma [4] directly, we have that $\lambda\|\mathcal{M}\| \le \phi(\rho)$ is sufficiently small given $\lambda = 1/sqrtn$ and $\rho$ is sufficiently small.

**216** *2) Bound $\|\mathcal{R}(\mathcal{M})\|$. For simplicity, let $\mathcal{Z} = \mathcal{R}(\mathcal{M})$, we have*

$$\|\mathcal{Z}\| = \|\overline{\mathbf{Z}}\| = \sup_{\boldsymbol{x} \in \mathbb{S}^{nr_3-1}} \|\overline{\mathbf{Z}}\boldsymbol{x}\|_2. \tag{37}$$

**217** *The optimal $\boldsymbol{x}$ to Eq. (37) is an eigenvector of $\overline{\mathbf{Z}} * \overline{\mathbf{Z}}$. Since $\overline{\mathbf{Z}}$ is a block diagonal matrix, the optimal $\boldsymbol{x}$*
**218** *has a block sparse structure, i.e., $\boldsymbol{x} \in B = \{\boldsymbol{x} \in \mathbb{R}^{nr_3} | \boldsymbol{x} = [\boldsymbol{x}_1^T, \cdots, \boldsymbol{x}_i^T, \cdots, \boldsymbol{x}_{r_3}^T], \text{ with } \boldsymbol{x}_i \in \mathbb{R}^n,$*
**219** *and there exist $j$ such that $\boldsymbol{x}_j \neq \mathbf{0}$ and $\boldsymbol{x}_i \neq \mathbf{0}, i \neq j\}$. Note that $\|\boldsymbol{x}\|_2 = \|\boldsymbol{x}_j\|_2 = 1$. Let $N$ be the*
**220** *$1/2$-net for $\mathbb{S}^{n-1}$ of size at most $5^n$ (see Lemma 5.2 in [5]). Then the $1/2$-net, denote as $N'$, for $B$*
**221** *has the size at most $r_3.5^n$. We have*

$$\|\mathcal{R}(\mathcal{M})\| = \|bdiag(\overline{\mathcal{R}(\mathcal{M})})\| = \sup_{\boldsymbol{x}, \boldsymbol{y} \in B} \left\langle \boldsymbol{x}, bdiag(\overline{\mathcal{R}(\mathcal{M})})\boldsymbol{y} \right\rangle$$
$$= \sup_{\boldsymbol{x}, \boldsymbol{y} \in B} \left\langle \boldsymbol{x}\boldsymbol{y}^*, bdiag(\overline{\mathcal{R}(\mathcal{M})}) \right\rangle = \sup_{\boldsymbol{x}, \boldsymbol{y} \in B} \left\langle bdiag^*(\boldsymbol{x}\boldsymbol{y}^*), \overline{\mathcal{R}(\mathcal{M})} \right\rangle, \tag{38}$$

**222** *where $bdiag^*$, the joint operator of bdiag (see definition in the manuscript), maps the block diagonal*
**223** *matrix $\boldsymbol{x}\boldsymbol{y}^*$ to a tensor of size $n \times n \times n_3$. Let $\mathcal{Z}' = bdiag^*(\boldsymbol{x}\boldsymbol{y}^*)$ and $\mathcal{Z} = \mathcal{Z}' \times_3 \mathbf{L}$. We have*

$$\|\mathcal{R}(\mathcal{M})\| = \sup_{\boldsymbol{x}, \boldsymbol{y} \in B} \left\langle \mathcal{Z}', \overline{\mathcal{R}(\mathcal{M})} \right\rangle = \sup_{\boldsymbol{x}, \boldsymbol{y} \in B} \left\langle \mathcal{Z}', \mathcal{R}(\mathcal{M}) \right\rangle$$
$$= \sup_{\boldsymbol{x}, \boldsymbol{y} \in B} \left\langle \mathcal{R}(\mathcal{Z}), \mathcal{M} \right\rangle = \sup_{\boldsymbol{x}, \boldsymbol{y} \in N'} 4 \left\langle \mathcal{R}(\mathcal{Z}), \mathcal{M} \right\rangle. \tag{39}$$

**224** *For a fixed pair $(\boldsymbol{x}, \boldsymbol{y})$ of unit-normed vectors, define the random variable*

$$X(\boldsymbol{x}, \boldsymbol{y}) = 4 \left\langle \mathcal{R}(\mathcal{Z}), \mathcal{M} \right\rangle. \tag{40}$$

**225** *Conditional on $\Omega = supp(\mathcal{M})$, the sign of $\mathcal{M}$ are independent and identically distributed symmetric*
**226** *and Hoeffding's inequality gives*

$$\mathbb{P}(|X(\boldsymbol{x}, \boldsymbol{y})| > t|\Omega) \leq 2 \exp\left(\frac{-2t^2}{\|4\mathcal{R}(\mathcal{Z})\|_F^2}\right). \tag{41}$$

**227** *Note that $\|4\mathcal{R}(\mathcal{Z})\|_F^2 \leq 4\|\mathcal{R}\|\|\mathcal{Z}\|_F = 4\|\mathcal{R}\|\|\mathcal{Z}'\|_F = 4\|\mathcal{R}\|$. Therefore, we have*

$$\mathbb{P}\left(\sup_{\boldsymbol{x}, \boldsymbol{y} \in N'} |X(\boldsymbol{x}, \boldsymbol{y})| > t|\Omega\right) \leq 2|N'|^2 \exp\left(\frac{-t^2}{8\|\mathcal{R}\|^2}\right). \tag{42}$$

**228** *Hence,*

$$\mathbb{P}\left(\|\mathcal{R}(\mathcal{M})\| > t|\Omega\right) \leq 2|N'|^2 \exp\left(\frac{-t^2}{8\|\mathcal{R}\|^2}\right). \tag{43}$$

**229** *On the event $\{\|\mathcal{P}_{\Omega}\mathcal{P}_T\| \leq \sigma\}$, $\|\mathcal{R}\| \leq \sum_{k \geq 1} \sigma^{2k} = \frac{\sigma^2}{1-\sigma^2}$, therefore, unconditionally,*

$$\mathbb{P}\left(\|\mathcal{R}(\mathcal{M})\| > t\right) \leq 2|N'|^2 \exp\left(\frac{-\gamma^2 t^2}{8}\right) + \mathbb{P}(\|\mathcal{P}_{\Omega}\mathcal{P}_T\| \geq \sigma), \gamma = \frac{1-\sigma^2}{\sigma^2}$$
$$= 2r_3^2.5^{2n} \exp\left(\frac{-\gamma^2 t^2}{8}\right) + \mathbb{P}(\|\mathcal{P}_{\Omega}\mathcal{P}_T\| \geq \sigma). \tag{44}$$

**230** *Let $t = c\sqrt{n}$, where $c$ can be a small absolute constant. Then the above inequality implies that*
**231** *$\|\mathcal{R}(\mathcal{M})\| \leq t$ with high probability.*

**232** *2. Proof of (b). Observe that*

$$\mathcal{P}_{\Omega^\perp}\mathcal{W}^{\mathcal{S}} = -\lambda \mathcal{P}_{\Omega^\perp}\mathcal{P}_T(\mathcal{P}_{\Omega} - \mathcal{P}_{\Omega}\mathcal{P}_T\mathcal{P}_{\Omega})^{-1}\mathcal{M}. \tag{45}$$

**233** *Note for $(i, j, k) \in \Omega^c, \mathcal{W}_{ijk}^{\mathcal{S}} = \langle \mathcal{W}, \mathfrak{e}_{ijk} \rangle$, and we have $\mathcal{W}_{ijk}^{\mathcal{S}} = \lambda \langle \mathcal{Q}(i, j, k), \mathcal{W} \rangle$, where*
**234** *$\mathcal{Q}(i, j, k)$ is the tensor $-(\mathcal{P}_{\Omega} - \mathcal{P}_{\Omega}\mathcal{P}_T\mathcal{P}_{\Omega})^{-1}\mathcal{P}_{\Omega}\mathcal{P}_T(\mathfrak{e}_{ijk})$. Conditional on $\Omega = supp(\mathcal{M})$,*
**235** *the signs of $\mathcal{M}$ are independent and identically distributed symmetric, and the Hoeffding's inequality*
**236** *gives*

$$\mathbb{P}\left(\|\mathcal{W}_{ijk}^{\mathcal{S}}\| > t\lambda|\Omega\right) \leq 2 \exp\left(-\frac{2t^2}{\|\mathcal{Q}(i, j, k)\|_F^2}\right), \tag{46}$$

*and*

$$\mathbb{P}\left(\sup_{i,j,k}\|\boldsymbol{\mathcal{W}}^{\boldsymbol{\mathcal{S}}}_{ijk}\| > t\lambda/n_3|\boldsymbol{\Omega}\right) \leq 2n^2 n_3 \exp\left(-\frac{2t^2}{\sup_{i,j,k}\|\boldsymbol{\mathcal{Q}}(i,j,k)\|^2_F}\right), \tag{47}$$

*By using Eq. (12), we have*

$$\|\boldsymbol{\mathcal{P}}_{\boldsymbol{\Omega}}\boldsymbol{\mathcal{P}}_{\boldsymbol{T}}(\mathfrak{e}_{ijk})\|_F \leq \|\boldsymbol{\mathcal{P}}_{\boldsymbol{\Omega}}\boldsymbol{\mathcal{P}}_{\boldsymbol{T}}\|\|\boldsymbol{\mathcal{P}}_{\boldsymbol{T}}(\mathfrak{e}_{ijk})\|_F \leq \sigma\sqrt{\frac{2\mu R}{n}}, \tag{48}$$

*on the event $\{\|\boldsymbol{\mathcal{P}}_{\boldsymbol{\Omega}}\boldsymbol{\mathcal{P}}_{\boldsymbol{T}}\| \leq \sigma\}$. On the same event, we have $\|(\boldsymbol{\mathcal{P}}_{\boldsymbol{\Omega}} - \boldsymbol{\mathcal{P}}_{\boldsymbol{\Omega}}\boldsymbol{\mathcal{P}}_{\boldsymbol{T}}\boldsymbol{\mathcal{P}}_{\boldsymbol{\Omega}})^{-1}\|(1 - \sigma^2)^{-1}$*
*and thus $\|\boldsymbol{\mathcal{Q}}(i,j,k)\|^2_F \leq \frac{2\sigma^2}{(1-\sigma^2)^2}\frac{\mu R}{n}$. Then, unconditionally,*

$$\mathbb{P}\left(\sup_{i,j,k}|\boldsymbol{\mathcal{W}}^{\boldsymbol{\mathcal{S}}}_{ijk}| > t\lambda\right) \leq 2n^2 n_3 \exp\left(-\frac{n\gamma^2 t^2}{\mu R}\right) + \mathbb{P}(\|\boldsymbol{\mathcal{P}}_{\boldsymbol{\Omega}}\boldsymbol{\mathcal{P}}_{\boldsymbol{T}}\| \geq \sigma), \tag{49}$$

*where $\gamma^2 = \frac{(1-\sigma^2)^2}{2\sigma^2}$. This proves the claim when $\mu R \leq \rho'_r n \log(n)^{-1}$ and $\rho'_r$ is sufficiently small.*

## 2.4 Proof of Some Lemmas

Before the proof, we introduce a theorem.

**Theorem 3 (Noncommutative Bernstein Inequality)** Let $\mathbf{X}_1, \mathbf{X}_2, \cdots, \mathbf{X}_L$ be independent zero-mean random matrices of dimension $d_1 \times d_2$. Suppose $\|\mathbf{X}_k\| \leq M$ and

$$\rho^2_k = \max\{\|\mathbb{E}[\mathbf{X}_k\mathbf{X}^T_k]\|, \|\mathbb{E}[\mathbf{X}^T_k\mathbf{X}_k]\|\} \tag{50}$$

almost surely for all $k$. Then for any $\tau > 0$,

$$\mathbb{P}\left[\|\sum_{k=1}^{L}\mathbf{X}_k\| > \tau\right] \leq (d_1 + d_2)\exp\left(\frac{-\tau^2/2}{\sum_{k=1}^{L}\rho^2_k + M\tau/3}\right) \tag{51}$$

This theorem is a corollary of a Chernoff bound for finite dimension operators developed by [6]. An extension of this theorem [7] states that if

$$\max\left\{\|\sum_{k=1}^{L}\mathbf{X}_k\mathbf{X}^T_k\|, \|\sum_{k=1}^{L}\mathbf{X}^T_k\mathbf{X}_k\|\right\} \leq \sigma^2 \tag{52}$$

and let

$$\tau = \sqrt{4c\sigma^2 \log(d_1 + d_2)} + cM\log(d_1 + d_2) \tag{53}$$

for any $c > 0$. Then Eq. (51) becomes

$$\mathbb{P}\left[\|\sum_{k=1}^{L}\mathbf{X}_k\| > \tau\right] \leq (d_1 + d_2)^{-(c-1)}. \tag{54}$$

### 2.4.1 Proof of Lemma 4

**Proof** *The proof has three steps.*

***Step 1: Approximation.*** *We first introduce some notations. Let $\boldsymbol{f}^*$ be the $i$-th row of $\mathbf{L}^T \in \mathbb{R}^{n_3 \times r_3}$, and $\mathbf{M}^{\boldsymbol{H}} = [\mathbf{M}^{\boldsymbol{H}}_1; \mathbf{M}^{\boldsymbol{H}}_2; \cdots; \mathbf{M}^{\boldsymbol{H}}_n] \in \mathbb{R}^{nr_3 \times n}$ be a matrix unfolded by $\boldsymbol{\mathcal{M}}$, where $\mathbf{M}^{\boldsymbol{H}}_i \in \mathbb{R}^{r_3 \times n}$ is the $i$-th horizontal slice of $\boldsymbol{\mathcal{M}}$, i.e., $[\mathbf{M}^{\boldsymbol{H}}_i]_{kj} = \boldsymbol{\mathcal{M}}_{ijk}$. Consider that $\overline{\boldsymbol{\mathcal{M}}} = \boldsymbol{\mathcal{M}} \times_3 \mathbf{L}^T$, we have $\overline{\boldsymbol{M}}_i = [\boldsymbol{f}^*_i\mathbf{M}^{\boldsymbol{H}}_1; \boldsymbol{f}^*_i * \mathbf{M}^{\boldsymbol{H}}_2; \cdots; \boldsymbol{f}^*_i\mathbf{M}^{\boldsymbol{H}}_n]$, where $\overline{\boldsymbol{M}}_i \in \mathbb{R}^{n \times n}$ is the $i$-th frontal slice of $\boldsymbol{\mathcal{M}}$. Note that*

$$\|\boldsymbol{\mathcal{M}}\| = \|\overline{\mathbf{M}}\| = \max_{i=1,\cdots,r_3}\|\overline{\mathbf{M}}_i\|. \tag{55}$$

*Let $N$ be the $1/2$-net for $\mathbb{S}^{n-1}$ of size at most $5^n$ (see Lemma 5.2 in [5]). Then Lemma 5.3 in [5] gives*

$$\|\overline{\mathbf{M}}_i\| \leq 2\max_{\boldsymbol{x} \in N}\|\overline{\mathbf{M}}_i\boldsymbol{x}\|_2. \tag{56}$$

*So we consider to bound $\|\overline{\mathbf{M}}_i\boldsymbol{x}\|_2$.*

**Step 2: Concentration.** *We can express* $\|\overline{\mathbf{M}}_i \boldsymbol{x}\|_2^2$ *as a sum of independent random variables*

$$\|\overline{\mathbf{M}}_i \boldsymbol{x}\|_2^2 = \sum_{j=1}^n (\boldsymbol{f}_i^* \mathbf{M}_j^H \boldsymbol{x})^2 := \sum_{j=1}^n z_j^2, \tag{57}$$

*where* $z_j = \langle \mathbf{M}_j^H, \boldsymbol{f}_i \boldsymbol{x}^* \rangle, j = 1, \cdots, n$ *are independent sub-gaussian random variables with* $\mathbb{E}(z_j^2) = \rho \|\boldsymbol{f}_i \boldsymbol{x}^*\|_f^2 = \rho r_3$. *Using Eq. (25), we have*

$$|[\mathbf{M}_j^H]_{kl}| = \begin{cases} 1, & w.p.\ \rho, \\ 0, & w.p.\ 1 - \rho. \end{cases} \tag{58}$$

*Thus, the sub-gaussian norm of* $[\mathbf{M}_j^H]_{kl}$*, denoted as* $\| \cdot \|_{\psi_2}$*, is*

$$\|[\mathbf{M}_j^H]_{kl}\|_{\psi_2} = \sup_{p \geq 1} p^{-0.5} (\mathbb{E}[|[\mathbf{M}_j^H]_{kl}|^p])^{1/p} = \sup_{p \geq 1} p^{-0.5} \rho^{1/p}. \tag{59}$$

*Define the function* $\phi(x) = x^{-1/2} \rho^{1/x}$ *on* $[1, +\infty)$*. The only stationary point occurs at* $x^* = \log \rho^{-2}$*. Thus,*

$$\phi(x) \leq \max\{\phi(1), \phi(x^*)\} = \max \left( \rho, (\log \rho^{-2})^{-0.5} \rho^{1/\log \rho^{-2}} \right) := \psi(\rho). \tag{60}$$

*Therefore,* $\|[\mathbf{M}_j^H]_{kl}\|_{\psi_2} \leq \psi(\rho)$*. Consider that* $z_j$ *is a sum of independent centered sub-gaussian random variables* $[\mathbf{M}_j^H]_{kl}$*'s, bu using Lemma 5.9 in [5], we have* $\|z_j\|_{\psi_2}^2 \leq c_1 (\psi(\rho))^2 r_3$*, where* $c_1$ *is an absolute constant. Therefore, by Remark 5.18 and Lemma 5.14 in [5],* $z_j^2 - \rho r_3$ *are independent centered sub-exponential random variables with* $\|z_j^2 - \rho r_3\|_{\psi_1} \leq 2\|z_j\|_{\psi_1}^2 \leq 4\|z_j\|_{\psi_2}^2 \leq 4c_1 (\psi(\rho))^2 r_3$*.*

*Now, we use an exponential deviation inequality, Corollary 5.17 in [5], to control the sum of Eq. (57). We have*

$$
\begin{aligned}
\mathbb{P}(|\|\overline{\mathbf{M}}_i \boldsymbol{x}\|_2^2 - \rho n r_3| \geq tn) &= \mathbb{P}\left( \left| \sum_{j=1}^n (z_j^2 - \rho r_3) \right| \right) \\
&\leq 2 \exp \left( -c_2 n \min \left( \left( \frac{t}{4c_1 (\psi(\rho))^2 r_3} \right)^2, \frac{t}{4c_1 (\psi(\rho))^2 r_3} \right) \right),
\end{aligned}
\tag{61}
$$

*where* $c_2 > 0$*. Let* $t = c_3 (\psi(\rho))^2 r_3$ *for some absolute constant* $c_3$*, we have*

$$\mathbb{P}(|\|\overline{\mathbf{M}}_i \boldsymbol{x}\|_2^2 - \rho n r_3| \geq c_3 (\psi(\rho))^2 n r_3) \leq 2 \exp \left( -c_2 n \min \left( \left( \frac{c_3}{4c_1} \right)^2, \frac{c_3}{4c_1} \right) \right). \tag{62}$$

**Step 3: Union bound.** *Taking the union bound over all* $\boldsymbol{x}$ *in the Net* $N$ *of cardinality* $|N| \leq 5^n$*, we obtain*

$$\mathbb{P}\left( \left| \max_{\boldsymbol{x} \in N} \|\overline{\mathbf{M}}_i \boldsymbol{x}\|_2^2 - \rho n r_3 \right| \geq c_3 (\psi(\rho))^2 n r_3 \right) \leq 2 \cdot 5^n \cdot \exp \left( -c_2 n \min \left( \left( \frac{c_3}{4c_1} \right)^2, \frac{c_3}{4c_1} \right) \right). \tag{63}$$

*Furthermore, taking the union over all* $i = 1, \cdots, r_3$*, we have*

$$
\begin{aligned}
&\mathbb{P}\left( \max_i \left| \max_{\boldsymbol{x} \in N} \|\overline{\mathbf{M}}_i \boldsymbol{x}\|_2^2 - \rho n r_3 \right| \geq c_3 (\psi(\rho))^2 n r_3 \right) \\
&\leq 2 \cdot 5^n \cdot r_3 \cdot \exp \left( -c_2 n \min \left( \left( \frac{c_3}{4c_1} \right)^2, \frac{c_3}{4c_1} \right) \right).
\end{aligned}
\tag{64}
$$

*This implies that, with high probability (when the constant* $c_3$ *is large enough),*

$$\max_i \max_{\boldsymbol{x} \in N} \|\overline{\mathbf{M}}_i \boldsymbol{x}\|_2^2 \leq \left( \rho + c_3 (\psi(\rho))^2 \right) n r_3 \tag{65}$$

*Let* $\phi(\rho) = 2\sqrt{\rho + c_3 (\psi(\rho))^2}$ *and it satisfies* $\lim_{\rho \to 0^+} \phi(\rho) = 0$ *by using Eq. (60). The proof is completed by further combing Eq. (55), (56) and (65).* ∎

### 2.4.2 Proof of Lemma 5

**Proof** *For any tensor $\boldsymbol{\mathcal{Z}}$, we can write*

$$(\rho^{-1}\boldsymbol{\mathcal{P}_T}\boldsymbol{\mathcal{P}_\Omega}\boldsymbol{\mathcal{P}_T} - \boldsymbol{\mathcal{P}_T})\boldsymbol{\mathcal{Z}} = \sum_{ijk}(\rho^{-1}\delta_{ijk} - 1)\langle \mathfrak{e}_{ijk}, \boldsymbol{\mathcal{P}_T}\boldsymbol{\mathcal{Z}}\rangle \boldsymbol{\mathcal{P}_T}(\mathfrak{e}_{ijk}) := \sum_{ijk}\boldsymbol{\mathcal{H}}_{ijk}(\boldsymbol{\mathcal{Z}}) \quad (66)$$

*where $\boldsymbol{\mathcal{H}}_{ijk} : \mathbb{R}^{n \times n \times n_3} \to \mathbb{R}^{n \times n \times n_3}$ is a self-adjoint random operator with $\mathbb{E}[\boldsymbol{\mathcal{H}}_{ijk}] = \mathbf{0}$. Define the matrix operator $\overline{\mathbf{H}}_{ijk} : \mathbb{B} \to \mathbb{B}$, where $\mathbb{B} = \{\overline{\mathbf{B}} : \boldsymbol{\mathcal{B}} \in \mathbb{R}^{n \times n \times n_3}\}$ denotes the set consists of block diagonal matrices with the blocks as the frontal slices of $\overline{\boldsymbol{\mathcal{B}}}$, as*

$$\overline{\mathbf{H}}_{ijk}(\overline{\mathbf{Z}}) = (\rho^{-1}\delta_{ijk} - 1)\langle \mathfrak{e}_{ijk}, \boldsymbol{\mathcal{P}_T}(\boldsymbol{\mathcal{Z}})\rangle \, bdiag(\overline{\boldsymbol{\mathcal{P}_T}(\mathfrak{e}_{ijk})}). \quad (67)$$

*By the above definitions, we have $\boldsymbol{\mathcal{H}}_{ijk} = \overline{\mathbf{H}}_{ijk}$ and $\|\sum_{ijk}\boldsymbol{\mathcal{H}}_{ijk}\| = \|\sum_{ijk}\overline{\mathbf{H}}_{ijk}\|$. Also, $\overline{\mathbf{H}}_{ijk}$ is self-adjoint and $\mathbb{E}[\overline{\mathbf{H}}_{ijk}] = 0$. To prove the result by the non-commutative Bernstein inequality, we need to bound $\|\overline{\mathbf{H}}_{ijk}\|$ and $\|\sum_{ijk}\mathbb{E}[\overline{\mathbf{H}}_{ijk}^2]\|$. First, we have*

$$\|\overline{\mathbf{H}}_{ijk}\| = \sup_{\|\overline{\mathbf{Z}}\|_F = 1}\|\overline{\mathbf{H}}_{ijk}(\overline{\mathbf{Z}})\|_F \leq \sup_{\|\overline{\mathbf{Z}}\|_F = 1}\|\boldsymbol{\mathcal{P}_T}(\mathfrak{e}_{ijk})\|_F\|bdiag(\overline{\boldsymbol{\mathcal{P}_T}(\mathfrak{e}_{ijk})})\|_F\|\boldsymbol{\mathcal{Z}}\|_F$$
$$= \sup_{\|\overline{\mathbf{Z}}\|_F = 1}\|\boldsymbol{\mathcal{P}_T}(\mathfrak{e}_{ijk})\|_F^2\|\overline{\mathbf{Z}}\|_F \leq \frac{2\mu R}{n\rho}, \quad (68)$$

*where the last inequality use Eq. (12). On the other hand, by direct computation, we have $\mathbf{H}_{ijk}^2(\overline{\mathbf{Z}}) = (\rho^{-1}\delta_{ijk} - 1)^2\langle \mathfrak{e}_{ijk}, \boldsymbol{\mathcal{P}_T}(\boldsymbol{\mathcal{Z}})\rangle\langle \mathfrak{e}_{ijk}, \boldsymbol{\mathcal{P}_T}(\mathfrak{e}_{ijk})\rangle \, bdiag(\overline{\boldsymbol{\mathcal{P}_T}(\mathfrak{e}_{ijk})})$. Note that $\mathbb{E}[(\rho^{-1}\delta_{ijk} - 1)^2] \leq \rho^{-1}$. We have*

$$\left\|\sum_{ijk}\mathbb{E}[\overline{\mathbf{H}}_{ijk}^2(\overline{\mathbf{Z}})]\right\|_F \leq \rho^{-1}\left\|\sum_{ijk}\langle \mathfrak{e}_{ijk}, \boldsymbol{\mathcal{P}_T}(\boldsymbol{\mathcal{Z}})\rangle\langle \mathfrak{e}_{ijk}, \boldsymbol{\mathcal{P}_T}(\mathfrak{e}_{ijk})\rangle \, bdiag(\overline{\boldsymbol{\mathcal{P}_T}(\mathfrak{e}_{ijk})})\right\|_F$$
$$\leq \rho^{-1}\|\boldsymbol{\mathcal{P}_T}(\mathfrak{e}_{ijk})\|_F^2\left\|\sum_{ijk}\langle \mathfrak{e}_{ijk}, \boldsymbol{\mathcal{P}_T}(\boldsymbol{\mathcal{Z}})\rangle\right\|_F \quad (69)$$
$$\leq \rho^{-1}\|\boldsymbol{\mathcal{P}_T}(\mathfrak{e}_{ijk})\|_F^2\|\boldsymbol{\mathcal{P}_T}(\boldsymbol{\mathcal{Z}})\|_F \leq \rho^{-1}\|\boldsymbol{\mathcal{P}_T}(\mathfrak{e}_{ijk})\|_F^2\|\boldsymbol{\mathcal{Z}}\|_F$$
$$\leq \rho^{-1}\|\boldsymbol{\mathcal{P}_T}(\mathfrak{e}_{ijk})\|_F^2\|\overline{\mathbf{Z}}\|_F \leq \frac{2\mu R}{n\rho}\|\overline{\mathbf{Z}}\|_F.$$

*By Theorem 3, we have*

$$\mathbb{P}[\|\rho^{-1}\boldsymbol{\mathcal{P}_T}\boldsymbol{\mathcal{P}_\Omega}\boldsymbol{\mathcal{P}_T} - \boldsymbol{\mathcal{P}_T}\| > \epsilon] = \mathbb{P}\left[\left\|\sum_{ijk}\boldsymbol{\mathcal{H}}_{ijk}\right\| > \epsilon\right] = \mathbb{P}\left[\left\|\sum_{ijk}\overline{\mathbf{H}}_{ijk}\right\| > \epsilon\right]$$
$$\leq 2nr_3\exp\left(-\frac{3}{8}\cdot\frac{\epsilon^2}{2\mu R/(n\rho)}\right) \leq 2(n)^{1 - 3C_0/16}, \quad (70)$$

*where the last inequality uses $\rho \geq C_0\epsilon^{-2}\mu R\log(n)/(n)$. Thus, $\|\rho^{-1}\boldsymbol{\mathcal{P}_T}\boldsymbol{\mathcal{P}_\Omega}\boldsymbol{\mathcal{P}_T} - \boldsymbol{\mathcal{P}_T}\| \leq \epsilon$ holds with high probability for smoe numerical constant $C_0$.*

∎

### 2.4.3 Proof of Lemma 6

**Proof** *From the proof of Lemma 5 (i.e., the last subsection), we have*

$$\|\boldsymbol{\mathcal{P}_T} - (1 - \rho)^{-1}\boldsymbol{\mathcal{P}_T}\boldsymbol{\mathcal{P}_{\Omega^\perp}}\boldsymbol{\mathcal{P}_T}\| \leq \epsilon, \quad (71)$$

*provided that $1 - \rho \geq C_0\epsilon^{-2}(\mu R\log(n)/n)$. Note that $\boldsymbol{\mathcal{I}} = \boldsymbol{\mathcal{P}_\Omega} + \boldsymbol{\mathcal{P}_{\Omega^\perp}}$, we have*

$$\|\boldsymbol{\mathcal{P}_T} - (1 - \rho)^{-1}\boldsymbol{\mathcal{P}_T}\boldsymbol{\mathcal{P}_{\Omega^\perp}}\boldsymbol{\mathcal{P}_T}\| = (1 - \rho)^{-1}(\boldsymbol{\mathcal{P}_T}\boldsymbol{\mathcal{P}_\Omega}\boldsymbol{\mathcal{P}_T} - \rho\boldsymbol{\mathcal{P}_T}). \quad (72)$$

*Then, by the triangular inequality*

$$\|\boldsymbol{\mathcal{P}_T}\boldsymbol{\mathcal{P}_\Omega}\boldsymbol{\mathcal{P}_T}\| \leq \epsilon(1 - \rho) + \rho\|\boldsymbol{\mathcal{P}_T}\| = \rho + \epsilon(1 - \rho). \quad (73)$$

*This proof is completed by using $\|\boldsymbol{\mathcal{P}_\Omega}\boldsymbol{\mathcal{P}_T}\|^2 = \|\boldsymbol{\mathcal{P}_T}\boldsymbol{\mathcal{P}_\Omega}\boldsymbol{\mathcal{P}_T}\|$.*

∎

### 2.4.4 Proof of Lemma 7

**Proof** *For any tensor $\mathcal{Z} \in T$, we write*

$$\rho^{-1}\mathcal{P}_{\Omega}\mathcal{P}_{T}(\mathcal{Z}) = \sum_{ijk} \rho^{-1}\delta_{ijk}z_{ijk}\mathcal{P}_{T}(\mathfrak{e}_{ijk}).$$

*The $(a, b, c)$-th entry of $\rho^{-1}\mathcal{P}_{\Omega}\mathcal{P}_{T}(\mathcal{Z}) - \mathcal{Z}$ can be written as a sum of independent random variables, i.e.,*

$$\left\langle \rho^{-1}\mathcal{P}_{\Omega}\mathcal{P}_{T}(\mathcal{Z}) - \mathcal{Z}, \mathfrak{e}_{abc}\right\rangle = \sum_{ijk}(\rho^{-1}\delta_{ijk} - 1)z_{ijk}\left\langle \mathcal{P}_{T}(\mathfrak{e}_{ijk}), \mathfrak{e}_{abc}\right\rangle := \sum_{ijk} t_{ijk}, \tag{74}$$

*where $t_{ijk}$'s are independent and $\mathbb{E}(t_{ijk}) = 0$. Now next bound $|t_{ijk}|$ and $|\sum_{ijk}\mathbb{E}[t_{ijk}^2]|$. First*

$$|t_{ijk}| \le \rho^{-1}\|\mathcal{Z}\|_{\infty}\|\mathcal{P}_{T}(\mathfrak{e}_{ijk})\|_F\|\mathcal{P}_{T}(\mathfrak{e}_{abc})\|_F \le \frac{2\mu R}{n\rho}\|\mathcal{Z}\|_{\infty}. \tag{75}$$

*Second, we have*

$$\left|\sum_{ijk}\mathbb{E}[t_{ijk}^2]\right| \le \rho^{-1}\|\mathcal{Z}\|_{\infty}^2\sum_{ijk}\left\langle \mathcal{P}_{T}(\mathfrak{e}_{ijk}), \mathfrak{e}_{abc}\right\rangle = \rho^{-1}\|\mathcal{Z}\|_{\infty}^2\sum_{ijk}\left\langle \mathfrak{e}_{ijk}, \mathcal{P}_{T}(\mathfrak{e}_{abc})\right\rangle$$
$$= \rho^{-1}\|\mathcal{Z}\|_{\infty}^2\|\mathcal{P}_{T}(\mathfrak{e}_{abc})\|_F^2 \le \frac{2\mu R}{n\rho}\|\mathcal{Z}\|_{\infty}^2. \tag{76}$$

*Let $\epsilon \le 1$. By Theorem 3, we have*

$$\mathbb{P}\left[|[\rho^{-1}\mathcal{P}_{T}\mathcal{P}_{\Omega}(\mathcal{Z}) - \mathcal{Z}]_{abc}| \ge \epsilon\|\mathcal{Z}\|_{\infty}\right] = \mathbb{P}\left[\left|\sum_{ijk}[t_{ijk}]\right| \ge \epsilon\|\mathcal{Z}\|_{\infty}\right]$$
$$\le 2\exp\left(-\frac{3}{8}\cdot\frac{\epsilon^2\|\mathcal{Z}\|_{\infty}^2}{2\mu R\|\mathcal{Z}\|_{\infty}^2/(n\rho)}\right) \le 2n^{-\frac{3}{16}C_0}, \tag{77}$$

*where the last inequality uses $\rho \ge C_0\epsilon^{-2}\mu R\log(n)/n$. Thus, $\|\rho^{-1}\mathcal{P}_{T}\mathcal{P}_{\Omega}(\mathcal{Z}) - \mathcal{Z}\|_{\infty} \le \epsilon\|\mathcal{Z}\|_{\infty}$ holds with high probability for some numerical constant $C_0$.* ∎

### 2.4.5 Proof of Lemma 8

**Proof** *Denote the tensor $\mathcal{H}_{ijk} = (1 - \rho^{-1}\delta_{ijk})z_{ijk}\mathfrak{e}_{ijk}$. Then we have*

$$(\mathcal{I} - \rho^{-1}\mathcal{P}_{\Omega})\mathcal{Z} = \sum_{ijk}\mathcal{H}_{ijk}. \tag{78}$$

*Note that $\delta_{ijk}$'s are independent random scalars. Thus, $\mathcal{H}_{ijk}$'s are independent random tensors and $\overline{\mathbf{H}}_{ijk}$'s are independent random matrices. Observe that $\mathbb{E}[\overline{\mathbf{H}}_{ijk}] = 0$ and $\|\overline{\mathbf{H}}_{ijk}\| \le \rho^{-1}\|\mathcal{Z}\|_{\infty}$, we have*

$$\left\|\sum_{ijk}\mathbb{E}[\overline{\mathbf{H}}_{ijk}^*\overline{\mathbf{H}}_{ijk}]\right\| = \left\|\sum_{ijk}\mathbb{E}[\mathcal{H}_{ijk}^*\mathcal{H}_{ijk}]\right\| = \left\|\sum_{ijk}\mathbb{E}[(1 - \rho^{-1}\delta_{ijk})^2]z_{ijk}^2(\dot{\mathfrak{e}}_j *_L \dot{\mathfrak{e}}_j^*)\right\|$$
$$= \left\|\frac{1 - \rho}{\rho}\sum_{ijk}z_{ijk}^2(\dot{\mathfrak{e}}_j *_L \dot{\mathfrak{e}}_j^*)\right\| \le \frac{nn_3}{\rho}\|\mathcal{Z}\|_{\infty}^2. \tag{79}$$

*A similar calculation yields $\left\|\sum_{ijk}\mathbb{E}[\overline{\mathbf{H}}_{ijk}^*\overline{\mathbf{H}}_{ijk}]\right\| \le \rho^{-1}nn_3\|\mathcal{Z}\|_{\infty}^2$. Let $t = \sqrt{C_0nn_3\log(nn_3)/\rho}\|\mathcal{Z}\|_{\infty}$. When $\rho \ge C_0\log(n)/n$, we apply Theorem 3 and obtain*

$$\mathbb{P}[\|(\mathcal{I} - \rho^{-1}\mathcal{P}_{\Omega})\mathcal{Z}\| > t] = \mathbb{P}\left[\left\|\sum_{ijk}\mathcal{H}_{ijk}\right\| > t\right] = \mathbb{P}\left[\left\|\sum_{ijk}\overline{\mathbf{H}}_{ijk}\right\| > t\right]$$
$$\le 2nr_3\exp\left(-\frac{3}{8}\cdot\frac{C_0nn_3\log(nn_3)\|\mathcal{Z}\|_{\infty}^2/\rho}{nn_3\|\mathcal{Z}\|_{\infty}^2/\rho}\right) \le 2(nr_3)^{1-3C_0/8}. \tag{80}$$

*Thus, $\|(\mathcal{I} - \rho^{-1}\mathcal{P}_{\Omega})\mathcal{Z}\| > t$ holds with high probability for some numerical constant $C_0$.* ∎

## 3 The Proof of Exact Recovery Theorem about TC Model

The following fact is used frequently in this section.

**Lemma 9** *Suppose $\mathbf{\Omega} \sim \mathrm{Ber}(p)$. Then with high probability,*

$$\|\mathcal{P}_T \mathcal{R}_\Omega \mathcal{P}_T - \mathcal{P}_T\| \leq \epsilon, \tag{81}$$

*provided that $p \geq c_0 \epsilon^{-2} (\mu R \log(n))/(n)$ for some numerical constant $c_0 > 0$. For the tensor of rectangular frontal slices, we need $p \geq c_0 \epsilon^{-2} (\mu R \log(n_{(1)}))/(n_{(2)})$, where $n_{(1)} = \max\{n_1, n_2\}, n_{(2)} = \min\{n_1, n_2\}$.*

**Proof** *By replacing $\rho^{-1}\mathcal{P}$ with $\mathcal{R}_\Omega$ in Lemma 5, this Lemma holds.* ∎

**Lemma 10** *Suppose that $\mathbf{\mathcal{Z}}$ is fixed, and $\mathbf{\Omega} \sim \mathrm{Ber}(p)$. Then, with high probability,*

$$\|(\mathcal{R}_\Omega - \mathcal{I})\mathbf{\mathcal{Z}}\| \leq c \left( \frac{\log(n)}{p} \|\mathbf{\mathcal{Z}}\|_\infty + \frac{\log(n)}{p} \|\mathbf{\mathcal{Z}}\|_{\infty,2} \right), \tag{82}$$

*for some numerical constant $c > 0$.*

**Lemma 11** *Suppose that $\mathbf{\mathcal{Z}} \in T$ is a fixed tensor and $\mathbf{\Omega} \sim \mathrm{Ber}(p)$. Then, with high probability,*

$$\|\mathcal{P}_T \mathcal{R}_\Omega \mathbf{\mathcal{Z}} - \mathbf{\mathcal{Z}}\|_{\infty,2} \leq \frac{1}{2}\sqrt{\frac{n}{\mu R}} \|\mathbf{\mathcal{Z}}\|_\infty + \frac{1}{2}\|\mathbf{\mathcal{Z}}\|_{\infty,2}, \tag{83}$$

*provided that $p \geq c_0 \mu R \log(n)/(n)$.*

**Lemma 12** *Suppose that $\mathbf{\mathcal{Z}} \in T$ is a fixed tensor and $\mathbf{\Omega} \sim \mathrm{Ber}(p)$. Then, with high probability,*

$$\|\mathbf{\mathcal{Z}} - \mathcal{P}_T \mathcal{R}_\Omega(\mathbf{\mathcal{Z}})\|_\infty \leq \epsilon \|\mathbf{\mathcal{Z}}\|_\infty, \tag{84}$$

*provided that $p \geq c_0 \epsilon^{-2} (\mu R \log(n))/(n)$ (for the tensor of rectangular frontal slice, $p \geq c_0 \epsilon^{-2} (\mu R \log(n_{(1)}))/(n_{(2)})$ for some numerical constant $c_0 > 0$.)*

**Proof** *By replacing $\rho^{-1}\mathcal{P}$ with $\mathcal{R}_\Omega$ in Lemma 7, this Lemma holds.* ∎

### 3.1 The Proof of Exact Recovery Theorem about TC Model

**Proposition 1** *The tensor $\mathbf{\mathcal{X}}_0$ is the unique optimal solution of TC model (14) if the following conditions hold: 1. $\|\mathcal{P}_T \mathcal{R}_\Omega \mathcal{P}_T - \mathcal{P}_T\| \leq \frac{1}{2}$.*

*2. There exists a dual certificate $\mathbf{\mathcal{W}} \in \mathbb{R}^{n_1 \times n_2 \times n_3}$ which satisfies $\mathcal{P}_\Omega(\mathbf{\mathcal{W}}) = \mathbf{\mathcal{W}}$ and*

*(a) $\|\mathcal{P}_{\Omega^\perp}(\mathbf{\mathcal{W}})\| \leq \frac{1}{2}$.*

*(b) $\|\mathcal{P}_\Omega(\mathbf{\mathcal{W}} - \mathbf{\mathcal{U}} *_L \mathbf{\mathcal{V}}^T\| \leq \frac{1}{4}\sqrt{\frac{p}{r_3}}$.*

**Proof** *Consider any feasible solution $\mathbf{\mathcal{X}}$ to TC model (14). Let $\mathbf{\mathcal{G}}$ be an $n \times n \times n_3$ tensor which satisfies $\|\mathcal{P}_{\Omega^\perp}\mathbf{\mathcal{G}}\| = 1$ and $\langle \mathcal{P}_{\Omega^\perp}\mathbf{\mathcal{G}}, \mathcal{P}_{\Omega^\perp}(\mathbf{\mathcal{X}} - \mathbf{\mathcal{X}}_0)\rangle = \|\mathcal{P}_{\Omega^\perp}(\mathbf{\mathcal{X}} - \mathbf{\mathcal{X}}_0)\|_*$. Such $\mathbf{\mathcal{G}}$ always exists by duality between the tensor nuclear norm and tensor spectral norm. Note that $\mathbf{\mathcal{U}} *_L \mathbf{\mathcal{V}}^T + \mathcal{P}_{\Omega^\perp}\mathbf{\mathcal{G}}$ is a subgradient of $\mathbf{\mathcal{Z}}$ and $\mathbf{\mathcal{Z}} = \mathbf{\mathcal{X}}_0$, we have*

$$\|\mathbf{\mathcal{X}}\|_* - \|\mathbf{\mathcal{X}}_0\|_* \geq \left\langle \mathbf{\mathcal{U}} *_L \mathbf{\mathcal{V}}^T + \mathcal{P}_{\Omega^\perp}\mathbf{\mathcal{G}}, \mathbf{\mathcal{X}} - \mathbf{\mathcal{X}}_0 \right\rangle. \tag{85}$$

*We also have $\langle \mathbf{\mathcal{W}}, \mathbf{\mathcal{X}} - \mathbf{\mathcal{X}}_0\rangle = \langle \mathcal{P}_\Omega \mathbf{\mathcal{W}}, \mathcal{P}_\Omega(\mathbf{\mathcal{X}} - \mathbf{\mathcal{X}}_0)\rangle = 0$ since $\mathcal{P}_\Omega(\mathbf{\mathcal{W}}) = \mathbf{\mathcal{W}}$. It follows that*

$$\begin{aligned}
\|\mathbf{\mathcal{X}}\|_* - \|\mathbf{\mathcal{X}}_0\|_* &\geq \left\langle \mathbf{\mathcal{U}} *_L \mathbf{\mathcal{V}}^T + \mathcal{P}_{\Omega^\perp}\mathbf{\mathcal{G}} - \mathbf{\mathcal{W}}, \mathbf{\mathcal{X}} - \mathbf{\mathcal{X}}_0 \right\rangle \\
&= \|\mathcal{P}_{\Omega^\perp}(\mathbf{\mathcal{X}} - \mathbf{\mathcal{X}}_0)\|_* + \left\langle \mathbf{\mathcal{U}} *_L \mathbf{\mathcal{V}}^T - \mathcal{P}_T\mathbf{\mathcal{W}}, \mathbf{\mathcal{X}} - \mathbf{\mathcal{X}}_0 \right\rangle - \langle \mathcal{P}_{T^\perp}\mathbf{\mathcal{W}}, \mathbf{\mathcal{X}} - \mathbf{\mathcal{X}}_0 \rangle \\
&\geq \|\mathcal{P}_{\Omega^\perp}(\mathbf{\mathcal{X}} - \mathbf{\mathcal{X}}_0)\|_* + \|\mathbf{\mathcal{U}} *_L \mathbf{\mathcal{V}}^T - \mathcal{P}_T\mathbf{\mathcal{W}}\|_F \|\mathcal{P}_T(\mathbf{\mathcal{X}} - \mathbf{\mathcal{X}}_0)\|_F \\
&\quad - \|\mathcal{P}_{T^\perp}\mathbf{\mathcal{W}}\|\|\mathcal{P}_{T^\perp}(\mathbf{\mathcal{X}} - \mathbf{\mathcal{X}}_0)\|_* \\
&\geq \frac{1}{2}\|\mathcal{P}_{\Omega^\perp}(\mathbf{\mathcal{X}} - \mathbf{\mathcal{X}}_0)\|_* - \frac{1}{4}\sqrt{\frac{p}{r_3}}\|\mathcal{P}_T(\mathbf{\mathcal{X}} - \mathbf{\mathcal{X}}_0)\|_F
\end{aligned}$$

$$\tag{86}$$

 *where the last inequality uses Conditions (1) and (2) in the proposition. Now, by using Lemma 13*
 *below, we have*

$$\|\boldsymbol{\mathcal{X}}\|_* - \|\boldsymbol{\mathcal{X}}_0\|_* \geq \frac{1}{2}\|\boldsymbol{\mathcal{P}}_{\boldsymbol{\Omega}^{\perp}}(\boldsymbol{\mathcal{X}} - \boldsymbol{\mathcal{X}}_0)\|_* - \frac{1}{4}\sqrt{\frac{p}{r_3}}\sqrt{\frac{2r_3}{p}}\|\boldsymbol{\mathcal{P}}_{\boldsymbol{T}}(\boldsymbol{\mathcal{X}} - \boldsymbol{\mathcal{X}}_0)\|_* \tag{87}$$
$$> \frac{1}{8}\|\boldsymbol{\mathcal{P}}_{\boldsymbol{\Omega}^{\perp}}(\boldsymbol{\mathcal{X}} - \boldsymbol{\mathcal{X}}_0)\|_*.$$

346 *Note that the right hand side of the above inequality is strictly positive for all $\boldsymbol{\mathcal{X}}$ with $\boldsymbol{\mathcal{P}}_{\boldsymbol{\Omega}}(\boldsymbol{\mathcal{X}} - \boldsymbol{\mathcal{X}}_0) =$*
347 *0 and $\boldsymbol{\mathcal{X}} \neq \boldsymbol{\mathcal{X}}_0$. Otherwise, we must have $\boldsymbol{\mathcal{P}}_{\boldsymbol{T}}(\boldsymbol{\mathcal{X}} - \boldsymbol{\mathcal{X}}_0) = \boldsymbol{\mathcal{X}} - \boldsymbol{\mathcal{X}}_0$ and $\boldsymbol{\mathcal{P}}_{\boldsymbol{T}}\boldsymbol{\mathcal{R}}_{\boldsymbol{\Omega}}\boldsymbol{\mathcal{P}}_{\boldsymbol{T}}(\boldsymbol{\mathcal{X}} - \boldsymbol{\mathcal{M}}) = 0$,*
348 *contradicting the assumption $\|\boldsymbol{\mathcal{P}}_{\boldsymbol{T}}\boldsymbol{\mathcal{R}}_{\boldsymbol{\Omega}}\boldsymbol{\mathcal{P}}_{\boldsymbol{T}} - \boldsymbol{\mathcal{P}}_{\boldsymbol{T}}\| \leq \frac{1}{2}$. Therefore, $\boldsymbol{\mathcal{X}}_0$ is the unique optimum.* ∎

349 **Lemma 13** *If $\|\boldsymbol{\mathcal{P}}_{\boldsymbol{T}}\boldsymbol{\mathcal{R}}_{\boldsymbol{\Omega}}\boldsymbol{\mathcal{P}}_{\boldsymbol{T}} - \boldsymbol{\mathcal{P}}_{\boldsymbol{T}}\| \leq \frac{1}{2}$, then we have*

$$\|\boldsymbol{\mathcal{P}}_{\boldsymbol{T}}\boldsymbol{\mathcal{Z}}\|_F \leq \sqrt{\frac{2r_3}{p}}\|\boldsymbol{\mathcal{P}}_{\boldsymbol{T}^{\perp}}\boldsymbol{\mathcal{Z}}\|_*, \forall \boldsymbol{\mathcal{Z}} \in \{\boldsymbol{\mathcal{Z}}' : \boldsymbol{\mathcal{P}}_{\boldsymbol{\Omega}}(\boldsymbol{\mathcal{Z}}') = 0\}. \tag{88}$$

350 **Proof** *We deduce*

$$\|\sqrt{p}\boldsymbol{\mathcal{R}}_{\boldsymbol{\Omega}}\boldsymbol{\mathcal{P}}_{\boldsymbol{T}}\boldsymbol{\mathcal{Z}}\|_F = \sqrt{\langle(\boldsymbol{\mathcal{P}}_{\boldsymbol{T}}\boldsymbol{\mathcal{R}}_{\boldsymbol{\Omega}}\boldsymbol{\mathcal{P}}_{\boldsymbol{T}} - \boldsymbol{\mathcal{P}}_{\boldsymbol{T}})\boldsymbol{\mathcal{Z}}, \boldsymbol{\mathcal{P}}_{\boldsymbol{T}}\boldsymbol{\mathcal{Z}}\rangle + \langle\boldsymbol{\mathcal{P}}_{\boldsymbol{T}}\boldsymbol{\mathcal{Z}}, \boldsymbol{\mathcal{P}}_{\boldsymbol{T}}\boldsymbol{\mathcal{Z}}\rangle}$$
$$= \sqrt{\|\boldsymbol{\mathcal{P}}_{\boldsymbol{T}}\boldsymbol{\mathcal{Z}}\|_F^2 - \|\boldsymbol{\mathcal{P}}_{\boldsymbol{T}}\boldsymbol{\mathcal{R}}_{\boldsymbol{\Omega}}\boldsymbol{\mathcal{P}}_{\boldsymbol{T}} - \boldsymbol{\mathcal{P}}_{\boldsymbol{T}}\|\|\boldsymbol{\mathcal{P}}_{\boldsymbol{T}}\boldsymbol{\mathcal{Z}}\|_F^2} \tag{89}$$
$$\geq \frac{1}{\sqrt{2}}\|\boldsymbol{\mathcal{P}}_{\boldsymbol{T}}\boldsymbol{\mathcal{Z}}\|_F$$

351 *where the last inequality uses $\|\boldsymbol{\mathcal{P}}_{\boldsymbol{T}}\boldsymbol{\mathcal{R}}_{\boldsymbol{\Omega}}\boldsymbol{\mathcal{P}}_{\boldsymbol{T}} - \boldsymbol{\mathcal{P}}_{\boldsymbol{T}}\| \leq \frac{1}{2}$. On the other hand, $\boldsymbol{\mathcal{P}}_{\boldsymbol{\Omega}}(\boldsymbol{\mathcal{Z}}) = 0$ implies*
352 *that $\boldsymbol{\mathcal{R}}_{\boldsymbol{\Omega}}(\boldsymbol{\mathcal{Z}}) = 0$ and thus*

$$\|\sqrt{p}\boldsymbol{\mathcal{R}}_{\boldsymbol{\Omega}}\boldsymbol{\mathcal{P}}_{\boldsymbol{T}}\boldsymbol{\mathcal{Z}}\|_F = \|\sqrt{p}\boldsymbol{\mathcal{R}}_{\boldsymbol{\Omega}}\boldsymbol{\mathcal{P}}_{\boldsymbol{T}^{\perp}}\boldsymbol{\mathcal{Z}}\|_F \leq \frac{1}{\sqrt{p}}\|\boldsymbol{\mathcal{P}}_{\boldsymbol{T}}\boldsymbol{\mathcal{Z}}\|_F \leq \sqrt{\frac{r_3}{p}}\|\boldsymbol{\mathcal{P}}_{\boldsymbol{T}}\boldsymbol{\mathcal{Z}}\|_*, \tag{90}$$

353 *where the last inequality uses*

$$\|\boldsymbol{\mathcal{A}}\|_F = \|\overline{\boldsymbol{A}}\|_F \leq \|\overline{\boldsymbol{A}}\|_* \leq \|\boldsymbol{\mathcal{A}}\|_*. \tag{91}$$

354 *The proof is completed by combining Eq. (89) and (90).* ∎

355 New we give the completed proof of the Exact Recovery Theorem (i.e., Theorem 3 in the manuscript)
356 about TC model.

357 **Proof** *First, as shown in Lemma 9, the Condition 1 of Proposition 1 holds with high probability.*
358 *Now we construct a dual certificate $\boldsymbol{\mathcal{W}}$ which satisfies Condition 2 in Proposition 1. We do this using*
359 *the Golfing Scheme. For the choice of $p$ in Theorem 3 in the manuscript, we have*

$$p \geq \frac{c_0 \mu R (\log(n))^2}{n} \geq \frac{1}{n}, \tag{92}$$

360 *for some sufficiently large $c_0 > 0$. Set $t_0 := 20\log(n)$. Assume that the set $\boldsymbol{\Omega}$ of observed entries*
361 *is generated from $\boldsymbol{\Omega} = \cup_{t=1}^{t_0}\boldsymbol{\Omega}_t$, where each $t$ and tensor index $(i, j, k), \mathbb{P}[(i, j, k) \in \boldsymbol{\Omega}_t] = q :=$*
362 *$1 - (1 - p)^{1/t_0}$ and is independent of all others. Clearly this $\boldsymbol{\Omega}$ has the same distribution as the*
363 *original model. Let $\boldsymbol{\mathcal{W}}_0 := \boldsymbol{0}$ and for $t = 1, \cdots, t_0$, define*

$$\boldsymbol{\mathcal{W}}_t = \boldsymbol{\mathcal{W}}_{t-1} + \boldsymbol{\mathcal{R}}_{\boldsymbol{\Omega}_t}\boldsymbol{\mathcal{P}}_{\boldsymbol{T}}(\boldsymbol{\mathcal{U}} *_L \boldsymbol{\mathcal{V}}^T - \boldsymbol{\mathcal{P}}_{\boldsymbol{\Omega}}\boldsymbol{\mathcal{W}}_{t-1}), \tag{93}$$

364 *where the operator $\boldsymbol{\mathcal{R}}_{\boldsymbol{\Omega}_t}$ is defined analogously to $\boldsymbol{\mathcal{R}}_{\boldsymbol{\Omega}}$ as $\boldsymbol{\mathcal{R}}_{\boldsymbol{\Omega}_t}(\boldsymbol{\mathcal{Z}}) := \sum_{ijk} q^{-1}\mathbb{1}_{(i,j,k)\in\boldsymbol{\Omega}_t} z_{ijk}\mathfrak{e}_{ijk}$.*
365 *Then the dual certificate is given by $\boldsymbol{\mathcal{W}} := \boldsymbol{\mathcal{W}}_{t_0}$. We have $\boldsymbol{\mathcal{P}}_{\boldsymbol{\Omega}}(\boldsymbol{\mathcal{W}}) = \boldsymbol{\mathcal{W}}$ by construction. To prove*
366 *Theorem 2, we only need to show that $\boldsymbol{\mathcal{W}}$ satisfies Conditions 2 in Proposition 1 w.h.p.*

367 **Validating Condition 2(b).** *Denote $\boldsymbol{\mathcal{D}}_t := \boldsymbol{\mathcal{U}} *_L \boldsymbol{\mathcal{V}}^T - \boldsymbol{\mathcal{P}}_{\boldsymbol{T}}\boldsymbol{\mathcal{W}}_k$ for $t = 0, \cdots, t_0$. By the definition*
368 *of $\boldsymbol{\mathcal{W}}_k$, we have $\boldsymbol{\mathcal{D}}_0 = \boldsymbol{\mathcal{U}} *_L \boldsymbol{\mathcal{V}}^T$ and*

$$\boldsymbol{\mathcal{D}}_t = (\boldsymbol{\mathcal{P}}_{\boldsymbol{T}} - \boldsymbol{\mathcal{P}}_{\boldsymbol{T}}\boldsymbol{\mathcal{R}}_{\boldsymbol{\Omega}_t}\boldsymbol{\mathcal{P}}_{\boldsymbol{T}})\boldsymbol{\mathcal{D}}_{t-1}. \tag{94}$$

369 *Obviously $\boldsymbol{\mathcal{D}}_t \in \boldsymbol{T}$ for all $t > 0$. Note that $\boldsymbol{\Omega}_t$ is independent of $\boldsymbol{\mathcal{D}}_{t-1}$ and by the choice of $p$ in*
370 *Theorem 3 in the manuscript, we have*

$$q \geq \frac{p}{t_0} \geq \frac{c_0 \mu R \log(n)}{n}. \tag{95}$$

Applying Lemma 9 with $\boldsymbol{\Omega}$ replaced by $\boldsymbol{\Omega}_t$, we obtain that w.h.p.

$$\|\boldsymbol{\mathcal{D}}_t\|_F \leq \|\boldsymbol{\mathcal{P}_T} - \boldsymbol{\mathcal{P}_T}\boldsymbol{\mathcal{R}}_{\boldsymbol{\Omega}_t}\boldsymbol{\mathcal{P}_T}\|\|\boldsymbol{\mathcal{D}}_{t-1}\|_F \leq \frac{1}{2}\|\boldsymbol{\mathcal{D}}_{t-1}\|_F \tag{96}$$

for each $t$. Applying the above inequality recursively with $t = t_0, t_0 - 1, \cdots, 1$ gives

$$\|\boldsymbol{\mathcal{P}_T}\boldsymbol{\mathcal{Y}} - \boldsymbol{\mathcal{U}} *_L \boldsymbol{\mathcal{V}}^T\|_F = \|\boldsymbol{\mathcal{D}}_{t_0}\|_F \leq (\frac{1}{2})^{t_0}\|\boldsymbol{\mathcal{U}} *_L \boldsymbol{\mathcal{V}}^T\|_F \leq \frac{1}{4nn_3}\sqrt{R} \leq \frac{1}{4\sqrt{n}n_3} \leq \frac{1}{4}\sqrt{\frac{p}{r_3}}, \tag{97}$$

where the last inequality uses Eq. (92) and $r_3 \leq n_3$.

**Validating Condition 2(a).** Note that $\boldsymbol{\mathcal{W}} = \sum_{t=1}^{t_0} \boldsymbol{\mathcal{R}}_{\boldsymbol{\Omega}_t}\boldsymbol{\mathcal{P}_T}\boldsymbol{\mathcal{D}}_{t-1}$ by construction. We have

$$\|\boldsymbol{\mathcal{P}}_{\boldsymbol{\Omega}^\perp}\boldsymbol{\mathcal{W}}\|_F \leq \sum_{t=1}^{t_0} \|\boldsymbol{\mathcal{P}}_{\boldsymbol{T}^\perp}(\boldsymbol{\mathcal{R}}_{\boldsymbol{\Omega}_t}\boldsymbol{\mathcal{P}_T} - \boldsymbol{\mathcal{P}_T})\boldsymbol{\mathcal{D}}_{t-1}\| \leq \sum_{t=1}^{t_0} \|(\boldsymbol{\mathcal{R}}_{\boldsymbol{\Omega}_t} - \boldsymbol{\mathcal{I}})\boldsymbol{\mathcal{P}_T}\boldsymbol{\mathcal{D}}_{t-1}\|. \tag{98}$$

Applying Lemma 10 with $\boldsymbol{\Omega}$ replaced by $\boldsymbol{\Omega}_t$ to the above inequality, we get that w.h.p.

$$\begin{aligned}
\|\boldsymbol{\mathcal{P}}_{\boldsymbol{\Omega}^\perp}\boldsymbol{\mathcal{Y}}\|_F &\leq c \sum_{t=1}^{t_0} \left( \frac{\log(n)}{q}\|\boldsymbol{\mathcal{D}}_{t-1}\|_\infty + \sqrt{\frac{\log(n)}{q}}\|\boldsymbol{\mathcal{D}}_{t-1}\|_{\infty,2} \right) \\
&\leq \frac{c}{\sqrt{c_0}} \sum_{t=1}^{t_0} \left( \frac{n}{\mu R}\|\boldsymbol{\mathcal{D}}_{t-1}\|_\infty + \sqrt{\frac{n}{\mu R}}\|\boldsymbol{\mathcal{D}}_{t-1}\|_{\infty,2} \right)
\end{aligned} \tag{99}$$

where the last inequality uses Eq. (95). Now we bound $\|\boldsymbol{\mathcal{D}}_{t-1}\|_\infty$and $\|\boldsymbol{\mathcal{D}}_{t-1}\|_{\infty,2}$. Using Eq. (94) and repeatedly applying Lemma 4 with $\boldsymbol{\Omega}$ replaced as $\boldsymbol{\Omega}_t$, we obtain that w.h.p.

$$\|\boldsymbol{\mathcal{D}}_{t-1}\|_\infty = \|(\boldsymbol{\mathcal{P}_T} - \boldsymbol{\mathcal{P}_T}\boldsymbol{\mathcal{R}}_{\boldsymbol{\Omega}_{t-1}}\boldsymbol{\mathcal{P}_T})\cdots(\boldsymbol{\mathcal{P}_T} - \boldsymbol{\mathcal{P}_T}\boldsymbol{\mathcal{R}}_{\boldsymbol{\Omega}_1}\boldsymbol{\mathcal{P}_T})\boldsymbol{\mathcal{D}}_0\|_\infty \leq (\frac{1}{2})^{t-1}\|\boldsymbol{\mathcal{U}} *_L \boldsymbol{\mathcal{V}}^T\|_\infty. \tag{100}$$

By Lemma 11 with $\boldsymbol{\Omega}$ replaced by $\boldsymbol{\Omega}_t$, we obtain that w.h.p.

$$\|\boldsymbol{\mathcal{D}}_{t-1}\|_{\infty,2} = \|(\boldsymbol{\mathcal{P}_T} - \boldsymbol{\mathcal{P}_T}\boldsymbol{\mathcal{R}}_{\boldsymbol{\Omega}_{t-1}}\boldsymbol{\mathcal{P}_T})\boldsymbol{\mathcal{D}}_{t-2}\|_{\infty,2} \leq \frac{1}{2}\sqrt{\frac{n}{\mu R}}\|\boldsymbol{\mathcal{D}}_{t-2}\|_\infty + \frac{1}{2}\|\boldsymbol{\mathcal{D}}_{t-2}\|_{\infty,2}. \tag{101}$$

Using Eq. (94) and combining the last two display equations gives w.h.p.

$$\|\boldsymbol{\mathcal{D}}_{t-1}\|_{\infty,2} \leq t(\frac{1}{2})^{t-1}\sqrt{\frac{nn_3}{\mu R}}\|\boldsymbol{\mathcal{U}} *_L \boldsymbol{\mathcal{V}}^T\|_\infty + (\frac{1}{2})^{t-1}\|\boldsymbol{\mathcal{U}} *_L \boldsymbol{\mathcal{V}}^T\|_{\infty,2}. \tag{102}$$

Substituting back to Eq. (99), we get w.h.p.

$$\begin{aligned}
\|\boldsymbol{\mathcal{P}}_{\boldsymbol{\Omega}^\perp}\boldsymbol{\mathcal{Y}}\|_F &\leq \frac{c}{\sqrt{c_0}}\frac{n}{\mu R}\|\boldsymbol{\mathcal{U}} *_L \boldsymbol{\mathcal{V}}^T\|_\infty \sum_{t=1}^{t_0}(t+1)(\frac{1}{2})^{t+1} + \frac{c}{\sqrt{c_0}}\sqrt{\frac{n}{\mu R}}\|\boldsymbol{\mathcal{U}} *_L \boldsymbol{\mathcal{V}}^T\|_{\infty,2}\sum_{t=1}^{t_0}(\frac{1}{2})^{t+1} \\
&\leq \frac{6c}{\sqrt{c_0}}\frac{n}{\mu R}\|\boldsymbol{\mathcal{U}} *_L \boldsymbol{\mathcal{V}}^T\|_\infty + \frac{2c}{\sqrt{c_0}}\sqrt{\frac{n}{\mu R}}\|\boldsymbol{\mathcal{U}} *_L \boldsymbol{\mathcal{V}}^T\|_{\infty,2}.
\end{aligned} \tag{103}$$

Now we proceed to bound $\|\boldsymbol{\mathcal{U}} *_L \boldsymbol{\mathcal{V}}^T\|_\infty$ and $\|\boldsymbol{\mathcal{U}} *_L \boldsymbol{\mathcal{V}}^T\|_{\infty,2}$. First, by the definition of t-product, we have

$$\begin{aligned}
\|\boldsymbol{\mathcal{U}} *_L \boldsymbol{\mathcal{V}}^T\|_\infty &= \max_{i,j} \left\|\sum_{t=1}^r \boldsymbol{\mathcal{U}}(i,t,:) *_L \boldsymbol{\mathcal{V}}(j,t,:)\right\|_\infty \leq \max_{i,j} \sum_{t=1}^r \|\boldsymbol{\mathcal{U}}(i,t,:)\|_F\|\boldsymbol{\mathcal{V}}(j,t,:)\|_F \\
&\leq \max_{i,j} \sum_{t=1}^r \frac{1}{2}\left(\|\boldsymbol{\mathcal{U}}(i,t,:)\|_F^2 + \|\boldsymbol{\mathcal{V}}(j,t,:)\|_F^2\right) \\
&= \max_{i,j} \frac{1}{2}\left(\|\boldsymbol{\mathcal{U}}^T *_L \mathring{\mathfrak{e}}_i\|_F^2 + \|\boldsymbol{\mathcal{V}}^T *_L \mathring{\mathfrak{e}}_j\|_F^2\right) \leq \frac{\mu R}{n},
\end{aligned} \tag{104}$$

Also, we have

$$\|\boldsymbol{\mathcal{U}} *_L \boldsymbol{\mathcal{V}}^T\|_{\infty,2} \leq \max\left\{\max_i \|\mathring{\mathfrak{e}}_i^T *_L \boldsymbol{\mathcal{U}} *_L \boldsymbol{\mathcal{V}}\|_F, \max_i \|\boldsymbol{\mathcal{U}} *_L \boldsymbol{\mathcal{V}} *_L \mathring{\mathfrak{e}}_j\|_F\right\} \leq \sqrt{\frac{\mu R}{n}}. \tag{105}$$

It follows that w.h.p.

$$\|\boldsymbol{\mathcal{P}}_{\boldsymbol{T}^\perp}\boldsymbol{\mathcal{Y}}\| \leq \frac{6c}{\sqrt{c_0}} + \frac{2c}{\sqrt{c_0}} \leq \frac{1}{2}, \tag{106}$$

provided that $c_0$ is sufficiently large. This completes the proof of Theorem 3 in the manuscript. ∎

### 3.2 Proof of Some Lemmas

#### 3.2.1 Proof of Lemma 10

**Proof** *Denote the tensor $\mathcal{H}_{ijk} = (1 - \rho^{-1}\delta_{ijk})z_{ijk}\mathfrak{e}_{ijk}$. Then we have*

$$(\mathcal{I} - \mathcal{R}_{\boldsymbol{\Omega}})\boldsymbol{\mathcal{Z}} = \sum_{ijk} \mathcal{H}_{ijk}. \tag{107}$$

*Note that $\delta_{ijk}$'s are independent random scalars. Thus, $\mathcal{H}_{ijk}$'s are independent random tensors and $\overline{\mathbf{H}}_{ijk}$'s are independent random matrices. Observe that $\mathbb{E}[\overline{\mathbf{H}}_{ijk}] = \mathbf{0}$ and $\|\overline{\mathbf{H}}_{ijk}\| \leq \rho^{-1}\|\boldsymbol{\mathcal{Z}}\|_{\infty}$, we have*

$$\left\| \sum_{ijk} \mathbb{E}[\overline{\mathbf{H}}_{ijk}^{*}\overline{\mathbf{H}}_{ijk}] \right\| = \left\| \sum_{ijk} \mathbb{E}[\mathcal{H}_{ijk}^{*}\mathcal{H}_{ijk}] \right\| = \left\| \sum_{ijk} \mathbb{E}[(1 - \rho^{-1}\delta_{ijk})^2]z_{ijk}^2(\mathring{\mathfrak{e}}_j *_L \mathring{\mathfrak{e}}_j^{*}) \right\|$$
$$= \left\| \frac{1 - \rho}{\rho} \sum_{ijk} z_{ijk}^2(\mathring{\mathfrak{e}}_j *_L \mathring{\mathfrak{e}}_j^{*}) \right\| \leq \rho^{-1} \max_j \left| \sum_{i,k} z_{ijk}^2 \right| \leq \rho^{-1}\|\boldsymbol{\mathcal{Z}}\|_{\infty,2}^2. \tag{108}$$

*A similar calculation yields $\left\| \sum_{ijk} \mathbb{E}[\overline{\mathbf{H}}_{ijk}^{*}\overline{\mathbf{H}}_{ijk}] \right\| \leq \rho^{-1}\|\boldsymbol{\mathcal{Z}}\|_{\infty,2}^2$. Then the proof is completed by applying the matrix Bernstein inequality in Theorem 3.* ∎

#### 3.2.2 Proof of Lemma 11

**Proof** *For fixed $\boldsymbol{\mathcal{Z}} \in \boldsymbol{T}$ and fixed $b \in [n]$, the $b$-th column of the tensor $\mathcal{P}_{\boldsymbol{T}}\mathcal{R}_{\boldsymbol{\Omega}}(\boldsymbol{\mathcal{Z}}) - \boldsymbol{\mathcal{Z}}$ can be written as*

$$(\mathcal{P}_{\boldsymbol{T}}\mathcal{R}_{\boldsymbol{\Omega}}(\boldsymbol{\mathcal{Z}}) - \boldsymbol{\mathcal{Z}}) *_L \mathring{\mathfrak{e}}_b = \sum_{ijk} (\rho^{-1} - 1)\delta_{ijk}z_{ijk}\mathcal{P}_{\boldsymbol{T}}(\mathfrak{e}_{ijk}) *_L \mathring{\mathfrak{e}}_b := \sum_{ijk} \mathcal{H}_{ijk}, \tag{109}$$

*where $\mathcal{H}_{ijk}$'s are independent column tensor in $\mathbb{R}^{n \times 1 \times n_3}$ and $\mathbb{P}[\mathcal{H}_{ijk}] = \mathbf{0}$. Let $\boldsymbol{h}_{ijk} \in \mathbb{R}^{nn_3}$ be the column vector obtained by vectorizing $\mathcal{H}_{ijk}$. Then we have*

$$\|\boldsymbol{h}_{ijk}\| \leq \rho^{-1}|z_{ijk}|\|\mathcal{P}_{\boldsymbol{T}}(\mathfrak{e}_{ijk}) *_L \mathring{\mathfrak{e}}_b\|_F \leq \rho^{-1}\|\boldsymbol{\mathcal{Z}}\|_{\infty}\sqrt{\frac{2\mu R}{n}} \leq \frac{1}{c_0 \log(n)}\sqrt{\frac{2n}{\mu R}}\|\boldsymbol{\mathcal{Z}}\|_{\infty}. \tag{110}$$

*We also have*

$$\left| \sum_{ijk} \mathbb{E}[\boldsymbol{h}_{ijk}^{*}\boldsymbol{h}_{ijk}] \right| = \left| \sum_{ijk} \mathbb{E}[\|\mathcal{H}_{ijk}\|_F^2] \right| \leq \frac{1 - \rho}{\rho} \sum_{ijk} z_{ijk}^2\|\mathcal{P}_{\boldsymbol{T}}(\mathfrak{e}_{ijk}) *_L \mathring{\mathfrak{e}}_b\|_F^2. \tag{111}$$

*Note that*

$$\|\mathcal{P}_{\boldsymbol{T}}(\mathfrak{e}_{ijk}) *_L \mathring{\mathfrak{e}}_b\|_F^2$$
$$= \|\boldsymbol{\mathcal{U}} *_L \boldsymbol{\mathcal{U}}^T *_L \mathring{\mathfrak{e}}_i *_L \dot{\mathfrak{e}}_k *_L \mathring{\mathfrak{e}}_j^{*} *_L \mathring{\mathfrak{e}}_b - (\mathcal{I} - \boldsymbol{\mathcal{U}} *_L \boldsymbol{\mathcal{U}}^T) *_L \mathring{\mathfrak{e}}_i *_L \dot{\mathfrak{e}}_k *_L \mathring{\mathfrak{e}}_j^{*} *_L \boldsymbol{\mathcal{V}} *_L \boldsymbol{\mathcal{V}}^T *_L \mathring{\mathfrak{e}}_b\|_F$$
$$\leq \|\boldsymbol{\mathcal{U}} *_L \boldsymbol{\mathcal{U}}^T *_L \mathring{\mathfrak{e}}_i *_L \dot{\mathfrak{e}}_k\|_F\|\mathring{\mathfrak{e}}_j^{*} *_L \mathring{\mathfrak{e}}_b\|_F$$
$$\qquad + \|(\mathcal{I} - \boldsymbol{\mathcal{U}} *_L \boldsymbol{\mathcal{U}}^T) *_L \mathring{\mathfrak{e}}_i *_L \dot{\mathfrak{e}}_k\|\|\mathring{\mathfrak{e}}_j^{*} *_L \boldsymbol{\mathcal{V}} *_L \boldsymbol{\mathcal{V}}^T *_L \mathring{\mathfrak{e}}_b\|_F$$
$$\leq \sqrt{\frac{\mu R}{n}}\|\mathring{\mathfrak{e}}_j^{*} *_L \mathring{\mathfrak{e}}_b\|_F + \|\mathring{\mathfrak{e}}_j^{*} *_L \boldsymbol{\mathcal{V}} *_L \boldsymbol{\mathcal{V}}^T *_L \mathring{\mathfrak{e}}_b\|_F. \tag{112}$$

*It follows that*

$$\left| \sum_{ijk} \mathbb{E}[\boldsymbol{h}_{ijk}^* \boldsymbol{h}_{ijk}] \right| = \frac{2}{\rho} \sum_{ijk} z_{ijk}^2 \frac{\mu R}{n} \|\mathring{\mathfrak{e}}_j^* *_L \mathring{\mathfrak{e}}_b\|_F^2 + \frac{2}{\rho} \sum_{ijk} z_{ijk}^2 \|\mathring{\mathfrak{e}}_j^* *_L \boldsymbol{\mathcal{V}} *_L \boldsymbol{\mathcal{V}}^T *_L \mathring{\mathfrak{e}}_b\|_F^2$$

$$= \frac{2\mu R}{\rho n} \sum_{ijk} z_{ijk}^2 + \frac{2}{p} \sum_j \|\mathring{\mathfrak{e}}_j^* *_L \boldsymbol{\mathcal{V}} *_L \boldsymbol{\mathcal{V}}^T *_L \mathring{\mathfrak{e}}_b\|_F^2 \sum_{ik} z_{ijk}^2 \tag{113}$$

$$\leq \frac{2\mu R}{\rho n} \|\boldsymbol{\mathcal{Z}}\|_{\infty,2}^2 + \frac{2}{\rho} \|\boldsymbol{\mathcal{V}} *_L \boldsymbol{\mathcal{V}}^T *_L \mathring{\mathfrak{e}}_b\|_F^2 \|\boldsymbol{\mathcal{Z}}\|_{\infty,2}^2$$

$$\leq \frac{4\mu R}{\rho n} \|\boldsymbol{\mathcal{Z}}\|_{\infty,2}^2 \leq \frac{4}{c_0 \log(n)} \|\boldsymbol{\mathcal{Z}}\|_{\infty,2}^2.$$

*We can bound $\|\sum_{ijk} \mathbb{E}[\boldsymbol{h}_{ijk} \boldsymbol{h}_{ijk}^*]\|$ by the same quantity in a similar manner. Treating $\boldsymbol{h}_{ijk}$'s as $nn_3 \times 1$ matrices and applying the matrix Bernstein inequality in Theorem 3 gives that w.h.p.*

$$\|(\boldsymbol{\mathcal{P}_T \mathcal{R}_\Omega}(\boldsymbol{\mathcal{Z}}) - \boldsymbol{\mathcal{Z}}) *_L \mathring{\mathfrak{e}}_b\|_F = \left\| \sum_{ijk} \boldsymbol{\mathcal{H}}_{ijk} \right\| = \left\| \sum_{ijk} \boldsymbol{h}_{ijk} \right\|$$

$$\leq \frac{C}{c_0} \sqrt{\frac{2n}{\mu R}} \|\boldsymbol{\mathcal{Z}}\|_\infty + 4\sqrt{\frac{C}{c_0}} \|\boldsymbol{\mathcal{Z}}\|_{\infty,2} \tag{114}$$

$$\leq \frac{1}{2} \sqrt{\frac{2n}{\mu R}} \|\boldsymbol{\mathcal{Z}}\|_\infty + \frac{1}{2} \|\boldsymbol{\mathcal{Z}}\|_{\infty,2},$$

*provided that $c_0$ in the lemma statement is large enough. In a similar way, we prove that $\|\mathring{\mathfrak{e}}_b^* *_L (\boldsymbol{\mathcal{P}_T \mathcal{R}_\Omega}(\boldsymbol{\mathcal{Z}}) - \boldsymbol{\mathcal{Z}})\|_F$ is bounded by the same quantity w.h.p. The lemma follows from a union bound over all $(a,b) \in [n] \times [n]$.* ∎

# 4 Algorithm Details

## 4.1 Details of Algorithm 1 about ATNN-TRPCA model

We first write the augmented Lagrangian function of the ATNN-TRPCA model (15) as:

$$\min_{\overline{\boldsymbol{\mathcal{M}}}, \boldsymbol{\mathcal{E}}, \boldsymbol{\Lambda}, \mathbf{L}^T \mathbf{L} = \boldsymbol{I}} \|\overline{\boldsymbol{\mathcal{M}}}\|_* + \lambda \|\boldsymbol{\mathcal{E}}\|_1 + \frac{\mu}{2} \|\boldsymbol{\mathcal{Y}} - \overline{\boldsymbol{\mathcal{M}}} \times_3 \mathbf{L} - \boldsymbol{\mathcal{E}} + \boldsymbol{\Lambda}/\mu\|_F^2, \tag{115}$$

where $\mu$ is the penalty parameter and $\boldsymbol{\Lambda}$ is the lagrange multiplier.

**Update $\overline{\boldsymbol{\mathcal{M}}}$.** Fixing other variables except $\overline{\boldsymbol{\mathcal{M}}}$ in Eq. (115), we obtain the following sub-problem:

$$\arg\min_{\overline{\boldsymbol{\mathcal{M}}}} \|\overline{\boldsymbol{\mathcal{M}}}\|_* + \frac{\mu}{2} \|\overline{\boldsymbol{\mathcal{M}}} - (\boldsymbol{\mathcal{Y}} - \boldsymbol{\mathcal{E}} + \boldsymbol{\Lambda}/\mu) \times_3 \mathbf{L}^T\|_F^2. \tag{116}$$

Using the following equation (i.e., Eq. (11) in the manuscript):

$$\|\boldsymbol{\mathcal{A}}\|_* = \|\boldsymbol{\mathcal{S}}\|_1 = \|\overline{\boldsymbol{\mathcal{S}}}\|_* = \|\overline{\boldsymbol{A}}\|_* = \|\overline{\boldsymbol{\mathcal{A}}}\|_* \tag{117}$$

and the definition of **bdiag** (i.e., Eq. (6) in the manuscript):

$$\overline{\boldsymbol{A}} = \mathrm{bdiag}(\overline{\boldsymbol{\mathcal{A}}}) = \begin{bmatrix} \overline{\mathbf{A}}^{(1)} & & & \\ & \overline{\mathbf{A}}^{(2)} & & \\ & & \ddots & \\ & & & \overline{\mathbf{A}}^{(R)} \end{bmatrix}, \overline{\boldsymbol{\mathcal{A}}} = \mathrm{bfold}\left(\overline{\boldsymbol{A}}\right), \tag{118}$$

Eq. (116) can be rewritten as the following $r_3$ equations:

$$\arg\min_{\overline{\mathbf{M}}^{(i)}} \|\overline{\mathbf{M}}^{(i)}\|_* + \frac{\mu}{2} \|\overline{\mathbf{M}}^{(i)} - \overline{\mathbf{Y}}^{(i)}\|_F^2, \tag{119}$$

415    for each $i = 1, \cdots, r_3$, where $\overline{\mathbf{Y}} := \text{bdiag}\left((\boldsymbol{\mathcal{Y}} - \boldsymbol{\mathcal{E}} + \boldsymbol{\Lambda}/\mu) \times_3 \mathbf{L}^T\right)$.

416    Then the each $\overline{\mathbf{M}}^{(i)}, i = 1, \cdots, r_3$ can be updated by the soft-thresholding operator $\text{SVD}\tau(\cdot)$ [8]:

$$\overline{\mathbf{M}}^{(i)} = \mathbf{B}\mathcal{S}_{1/\mu}(\mathbf{C})\mathbf{D}^T, \text{where } \overline{\mathbf{M}}^{(i)} \stackrel{\text{svd}}{=} \mathbf{B}\mathbf{C}\mathbf{D}^T, \tag{120}$$

417    where $\mathcal{S}_\tau$ is the soft threshold operator $\mathcal{S}$ defined by [9].

418    **Update** $\mathbf{L}$. Fixing other variables except $\mathbf{L}$ in Eq. (115), we obtain the following sub-problem:

$$\underset{\mathbf{L}^T\mathbf{L}=\mathbf{I}}{\arg\min} \|\overline{\boldsymbol{\mathcal{M}}} - (\boldsymbol{\mathcal{Y}} - \boldsymbol{\mathcal{E}} + \boldsymbol{\Lambda}/\mu) \times_3 \mathbf{L}^T\|_F^2. \tag{121}$$

419    The Eq. (121) can be rewritten as:

$$\underset{\mathbf{L}^T\mathbf{L}=\mathbf{I}}{\arg\max} \left\langle \text{unfold}_3(\boldsymbol{\mathcal{Y}} - \boldsymbol{\mathcal{E}} + \boldsymbol{\Lambda}/\mu)^T \text{unfold}_3(\text{bfold}(\overline{\boldsymbol{\mathcal{M}}})), \mathbf{L} \right\rangle. \tag{122}$$

420    According to Theorem 1 in [10], we can get the solution of Eq. (122) as follows:

$$\begin{cases} [\mathbf{B}, \mathbf{C}, \mathbf{D}] = \text{svd}(\text{unfold}_3(\boldsymbol{\mathcal{Y}} - \boldsymbol{\mathcal{E}} + \boldsymbol{\Lambda}/\mu)^T \text{unfold}_3(\text{bfold}(\overline{\boldsymbol{\mathcal{M}}}))), \\ \mathbf{L} = \mathbf{B}\mathbf{D}^T. \end{cases} \tag{123}$$

421    **Update** $\mathbf{E}$. Extracting all items containing $\mathbf{E}$ in Eq. (115), we can get:

$$\underset{\boldsymbol{\mathcal{E}}}{\arg\min} \lambda\|\boldsymbol{\mathcal{E}}\|_* + \frac{\mu}{2}\|\boldsymbol{\mathcal{Y}} - \overline{\boldsymbol{\mathcal{M}}} \times_3 \mathbf{L} - \boldsymbol{\mathcal{E}} + \boldsymbol{\Lambda}/\mu\|_F^2. \tag{124}$$

422    By using the soft-thresholding operator, the solution of Eq. (125) is:

$$\boldsymbol{\mathcal{E}} = \mathcal{S}_{\lambda/\mu}(\boldsymbol{\mathcal{Y}} - \overline{\boldsymbol{\mathcal{M}}} \times_3 \mathbf{L} + \boldsymbol{\Lambda}/\mu) \tag{125}$$

423    **Update** multiplier $\boldsymbol{\Lambda}$. Based on the general ADMM principle, the multiplier is further updated by the
424    following equations:

$$\begin{cases} \boldsymbol{\Lambda} = \boldsymbol{\Lambda} + \mu(\boldsymbol{\mathcal{Y}} - \overline{\boldsymbol{\mathcal{M}}} \times_3 \mathbf{L} - \boldsymbol{\mathcal{E}}) \\ \mu = \mu\rho, \end{cases} \tag{126}$$

425    where $\rho$ is a constant value greater than 1.

## 4.2   Details of Algorithm 2 about ATNN-TC model

427    Introducing the auxiliary variable $\boldsymbol{\mathcal{E}}$, the TC model can be written as follows:

$$\underset{\overline{\boldsymbol{\mathcal{M}}}, \mathbf{L}}{\max} \|\overline{\boldsymbol{\mathcal{M}}}\|_*, \ s.t. \ \boldsymbol{\mathcal{Y}} = \overline{\boldsymbol{\mathcal{M}}} \times_3 \mathbf{L} + \boldsymbol{\mathcal{E}}, \mathcal{P}_\Omega(\boldsymbol{\mathcal{E}}) = 0, \mathbf{L}^T\mathbf{L} = \boldsymbol{I}. \tag{127}$$

428    The augmented Lagrangian function of the ATNN-TC model (127) can be written as:

$$\underset{\overline{\boldsymbol{\mathcal{M}}}, \boldsymbol{\mathcal{E}}, \boldsymbol{\Lambda}, \mathbf{L}^T\mathbf{L}=\boldsymbol{I}}{\min} \|\overline{\boldsymbol{\mathcal{M}}}\|_* + \frac{\mu}{2}\|\boldsymbol{\mathcal{Y}} - \overline{\boldsymbol{\mathcal{M}}} \times_3 \mathbf{L} - \boldsymbol{\mathcal{E}} + \boldsymbol{\Lambda}/\mu\|_F^2, s.t. \mathcal{P}_\Omega(\boldsymbol{\mathcal{E}}) = 0, \tag{128}$$

429    where $\mu$ is the penalty parameter and $\boldsymbol{\Lambda}$ is the lagrange multiplier.

430    The Eq. (128) can be divided into four sub-problems:

$$\begin{cases} \overline{\boldsymbol{\mathcal{M}}} := \underset{\overline{\boldsymbol{\mathcal{M}}}}{\min} \|\overline{\boldsymbol{\mathcal{M}}}\|_* + \frac{\mu}{2}\|\boldsymbol{\mathcal{Y}} - \overline{\boldsymbol{\mathcal{M}}} \times_3 \mathbf{L} - \boldsymbol{\mathcal{E}} + \boldsymbol{\Lambda}/\mu\|_F^2 \\[2mm] \mathbf{L} := \underset{\mathbf{L}^T\mathbf{L}=\boldsymbol{I}}{\min} \|\boldsymbol{\mathcal{Y}} - \overline{\boldsymbol{\mathcal{M}}} \times_3 \mathbf{L} - \boldsymbol{\mathcal{E}} + \boldsymbol{\Lambda}/\mu\|_F^2 \\[2mm] \boldsymbol{\mathcal{E}} := \underset{\mathcal{P}_\Omega(\boldsymbol{\mathcal{E}})=0}{\min} \|\boldsymbol{\mathcal{Y}} - \overline{\boldsymbol{\mathcal{M}}} \times_3 \mathbf{L} - \boldsymbol{\mathcal{E}} + \boldsymbol{\Lambda}/\mu\|_F^2 \\[2mm] \boldsymbol{\Lambda} := \boldsymbol{\Lambda} + \mu(\boldsymbol{\mathcal{Y}} - \overline{\boldsymbol{\mathcal{M}}} \times_3 \mathbf{L} - \boldsymbol{\mathcal{E}}) \\[2mm] \mu := \mu\rho, \end{cases} \tag{129}$$

The update processes of $\overline{\mathcal{M}}, \mathbf{L}, \mathbf{\Lambda}$ of Eq. (129) are given in Eq. (119), Eq. (123) and Eq. (126), respectively.

Regarding the update of $\mathcal{E}$, a closed-form solution can be obtained using the following equation (130).

$$\mathcal{E} = \mathcal{P}_{\mathbf{\Omega}}(\mathcal{Y} - \overline{\mathcal{M}} \times_3 \mathbf{L} + \mathbf{\Lambda}/\mu). \tag{130}$$

## 4.3 Convergence Analysis of the Algorithm 1 and Algorithm 2

Since Algorithms 1 and 2 solve a non-convex model, we cannot directly apply the theory of convex optimization [11] to provide a proof of their global convergence. Here, we can establish the convergence of these two algorithms by relying on the following two lemmas.

**Lemma 14** *The sequence of dual variable $\mathbf{\Lambda}$ in Algorithm 1 and 2 is bounded.*

**Proof** *According to the optimality principle, we have*

$$
\begin{aligned}
\mathbf{0} &\in \partial(\|\overline{\mathcal{M}}^{k+1}\|_*) - \mathbf{\Lambda}^k \times_3 \mathbf{L}^{k+1T} - \mu_k \left( \mathcal{Y} - \overline{\mathcal{M}}^{k+1} \times_3 \mathbf{L}^{k+1} - \mathcal{E}^{k+1} \right) \times_3 \mathbf{L}^{k+1T}, \\
\mathbf{0} &\in \partial(\lambda\|\mathcal{E}^{k+1}\|_1) - \mathbf{\Lambda}^k - \mu_k \left( \mathcal{Y} - \overline{\mathcal{M}}^{k+1} \times_3 \mathbf{L}^{k+1} - \mathcal{E}^{k+1} \right).
\end{aligned}
\tag{131}
$$

*Combining this with the update criterion of the $\mathbf{\Lambda}^k$ in Algorithm 1 and 2, we have*

$$
\begin{aligned}
\mathbf{\Lambda}^{k+1} \times_3 \mathbf{L}^{k+1T} &\in \partial(\|\overline{\mathcal{M}}^{k+1}\|_*), \\
\mathbf{\Lambda}^{k+1} &\in \partial(\|\mathcal{E}^{k+1}\|_1).
\end{aligned}
\tag{132}
$$

*Note the fact that the dual norm of $\|\cdot\|_*$ and $\|\cdot\|_1$ are $\|\cdot\|_2$ and $\|\cdot\|_\infty$, respectively, and $\|\cdot\|_2 = \lambda^{-1}\|\cdot\|_\infty$ by the definition in [8, 12]. Thus, using Theorem 4 in [12], we get that $\mathbf{\Lambda}^{k+1}$ are bounded.* ∎

**Lemma 15** *The accumulation point $(\overline{\mathcal{M}}^k, \mathbf{L}^k, \mathcal{E}^k)$ generated by Algorithm 1 and 2 is a feasible solution of ATNN model (15).*

**Proof** *Based on the general ADMM principle, we have*

$$\|\mathbf{\Lambda}^{k+1} - \mathbf{\Lambda}^k\|_F = \mu^k \|\mathcal{Y} - \overline{\mathcal{M}}^{k+1} \times_3 \mathbf{L}^{k+1} - \mathcal{E}^{k+1}\|_F \tag{133}$$

*Since $\{\mu^k\}$ is an increasing sequence and $\lim_{k \to +\infty} \mu^k = +\infty$, and according to Lemma 14, we have*

$$\lim_{k \to +\infty} \|\mathcal{Y} - \overline{\mathcal{M}}^{k+1} \times_3 \mathbf{L}^{k+1} - \mathcal{E}^{k+1}\|_F = 0 \tag{134}$$

*This completes the proof.* ∎

Next, we give the following convergence theorem about Algorithm 1 and 2 in the manuscript.

**Theorem 4** *The sequence $(\mathcal{X}^k = \overline{\mathcal{M}}^k \times_3 \mathbf{L}^k, \mathcal{E}^k)$ generated by Algorithm 1 and 2 converge to the optimal solution of model (15).*

**Proof** *Suppose $(\mathcal{X}^*, \mathcal{E}^*)$ are the optimal solution of Algorithm 1 and 2. Since $\mathcal{X}^*$ has many equivalent decomposition forms according to Theorem 2, the decomposition form $\mathcal{X}^* = \overline{\mathcal{M}}^*_{\mathbf{L}^k} \times_3 \mathbf{L}_k$ will not lose information of $\mathcal{X}^*$, where $\mathbf{L}_k$ is the solution of model (15) in the $k$-th iteration.*

*Based on Eq. (132) and the definition of subgradient, we have*

$$
\begin{aligned}
\|\overline{\mathcal{M}}^k\|_* + \lambda\|\mathcal{E}^k\|_1 &\le \|\overline{\mathcal{M}}^k_{\mathbf{L}^k}\|_* - \langle \mathbf{\Lambda}^k \times_3 \mathbf{L}^{kT}, \overline{\mathcal{M}}^*_{\mathbf{L}^k} - \overline{\mathcal{M}}^k \rangle + \lambda\|\mathcal{E}^*\|_1 - \langle \mathbf{\Lambda}^k, \mathcal{E}^* - \mathcal{E}^k \rangle \\
&= \|\overline{\mathcal{M}}^k_{\mathbf{L}^k}\|_* + \lambda\|\mathcal{E}^*\|_1 - \langle \mathbf{\Lambda}^k, \overline{\mathcal{M}}^*_{\mathbf{L}^k} \times_3 \mathbf{L}^k - \overline{\mathcal{M}}^k \times_3 \mathbf{L}^k \rangle - \langle \mathbf{\Lambda}^k, \mathcal{E}^* - \mathcal{E}^k \rangle \\
&= \|\overline{\mathcal{M}}^k_{\mathbf{L}^k}\|_* + \lambda\|\mathcal{E}^*\|_1 - \langle \mathbf{\Lambda}^k, \mathcal{Y} - \overline{\mathcal{M}}^k \times_3 \mathbf{L}^k - \mathcal{E}^k \rangle \\
&= \|\overline{\mathcal{M}}^k_{\mathbf{L}^k}\|_* + \lambda\|\mathcal{E}^*\|_1 + \langle \mathbf{\Lambda}^k, \overline{\mathcal{M}}^k \times_3 \mathbf{L}^k + \mathcal{E}^k - \mathcal{Y} \rangle
\end{aligned}
$$

*Combining the above equation with Lemma 15, we further have*

$$\lim_{k \to +\infty} \|\overline{\boldsymbol{\mathcal{M}}}^k\|_* + \lambda \|\boldsymbol{\mathcal{E}}^k\|_1 = \lim_{k \to +\infty} \|\overline{\boldsymbol{\mathcal{M}}}^*_{\mathbf{L}^k}\|_* + \lambda \|\boldsymbol{\mathcal{E}}^*\|_1. \tag{135}$$

*According to the optimality criterion, we have*

$$\|\overline{\boldsymbol{\mathcal{M}}}^*_{\mathbf{L}^k}\|_* + \lambda \|\boldsymbol{\mathcal{E}}^*\|_1 \leq \|\overline{\boldsymbol{\mathcal{M}}}^*_{\mathbf{L}^k}\|_* + \lambda \|\boldsymbol{\mathcal{E}}^k\|_1 \leq \|\overline{\boldsymbol{\mathcal{M}}}^k\|_* + \lambda \|\boldsymbol{\mathcal{E}}^k\|_1. \tag{136}$$

*Taking the limit of $k$ on both sides of Eq. (136), we can get*

$$\lim_{k \to +\infty} \|\overline{\boldsymbol{\mathcal{M}}}^k\|_* = \lim_{k \to +\infty} \|\overline{\boldsymbol{\mathcal{M}}}^*_{\mathbf{L}^k}\|_*,$$
$$\lim_{k \to +\infty} \|\boldsymbol{\mathcal{E}}^k\|_1 = \lim_{k \to +\infty} \|\boldsymbol{\mathcal{E}}^*\|_1. \tag{137}$$

*Based on the above equation, we can deduce that $\lim_{k \to +\infty} \boldsymbol{\mathcal{E}}^k = \lim_{k \to +\infty} \boldsymbol{\mathcal{E}}^*$. Moreover, as per Lemma 15, we know that $\overline{\boldsymbol{\mathcal{M}}}^k, \mathbf{L}^k, \boldsymbol{\mathcal{E}}^k$ are all feasible solutions of the model (15). Consequently, we can derive that*

$$\lim_{k \to +\infty} \overline{\boldsymbol{\mathcal{M}}}^k \times_3 \mathbf{L}^k = \lim_{k \to +\infty} \boldsymbol{\mathcal{Y}} - \boldsymbol{\mathcal{E}}^k = \boldsymbol{\mathcal{Y}} - \lim_{k \to +\infty} \boldsymbol{\mathcal{E}}^k = \boldsymbol{\mathcal{Y}} - \boldsymbol{\mathcal{E}}^* = \boldsymbol{\mathcal{X}}^*. \tag{138}$$

*This completes the proof.* ∎

## 5 More Experiments about ATNN-based Models

In the manuscript, due to page limitations, we only include the recovery results for the TRPCA task with a sparse noise variance of 0.6 and the recovery results for the TC task with an observation rate of 0.05. Here, we present more experimental results. The experimental numerical results of all compared methods for the TRPCA task under various sparse noises are provided in Tables 1 and 2. The experimental numerical results of all compared methods for the TC task under various sparse noises are provided in Tables 3, 4, 5 and 6.

From the results in Tables 1 and 2, it can be observed that despite our method only utilizing low-rank prior on spectral bands, it outperforms methods such as CTV and TCTV, which simultaneously incorporate spatial smoothness and spectral low-rankness priors. This demonstrates the effectiveness of the proposed ATNN norm. Additionally, our method exhibits significant advantages in terms of speed. In certain cases with sparse noise, the running time of our method is even lower than that of RPCA methods based on matrix nuclear norm. It should be noted that the running time of our method varies in different scenarios of sparse noise. This is because the chosen rank, i.e., $r_3$, is different for different sparse noise scenarios. In more complex scenarios where the proportion of valid information in the data is lower and the proportion of erroneous information is higher, assigning a high value to $r_3$ would not only fail to learn an effective COM but also increase the algorithm's running time. Therefore, for sparse noise with large variance, a smaller rank, i.e., $r_3$, should be chosen.

Compared to the TRPCA task, the tensor completion (TC) task has received more extensive attention since it has the higher practical value. As a result, many strong comparative methods have emerged, such as S2NTNN based on nonlinear transformations using neural networks, KBR model based on Tucker and CP joint decomposition, and the recently proposed TCTV that integrates both low-rankness and local smoothness properties. Nevertheless, from Tables 3 to (6), we can observe that the performance of the proposed ATNN is comparable to these three state-of-the-art tensor methods. Considering the recoverability theory and running time of our proposed model, the ATNN model demonstrates strong competitiveness.

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

Table 1: Quantitative comparison of all RPCA-based competing methods on **WDC** dataset under salt-and-pepper noise with various variance. The best and second results are highlighted in bold italics and underline.

| Variance | Metric | Observed | RPCA | SNN | KBR | TNN | CTNN | CTV | TCTV | Ours |
|---|---|---|---|---|---|---|---|---|---|---|
| 0.1 | MPSNR | 14.26 | 45.35 | 47.68 | 35.65 | 47.27 | 28.79 | 50.11 | 49.46 | **50.19** |
| | MSSIM | 0.2830 | 0.9980 | 0.9990 | 0.9720 | 0.9940 | 0.8820 | 0.9980 | 0.9950 | **0.9990** |
| | MFSIM | 0.7200 | 0.9980 | 0.9990 | 0.9820 | 0.9960 | 0.9260 | 0.9990 | 0.9970 | **0.9990** |
| | ERGAS | 836.82 | 28.99 | 22.41 | 71.06 | 34.20 | 149.31 | 16.35 | 26.38 | **15.16** |
| | MSAM | 43.10 | 1.49 | 1.04 | 4.52 | 2.50 | 7.82 | 1.23 | 2.08 | **1.02** |
| | Times | / | **27.18** | 291.36 | 219.16 | 808.74 | 339.07 | 122.16 | 607.60 | 43.39 |
| 0.2 | MPSNR | 11.24 | 43.99 | 46.27 | 34.98 | 44.99 | 27.17 | 48.70 | 47.78 | **49.05** |
| | MSSIM | 0.1370 | 0.9970 | 0.9980 | 0.9680 | 0.9920 | 0.8190 | 0.9980 | 0.9950 | **0.9980** |
| | MFSIM | 0.5850 | 0.9970 | 0.9980 | 0.9800 | 0.9950 | 0.8970 | 0.9990 | 0.9970 | **0.9990** |
| | ERGAS | 1184.15 | 33.34 | 25.98 | 76.09 | 38.53 | 179.40 | 18.51 | 28.87 | **17.25** |
| | MSAM | 48.61 | 1.65 | 1.25 | 4.84 | 2.87 | 11.65 | 1.34 | 2.28 | **1.10** |
| | Times | / | **32.05** | 460.46 | 352.29 | 1115.37 | 498.99 | 210.54 | 838.54 | 37.67 |
| 0.3 | MPSNR | 9.48 | 42.33 | 44.55 | 34.05 | 41.94 | 25.06 | 47.21 | 45.70 | **47.62** |
| | MSSIM | 0.0830 | 0.9960 | 0.9970 | 0.9620 | 0.9880 | 0.7030 | 0.9980 | 0.9930 | **0.9980** |
| | MFSIM | 0.5020 | 0.9960 | 0.9980 | 0.9760 | 0.9920 | 0.8560 | 0.9980 | 0.9960 | **0.9992** |
| | ERGAS | 1450.57 | 39.61 | 30.98 | 84.55 | 45.26 | 228.60 | 21.92 | 32.33 | **20.12** |
| | MSAM | 50.91 | 1.88 | 1.70 | 5.49 | 3.53 | 16.90 | **1.50** | 2.56 | **1.23** |
| | Times | / | 37.83 | 474.72 | 340.09 | 1096.57 | 498.69 | 210.93 | 839.23 | **32.36** |
| 0.4 | MPSNR | 8.23 | 39.98 | 42.20 | 32.63 | 37.17 | 22.39 | 45.35 | 43.09 | **45.73** |
| | MSSIM | 0.0540 | 0.9940 | 0.9940 | 0.9480 | 0.9710 | 0.5230 | 0.9940 | 0.9900 | **0.9970** |
| | MFSIM | 0.4470 | 0.9940 | 0.9960 | 0.9670 | 0.9840 | 0.7950 | 0.9950 | 0.9940 | **0.9980** |
| | ERGAS | 1675.42 | 49.52 | 39.29 | 98.13 | 62.72 | 312.13 | 29.41 | 37.60 | **24.61** |
| | MSAM | 52.03 | 2.21 | 2.61 | 6.43 | 5.59 | 24.02 | 1.78 | 3.04 | **1.48** |
| | Times | / | **26.52** | 482.14 | 253.40 | 1108.16 | 456.48 | 211.40 | 873.60 | 42.30 |
| 0.5 | MPSNR | 7.26 | 36.58 | 38.01 | 28.55 | 28.92 | 19.70 | 41.07 | 39.49 | **42.82** |
| | MSSIM | 0.0370 | 0.9840 | 0.9780 | 0.8710 | 0.8210 | 0.3430 | 0.9890 | 0.9820 | **0.9950** |
| | MFSIM | 0.4090 | 0.9880 | 0.9870 | 0.9230 | 0.9220 | 0.7240 | 0.9910 | 0.9900 | **0.9970** |
| | ERGAS | 1873.12 | 68.70 | 58.84 | 153.34 | 147.67 | 427.94 | 41.11 | 48.65 | **32.55** |
| | MSAM | 52.52 | 2.83 | 5.19 | 9.27 | 13.59 | 31.54 | 2.53 | 4.14 | **2.20** |
| | Times | / | 42.45 | 481.57 | 289.15 | 1017.68 | 454.77 | 212.01 | 876.65 | **32.83** |
| 0.6 | MPSNR | 6.47 | 32.09 | 26.02 | 22.65 | 19.62 | 17.21 | 33.85 | 31.95 | **39.82** |
| | MSSIM | 0.0260 | 0.9520 | 0.7170 | 0.6430 | 0.3720 | 0.2030 | 0.9450 | 0.9080 | **0.9910** |
| | MFSIM | 0.3820 | 0.9700 | 0.8770 | 0.8390 | 0.7290 | 0.6530 | 0.9670 | 0.9500 | **0.9940** |
| | ERGAS | 2052.29 | 112.91 | 211.96 | 303.49 | 432.62 | 570.94 | **87.23** | 103.47 | **45.81** |
| | MSAM | 52.68 | 4.43 | 21.52 | 12.82 | 29.85 | 37.78 | 7.70 | 9.43 | **2.47** |
| | Times | / | 29.00 | 736.19 | 167.21 | 419.22 | 485.74 | 170.22 | 815.04 | **21.34** |

[3] Emmanuel J Candès, Xiaodong Li, Yi Ma, and John Wright. Robust principal component analysis? *Journal of the ACM (JACM)*, 58(3):1–37, 2011.

[4] Zemin Zhang and Shuchin Aeron. Exact tensor completion using t-svd. *IEEE Transactions on Signal Processing*, 65(6):1511–1526, 2016.

[5] Roman Vershynin. Introduction to the non-asymptotic analysis of random matrices. *arXiv preprint arXiv:1011.3027*, 2010.

[6] Rudolf Ahlswede and Andreas Winter. Strong converse for identification via quantum channels. *IEEE Transactions on Information Theory*, 48(3):569–579, 2002.

[7] Joel A Tropp. User-friendly tail bounds for sums of random matrices. *Foundations of computational mathematics*, 12:389–434, 2012.

[8] Jianfeng Cai, Emmanuel J Candès, and Zuowei Shen. A singular value thresholding algorithm for matrix completion. *SIAM Journal on optimization*, 20(4):1956–1982, 2010.

[9] David L Donoho. De-noising by soft-thresholding. *IEEE transactions on information theory*, 41(3):613–627, 1995.

Table 2: Quantitative comparison of all RPCA-based competing methods on **PaviaU** dataset under salt-and-pepper noise with various variance. The best and second results are highlighted in bold italics and underline.

| Variance | Metric | Observed | RPCA | SNN | KBR | TNN | CTNN | CTV | TCTV | Ours |
|---|---|---|---|---|---|---|---|---|---|---|
| 0.1 | MPSNR | 14.52 | 46.51 | 43.33 | 33.69 | 46.83 | 27.42 | 47.80 | 47.89 | **50.20** |
|  | MSSIM | 0.2750 | 0.9970 | 0.9970 | 0.9670 | 0.9900 | 0.8640 | 0.9980 | 0.9930 | **0.9987** |
|  | MFSIM | 0.7230 | 0.9970 | 0.9970 | 0.9800 | 0.9930 | 0.9050 | 0.9980 | 0.9950 | **0.9987** |
|  | ERGAS | 699.00 | 34.96 | 35.94 | 81.40 | 44.20 | 159.33 | 16.07 | 34.02 | **15.38** |
|  | MSAM | 39.55 | 0.107 | 1.12 | 4.25 | 3.81 | 6.05 | 1.06 | 3.26 | **1.04** |
|  | Times | / | **5.01** | 124.39 | 59.76 | 71.26 | 123.12 | 39.55 | 140.24 | 38.57 |
| 0.2 | MPSNR | 11.51 | 43.21 | 41.14 | 32.77 | 44.48 | 25.73 | 46.54 | 46.46 | **49.03** |
|  | MSSIM | 0.1280 | 0.9940 | 0.9950 | 0.9610 | 0.9870 | 0.7720 | 0.9970 | 0.9910 | **0.9985** |
|  | MFSIM | 0.5780 | 0.9940 | 0.9950 | 0.9760 | 0.9910 | 0.8700 | 0.9988 | 0.9940 | **0.9991** |
|  | ERGAS | 989.25 | 62.63 | 45.73 | 90.50 | 49.20 | 191.17 | 18.71 | 36.61 | **17.15** |
|  | MSAM | 45.48 | 1.26 | 1.38 | 4.97 | 4.17 | 8.96 | 1.12 | 3.40 | **1.10** |
|  | Times | / | **5.76** | 183.52 | 71.26 | 80.19 | 126.71 | 30.18 | 352.36 | 34.28 |
| 0.3 | MPSNR | 9.75 | 39.10 | 38.69 | 30.91 | 41.67 | 23.48 | 44.77 | 44.20 | **47.34** |
|  | MSSIM | 0.0750 | 0.9890 | 0.9900 | 0.9430 | 0.9830 | 0.6020 | 0.9960 | 0.9900 | **0.9979** |
|  | MFSIM | 0.4910 | 0.9890 | 0.9920 | 0.9640 | 0.9890 | 0.8140 | 0.9980 | 0.9930 | **0.9987** |
|  | ERGAS | 1210.8 | 95.28 | 58.77 | 109.37 | 55.73 | 244.06 | 22.76 | 40.25 | **20.73** |
|  | MSAM | 47.77 | 1.64 | 1.70 | 5.88 | 4.70 | 13.66 | 1.45 | 3.67 | **1.29** |
|  | Times | / | **4.68** | 122.71 | 48.22 | 71.34 | 124.86 | 41.42 | 139.44 | 24.56 |
| 0.4 | MPSNR | 8.51 | 34.25 | 36.22 | 30.50 | 36.84 | 20.47 | 42.47 | 41.89 | **45.68** |
|  | MSSIM | 0.0480 | 0.9750 | 0.9830 | 0.9390 | 0.9660 | 0.3730 | 0.9940 | 0.9860 | **0.9971** |
|  | MFSIM | 0.4340 | 0.9790 | 0.9880 | 0.9580 | 0.9790 | 0.7230 | 0.9970 | 0.9910 | **0.9982** |
|  | ERGAS | 1398.3 | 130.85 | 75.09 | 112.64 | 69.88 | 341.05 | 29.31 | 45.29 | **24.49** |
|  | MSAM | 48.76 | 2.18 | 2.14 | 5.94 | 6.31 | 21.62 | 1.73 | 4.01 | **1.46** |
|  | Times | / | **5.41** | 181.80 | 69.68 | 81.77 | 122.99 | 39.48 | 155.01 | 24.38 |
| 0.5 | MPSNR | 7.53 | 29.55 | 33.84 | 24.34 | 26.62 | 17.70 | 39.22 | 38.35 | **42.68** |
|  | MSSIM | 0.0320 | 0.9330 | 0.9710 | 0.6920 | 0.7440 | 0.2100 | 0.9860 | 0.9770 | **0.9949** |
|  | MFSIM | 0.3940 | 0.9590 | 0.9810 | 0.8130 | 0.8820 | 0.6280 | 0.9930 | 0.9850 | **0.9968** |
|  | ERGAS | 1564.1 | 174.67 | 94.08 | 221.35 | 173.54 | 467.76 | 41.93 | 55.62 | **33.38** |
|  | MSAM | 49.08 | 3.23 | 2.79 | 8.35 | 15.95 | 30.13 | 2.24 | 4.93 | **1.96** |
|  | Times | / | **4.63** | 160.38 | 41.61 | 91.40 | 93.01 | 30.50 | 140.86 | 19.64 |
| 0.6 | MPSNR | 6.74 | 24.99 | 31.34 | 20.92 | 17.09 | 15.38 | 31.91 | 29.63 | **38.86** |
|  | MSSIM | 0.0220 | 0.8260 | 0.9490 | 0.4470 | 0.2340 | 0.1160 | 0.8870 | 0.8550 | **0.9847** |
|  | MFSIM | 0.3660 | 0.9170 | 0.9700 | 0.7080 | 0.6410 | 0.5480 | 0.9460 | 0.9190 | **0.9908** |
|  | ERGAS | 1713.1 | 243.65 | 117.44 | 328.18 | 513.59 | 610.53 | 94.59 | 123.22 | **52.78** |
|  | MSAM | 49.04 | 5.74 | 4.79 | 8.61 | 33.63 | 36.86 | 6.10 | 11.54 | **4.44** |
|  | Times | / | **6.59** | 121.03 | 58.63 | 120.21 | 130.77 | 41.85 | 172.51 | 19.53 |

[10] Jiangjun Peng, Qi Xie, Qian Zhao, Yao Wang, Leung Yee, and Deyu Meng. Enhanced 3dtv regularization and its applications on hsi denoising and compressed sensing. *IEEE Transactions on Image Processing*, 29:7889–7903, 2020.

[11] Stephen Boyd, Neal Parikh, Eric Chu, Borja Peleato, Jonathan Eckstein, et al. Distributed optimization and statistical learning via the alternating direction method of multipliers. *Foundations and Trends® in Machine learning*, 3(1):1–122, 2011.

[12] Zhouchen Lin, Minming Chen, and Yi Ma. The augmented lagrange multiplier method for exact recovery of corrupted low-rank matrices. *arXiv preprint arXiv:1009.5055*, 2010.

Table 3: Quantitative comparison of all competing methods on **Akiyo** dataset under difference sampling ratio (SR). The best and second results are highlighted in bold italics and underline, respectively.

| SR | Metric | LRMC | HaLRTC | KBR | TNN | CTNN | FTNN | OITNN | TCTV | S2NTNN | Ours |
|---|---|---|---|---|---|---|---|---|---|---|---|
| 0.05 | MPSNR | 10.80 | 17.66 | 29.77 | 31.95 | 28.64 | 22.74 | 32.69 | 33.42 | 33.16 | **33.73** |
|  | MSSIM | 0.2620 | 0.5300 | 0.9110 | 0.9340 | 0.8460 | 0.7090 | 0.9530 | 0.9530 | 0.9519 | **0.9566** |
|  | MFSIM | 0.6590 | 0.7480 | 0.9440 | 0.9620 | 0.9190 | 0.8440 | 0.9700 | 0.9690 | 0.9716 | **0.9778** |
|  | ERGAS | 706.08 | 322.44 | 79.83 | 63.42 | 91.09 | 190.88 | 58.52 | 53.17 | 55.06 | **52.01** |
|  | MSAM | 19.94 | 7.17 | 2.53 | 2.40 | 4.05 | 6.87 | 1.98 | 2.13 | 2.02 | **1.84** |
|  | Times | **8.06** | 61.04 | 696.93 | 217.47 | 188.90 | 1204.61 | 397.54 | 874.80 | 99.96 | 79.89 |
| 0.1 | MPSNR | 22.75 | 21.68 | **38.94** | 34.95 | 32.11 | 27.88 | 36.01 | 37.54 | 36.40 | 37.84 |
|  | MSSIM | 0.6760 | 0.6670 | **0.9870** | 0.9630 | 0.9200 | 0.8480 | 0.9760 | 0.9800 | 0.9757 | 0.9807 |
|  | MFSIM | 0.8520 | 0.8120 | **0.9910** | 0.9780 | 0.9550 | 0.9130 | 0.9840 | 0.9870 | 0.9848 | 0.9890 |
|  | ERGAS | 183.5 | 201.9 | **28.8** | 45.7 | 61.4 | 104.2 | 40.9 | 33.9 | 38.3 | 32.8 |
|  | MSAM | 5.35 | 5.22 | **0.95** | 1.76 | 2.85 | 4.18 | 1.42 | 1.32 | 1.45 | 1.13 |
|  | Times | **10.51** | 42.92 | 689.19 | 197.99 | 175.24 | 870.32 | 347.57 | 876.51 | 99.67 | 92.87 |
| 0.2 | MPSNR | 39.04 | 25.23 | **45.17** | 39.09 | 36.70 | 33.73 | 40.23 | 41.95 | 39.66 | 41.07 |
|  | MSSIM | 0.9860 | 0.7960 | **0.9960** | 0.9840 | 0.9690 | 0.9680 | 0.9890 | 0.9920 | 0.9877 | 0.9910 |
|  | MFSIM | 0.9920 | 0.8810 | **0.9970** | 0.9900 | 0.9820 | 0.9790 | 0.9930 | 0.9940 | 0.9919 | 0.9950 |
|  | ERGAS | 29.30 | 134.11 | **14.50** | 29.39 | 36.70 | 51.59 | 26.06 | 21.17 | 26.48 | 22.41 |
|  | MSAM | 1.00 | 3.94 | **0.52** | 1.16 | 1.46 | 1.71 | 0.94 | 0.82 | 1.03 | 0.78 |
|  | Times | **10.46** | 35.44 | 835.68 | 220.64 | 179.44 | 655.12 | 381.47 | 922.82 | 100.95 | 158.19 |
| 0.3 | MPSNR | 44.39 | 27.67 | **48.81** | 42.08 | 40.02 | 36.89 | 43.23 | 44.82 | 42.02 | 45.17 |
|  | MSSIM | 0.9950 | 0.8670 | **0.9980** | 0.9910 | 0.9850 | 0.9830 | 0.9940 | 0.9950 | 0.9924 | 0.9960 |
|  | MFSIM | 0.9970 | 0.9210 | **0.9990** | 0.9940 | 0.9910 | 0.9890 | 0.9960 | 0.9970 | 0.9949 | 0.9970 |
|  | ERGAS | 16.61 | 101.26 | **9.41** | 21.35 | 25.26 | 36.21 | 18.88 | 15.52 | 20.35 | 14.04 |
|  | MSAM | 0.58 | 3.21 | **0.37** | 0.86 | 1.17 | 0.98 | 0.70 | 0.61 | 0.79 | 0.55 |
|  | Times | **9.02** | 25.10 | 888.15 | 207.23 | 181.30 | 545.51 | 382.85 | 905.03 | 102.42 | 189.51 |

Table 4: Quantitative comparison of all competing methods on **Carphone** dataset under difference sampling ratio (SR). The best and second results are highlighted in bold italics and underline, respectively.

| SR | Metric | LRMC | HaLRTC | KBR | TNN | CTNN | FTNN | OITNN | TCTV | S2NTNN | Ours |
|---|---|---|---|---|---|---|---|---|---|---|---|
| 0.05 | MPSNR | 11.58 | 14.20 | 26.49 | 26.27 | 25.06 | 25.43 | 27.14 | **29.10** | 27.33 | 27.44 |
|  | MSSIM | 0.2710 | 0.3440 | 0.8160 | 0.7650 | 0.7260 | 0.7770 | 0.8340 | **0.8740** | 0.8090 | 0.8095 |
|  | MFSIM | 0.6470 | 0.6410 | 0.8920 | 0.8820 | 0.8590 | 0.8810 | 0.9060 | **0.9240** | 0.9025 | 0.9056 |
|  | ERGAS | 676.72 | 499.98 | 122.16 | 127.62 | 144.13 | 139.69 | 115.92 | **91.71** | 112.74 | 110.17 |
|  | MSAM | 22.09 | 13.58 | 5.69 | 6.96 | 7.69 | 7.83 | 5.90 | **5.06** | 6.01 | 5.87 |
|  | Times | **6.92** | 21.12 | 798.21 | 493.22 | 195.56 | 1135.70 | 472.97 | 1103.24 | 100.76 | 80.11 |
| 0.1 | MPSNR | 21.88 | 19.79 | **32.00** | 28.23 | 27.84 | 28.16 | 29.31 | 31.29 | 30.31 | 30.38 |
|  | MSSIM | 0.6230 | 0.5890 | **0.9260** | 0.8240 | 0.8160 | 0.8560 | 0.8800 | 0.9110 | 0.8843 | 0.8870 |
|  | MFSIM | 0.8160 | 0.7800 | **0.9550** | 0.9110 | 0.9050 | 0.9210 | 0.9310 | 0.9470 | 0.9371 | 0.9380 |
|  | ERGAS | 210.44 | 262.79 | **65.14** | 102.32 | 104.97 | 102.30 | 90.66 | 71.57 | 80.76 | 80.88 |
|  | MSAM | 9.36 | 9.89 | **3.30** | 5.76 | 5.89 | 5.83 | 4.79 | 4.03 | 4.31 | 4.34 |
|  | Times | **7.91** | 42.07 | 606.52 | 158.53 | 136.74 | 722.46 | 275.82 | 759.54 | 98.32 | 88.81 |
| 0.2 | MPSNR | 30.97 | 19.73 | **36.63** | 30.94 | 31.27 | 31.04 | 32.04 | 33.59 | 33.34 | 33.71 |
|  | MSSIM | 0.9080 | 0.5540 | **0.9660** | 0.8880 | 0.8970 | 0.9130 | 0.9230 | 0.9413 | 0.9321 | 0.9310 |
|  | MFSIM | 0.9520 | 0.7610 | **0.9800** | 0.9420 | 0.9460 | 0.9510 | 0.9560 | 0.9640 | 0.9617 | 0.9650 |
|  | ERGAS | 76.79 | 264.66 | **38.82** | 75.34 | 71.01 | 73.78 | 66.53 | 54.29 | 56.86 | 53.33 |
|  | MSAM | 3.98 | 10.08 | **2.17** | 4.37 | 4.15 | 4.24 | 3.65 | 3.08 | 3.15 | 3.06 |
|  | Times | **9.27** | 21.94 | 849.07 | 416.23 | 166.91 | 608.54 | 337.33 | 884.53 | 100.88 | 151.87 |
| 0.3 | MPSNR | 34.84 | 23.64 | **39.58** | 33.01 | 33.80 | 33.21 | 34.11 | 35.83 | 35.38 | 36.40 |
|  | MSSIM | 0.9550 | 0.7280 | **0.9800** | 0.9230 | 0.9360 | 0.9420 | 0.9470 | 0.9610 | 0.9531 | 0.9580 |
|  | MFSIM | 0.9750 | 0.8510 | **0.9890** | 0.9600 | 0.9660 | 0.9670 | 0.9700 | 0.9770 | 0.9733 | 0.9780 |
|  | ERGAS | 50.08 | 168.64 | **27.80** | 59.55 | 53.23 | 57.66 | 52.57 | 42.77 | 45.02 | 40.18 |
|  | MSAM | 2.61 | 7.11 | **1.62** | 3.51 | 3.17 | 3.33 | 2.95 | 2.49 | 2.56 | 2.33 |
|  | Times | **12.92** | 29.82 | 881.66 | 409.21 | 170.54 | 462.46 | 342.69 | 870.83 | 93.99 | 196.46 |

Table 5: Quantitative comparison of all competing methods on **WDC** dataset under difference sampling ratio (SR). The best and second results are highlighted in bold italics and underline, respectively.

| SR | Metric | LRMC | HaLRTC | KBR | TNN | CTNN | FTNN | OITNN | TCTV | S2NTNN | Ours |
|---|---|---|---|---|---|---|---|---|---|---|---|
| 0.01 | MPSNR | 14.70 | 18.12 | 20.27 | 22.86 | 22.91 | 21.42 | 25.75 | 26.30 | **27.49** | 26.62 |
| | MSSIM | 0.0480 | 0.2890 | 0.3590 | 0.5510 | 0.5130 | 0.5490 | 0.7350 | 0.7360 | **0.8165** | 0.7630 |
| | MFSIM | 0.4840 | 0.5410 | 0.6100 | 0.7850 | 0.7550 | 0.7780 | 0.8520 | 0.8520 | **0.9031** | 0.8830 |
| | ERGAS | 773.63 | 512.73 | 399.62 | 299.11 | 295.56 | 402.06 | 219.65 | 201.04 | **174.34** | 197.95 |
| | MSAM | 21.45 | 17.85 | 16.99 | 16.71 | 14.53 | 21.45 | 13.06 | 12.23 | **10.73** | 12.15 |
| | Times | **14.00** | 88.58 | 1349.8 | 470.75 | 472.85 | 2472.0 | 957.36 | 1720.7 | 162.30 | 113.20 |
| 0.05 | MPSNR | 18.54 | 22.01 | 31.42 | 30.06 | 33.36 | 34.70 | 32.29 | 33.33 | 37.30 | **38.06** |
| | MSSIM | 0.4620 | 0.6670 | 0.9020 | 0.8800 | 0.9430 | 0.9530 | 0.9270 | 0.9390 | 0.9749 | **0.9790** |
| | MFSIM | 0.7610 | 0.8350 | 0.9420 | 0.9350 | 0.9660 | 0.9710 | 0.9570 | 0.9640 | 0.9846 | **0.9870** |
| | ERGAS | 521.83 | 374.09 | 117.56 | 132.83 | 90.49 | 83.78 | 106.21 | 91.52 | 56.89 | **52.93** |
| | MSAM | 17.24 | 23.33 | 7.37 | 10.46 | 6.64 | 6.82 | 8.27 | 7.64 | 4.71 | **4.35** |
| | Times | **24.38** | 54.38 | 1589.7 | 1019.6 | 378.99 | 4376.2 | 838.68 | 2116.5 | 168.75 | 149.70 |
| 0.1 | MPSNR | 27.62 | 29.55 | **44.11** | 33.58 | 39.48 | 39.75 | 36.40 | 37.68 | 40.57 | 43.47 |
| | MSSIM | 0.8620 | 0.9110 | **0.9930** | 0.9400 | 0.9680 | 0.9810 | 0.9670 | 0.9740 | 0.9868 | 0.9920 |
| | MFSIM | 0.9300 | 0.9460 | 0.9960 | 0.9660 | 0.9870 | 0.9880 | 0.9800 | 0.9840 | 0.9918 | **0.9962** |
| | ERGAS | 215.13 | 176.03 | **28.86** | 90.43 | 43.49 | 51.46 | 68.77 | 57.41 | 40.11 | 29.80 |
| | MSAM | 11.86 | 8.78 | **2.64** | 7.95 | 3.72 | 4.88 | 6.05 | 5.34 | 3.57 | 2.74 |
| | Times | **20.87** | 54.75 | 1585.2 | 1000.7 | 358.34 | 3021.0 | 828.60 | 2088.7 | 203.58 | 245.32 |
| 0.2 | MPSNR | 46.38 | 25.21 | **50.52** | 38.23 | 47.35 | 45.45 | 41.29 | 42.58 | 43.73 | 48.91 |
| | MSSIM | 0.9950 | 0.7040 | **0.9980** | 0.9750 | 0.9970 | 0.9930 | 0.9860 | 0.9890 | 0.9931 | 0.9974 |
| | MFSIM | 0.9970 | 0.8330 | **0.9990** | 0.9850 | 0.9980 | 0.9950 | 0.9910 | 0.9930 | 0.9958 | 0.9984 |
| | ERGAS | 24.94 | 224.65 | **14.87** | 55.10 | 19.77 | 31.12 | 41.80 | 35.07 | 28.33 | 17.32 |
| | MSAM | 2.45 | 11.85 | **1.52** | 5.33 | 1.96 | 3.21 | 4.04 | 3.50 | 2.65 | 1.69 |
| | Times | **36.58** | 53.34 | 1893.5 | 462.46 | 383.57 | 3128.3 | 881.75 | 2203.5 | 163.28 | 280.35 |

Table 6: Quantitative comparison of all competing methods on **Cloth** dataset under difference sampling ratio (SR). The best and second results are highlighted in bold italics and underline, respectively.

| SR | Metric | LRMC | HaLRTC | KBR | TNN | CTNN | FTNN | OITNN | TCTV | S2NTNN | Ours |
|---|---|---|---|---|---|---|---|---|---|---|---|
| 0.01 | MPSNR | 11.82 | 16.46 | 17.47 | 18.03 | 18.16 | 16.16 | 19.27 | **22.68** | 20.43 | 18.71 |
| | MSSIM | 0.0280 | 0.3050 | 0.2790 | 0.2270 | 0.2750 | 0.2430 | 0.3360 | **0.5850** | 0.3895 | 0.3425 |
| | MFSIM | 0.4230 | 0.5090 | 0.5390 | 0.6320 | 0.6440 | 0.7010 | 0.6810 | **0.8340** | 0.8017 | 0.7046 |
| | ERGAS | 904.39 | 539.99 | 487.47 | 458.03 | 453.74 | 549.95 | 394.66 | **264.38** | 344.39 | 420.81 |
| | MSAM | 24.56 | 17.90 | 21.91 | 21.14 | 20.91 | 22.47 | 17.24 | **11.03** | 14.02 | 17.64 |
| | Times | **10.54** | 110.61 | 1223.6 | 412.56 | 124.18 | 1930.5 | 517.37 | 1430.4 | 82.32 | 28.04 |
| 0.05 | MPSNR | 13.10 | 19.00 | 24.14 | 23.46 | 25.70 | 25.26 | 24.01 | **28.39** | 27.44 | 25.81 |
| | MSSIM | 0.1900 | 0.3570 | 0.6420 | 0.6010 | 0.7360 | 0.7250 | 0.6510 | **0.8440** | 0.7589 | 0.7340 |
| | MFSIM | 0.6250 | 0.6100 | 0.8800 | 0.8670 | 0.9170 | 0.9110 | 0.8790 | **0.9540** | 0.9491 | 0.9270 |
| | ERGAS | 783.44 | 417.77 | 223.50 | 240.91 | 183.60 | 193.82 | 226.05 | **135.61** | 151.35 | 182.67 |
| | MSAM | 22.17 | 15.20 | 9.51 | 12.01 | 8.92 | 8.82 | 10.94 | **6.50** | 7.21 | 8.65 |
| | Times | **10.95** | 92.65 | 1292.4 | 441.03 | 136.16 | 2054.4 | 391.88 | 1488.6 | 86.35 | 121.61 |
| 0.1 | MPSNR | 15.96 | 20.62 | 31.87 | 26.71 | 29.23 | 29.49 | 27.45 | 31.78 | **32.21** | 29.65 |
| | MSSIM | 0.3970 | 0.4540 | 0.9080 | 0.7640 | 0.8720 | 0.8680 | 0.8020 | **0.9140** | 0.9061 | 0.8510 |
| | MFSIM | 0.7950 | 0.7080 | 0.9800 | 0.9340 | 0.9640 | 0.9600 | 0.9420 | 0.9780 | **0.9802** | 0.9660 |
| | ERGAS | 574.58 | 346.60 | 90.79 | 164.37 | 119.83 | 118.33 | 150.88 | 92.23 | **89.16** | 116.70 |
| | MSAM | 18.93 | 12.61 | **4.52** | 8.90 | 5.84 | 5.97 | 8.02 | 4.81 | 4.77 | 6.18 |
| | Times | **10.61** | 73.22 | 1285.5 | 437.32 | 131.46 | 1553.0 | 397.20 | 1459.8 | 88.22 | 182.20 |
| 0.2 | MPSNR | 24.78 | 23.25 | **38.54** | 31.09 | 34.28 | 34.38 | 31.87 | 35.87 | 37.75 | 35.21 |
| | MSSIM | 0.7310 | 0.6320 | **0.9730** | 0.8890 | 0.9420 | 0.9440 | 0.9070 | 0.9580 | 0.9672 | 0.9440 |
| | MFSIM | 0.9460 | 0.8400 | **0.9950** | 0.9740 | 0.9810 | 0.9850 | 0.9780 | 0.9910 | 0.9938 | 0.9893 |
| | ERGAS | 233.65 | 255.15 | **44.78** | 99.67 | 68.33 | 69.51 | 91.39 | 58.83 | 49.15 | 66.54 |
| | MSAM | 9.24 | 9.55 | **2.66** | 5.89 | 3.83 | 3.90 | 5.35 | 3.39 | 2.90 | 3.80 |
| | Times | **9.85** | 49.62 | 1221.9 | 356.16 | 117.28 | 1052.4 | 412.30 | 1402.2 | 88.44 | 226.42 |