# OpenReview forum: "Efficient and Learnable Transformed Tensor Nuclear Norm with Exact Recoverable Theory"
_NeurIPS.cc/2023/Conference — NeurIPS 2023 Conference Withdrawn Submission_

### Official Review · Reviewer_kX6Y · 2023-06-09

**Soundness:** 3 good
**Presentation:** 3 good
**Contribution:** 3 good
**Rating:** 6
**Confidence:** 3

**Summary:**

This paper addresses the challenges associated with the tensor nuclear norm, which is a proxy for the low-rank property of tensors. Existing transformations for the tensor nuclear norm lack adaptability and theoretical guarantees due to fixed or non-invertible nonlinear approaches. This paper introduces a practical and data-adaptive framework and provides an exact recoverable theoretical guarantee. Experiments are included to support the theoretical guarantees.

**Strengths:**

The paper contains solid theoretical results for a very challenging problem. It provides efficient algorithms for recovery that are explained well and clearly. The experiments include both real and synthetic data.

**Weaknesses:**

The paper is dense to read in some places, like in the slurry of definitions at the end of Section 2. This is likely due to space limitations, but reduces the readability of the paper. The main results include some pretty strict assumptions like incoherence and uniformly distributed support sets. Although some of these are also used in other results, it could be discussed further what kinds of tensors have these properties in practice and how sensitive the methods are to these.

**Questions:**

It might be good to elaborate in lines 46-52 about the importance of data-dependency, why it is better than non-adaptive approaches.

In that same section, in (3), it would help to specify what "good" in this context means, i.e. what metrics are important in performance

In Remark 1, the phrase "the higher the solution efficiency of ATNN can be obtained" is not clear, perhaps this can be re-phrased

Definition 7 reads awkwardly -- is the "skinny t-SVD" an assumption or part of the definition?

**Limitations:**

The authors discuss interesting future work.

---

> ### Author Rebuttal · Authors · 2023-08-05
>
> We greatly appreciate your positive evaluation of our work. We sincerely thank you for your valuable feedback, which is incredibly important to us, and we truly appreciate the time and effort you've taken to review our work. As for your concerns, we give a detailed response which we hope can help you fully understand our work. Please feel free to let us know if you have any further concerns. We will deeply appreciate that you can raise your score if you find our responses resolve your concerns.
>
> **Q1: The paper is dense to read in some places, like in the slurry of definitions at the end of Section 2. This is likely due to space limitations, but reduces the readability of the paper. The main results include some pretty strict assumptions like incoherence and uniformly distributed support sets. Although some of these are also used in other results, it could be discussed further what kinds of tensors have these properties in practice and how sensitive the methods are to these.**
>
> **A1**: Indeed, this is a classic question. Many studies adopt assumptions without exploring their origins. The "incoherence condition" arose from Candes and Tao's observation that specific low-rank matrices deviate from recovery theory. For instance, a rank-1 matrix $M = e_1 * e_n$ with $e_i$ as a unit vector having only the $i$-th entry as 1 is challenging to recover unless most elements are observed. The "incoherence condition" measures differences between matrix singular value vectors and standard unit vectors. Larger values indicate greater deviations, leading to dispersed singular value vector elements and randomized non-zero element positions in matrix $M$. This enables accurate low-rank recovery from finite observations, ensuring global information inference. Generally, if low-rank matrix elements are randomly distributed, they satisfy this condition. Candes' 2012 paper "Exact matrix completion via convex optimization" rigorously proved this.
>
> Sparse matrix assumptions prevent non-zero elements from clustering, avoiding confusion between low-rank and sparse matrices. Robust PCA addresses outlier noise, and the outlier noise term satisfies uniformly distributed support sets.
>
> **Q2: It might be good to elaborate in lines 46-52 about the importance of data-dependency, and why it is better than non-adaptive approaches.**
>
> **A2**: Thank you for your suggestion. The appropriate transformation matrix should capture certain characteristics of the data. Since each data instance has distinct attributes, it is ideal to generate a suitable transformation matrix for each data. We will incorporate an explanation of data dependency in the upcoming revised version. Furthermore, in the rebuttal PDF we provided, it can be further observed that TNN methods based on learning transformation matrices significantly outperform those TNN methods that rely on fixed transformation matrices.
>
> **Q3: In that same section, in (3), it would help to specify what "good" in this context means, i.e. what metrics are important in performance.**
>
> **A3**: Thank you for your suggestion. We will include the following explanation in the upcoming versions. A "good" transformation matrix should be capable of effectively restoring data from the given degraded observations, bringing the repaired data closer to the ground truth. This effectiveness is reflected in metrics such as PSNR, SSIM, and AUC, which can quantitatively assess the restoration performance.
>
> **Q4: In Remark 1, the phrase "the higher the solution efficiency of ATNN can be obtained" is not clear, perhaps this can be re-phrased.**
>
> **A4**: Thank you for your suggestion. Due to the definition of ATNN in Equation (5), only the singular value decomposition of $r (r < n_3) $ matrices needs to be computed. This is because the tensor transformed by the column orthogonal transformation matrix in ATNN has only $r$ slices. On the other hand, for the fixed transformation matrix being the discrete Fourier matrix in Equation (2), since the Fourier matrix is full-rank, the computation of the matrix singular value decomposition still requires n_3 slices. As a result, Equation (5) can achieve higher computational efficiency. We will incorporate the explanation above into the upcoming revised version.
>
> **Q5: Definition 7 reads awkwardly -- is the "skinny t-SVD" an assumption or part of the definition?**
>
> **A5**: Here,  skinny t-SVD refers to the process of computing only the most important singular values and singular vectors during the singular value decomposition, effectively trimming down the computation by focusing on the crucial components and thereby enhancing efficiency. Taking a matrix as an example, for a matrix $M$ of size $m×n$ with rank $r$, the full SVD can be represented as $M = USV^T$, where $U$ is an $m×m$ orthogonal matrix, $S$ is an $m×n$ diagonal matrix, and $V^T$ is an $n×n$ orthogonal matrix. In contrast, in the context of "skinny SVD," $U$ becomes an $m×r$ orthogonal matrix, $S$ becomes an $r×r$ diagonal matrix, and $V^T$ is an $r×n$ orthogonal matrix.

---

> > ### Comment · Reviewer_kX6Y · 2023-08-11
> >
> > Thank you for the responses. It would be good if these clarifications could be incorporated into the text if the paper is accepted.

---

> > > ### Author Response · Authors · 2023-08-12
> > > **Response to Reviewer kX6Y**
> > >
> > > Thank you for the comments. If the paper is accepted, we will incorporate these clarifications into the text.

---

### Official Review · Reviewer_1VQ8 · 2023-07-04

**Soundness:** 2 fair
**Presentation:** 2 fair
**Contribution:** 2 fair
**Rating:** 3
**Confidence:** 5

**Summary:**

This paper presents a new approach for learning a data-adaptive and learnable column-orthogonal matrix (COM) transform for tensor nuclear norm (TNN) minimization. The authors show that the proposed transform can capture the low-rank structure of tensors more effectively than existing fixed or nonlinear transforms. They also provide theoretical guarantees for the exact recovery of tensors under the COM transform. The proposed method is applied to tensor completion and tensor robust principal component analysis tasks and achieves better performance and efficiency than several state-of-the-art methods.

**Strengths:**

1. The authors build up an adaptive tensor nuclear norm, and develop tensor completion robust tenor PCA model.
2. To guarantee the theoretical correctness of the proposed models, the authors analyze two theorems to ensure the exact recovery capability.

**Weaknesses:**

1. The proof architecture is regular and simple and has been used in [1,2], thus the theoretical novelty of this paper is rather limited.
2. The adaptive TNN model has been already proposed in many related works, but the authors missed some important related references, see the linear invertible TNN-based RPCA [1,2], the nonlinear transform TNN [3], and the subspace denoising strategy NGmeet [4]. Thus, a related works section should be considered to give detailed differences between these methods and the proposed ATNN.
3. In the experimental section, the authors only compare the computation-cost methods and neglect the related adaptive TNN methods.

[1] Lu C. Transforms based tensor robust PCA: Corrupted low-rank tensors recovery via convex optimization[C]//Proceedings of the IEEE/CVF International Conference on Computer Vision. 2021: 1145-1152.

[2] Lu C, Peng X, Wei Y. Low-rank tensor completion with a new tensor nuclear norm induced by invertible linear transforms[C]//Proceedings of the IEEE/CVF conference on computer vision and pattern recognition. 2019: 5996-6004.

[3] Yisi Luo, Xile Zhao, Deyu Meng, and Taixiang Jiang, ‘‘HLRTF: Hierarchical Low-Rank Tensor Factorization for Inverse Problems in Multi-Dimensional Imaging,’’ IEEE/CVF Conference on Computer Vision and Pattern Recognition (CVPR), 2022.

[4] He W, Yao Q, Li C, et al. Non-local meets global: An integrated paradigm for hyperspectral denoising[C]//Proceedings of the IEEE/CVF Conference on Computer Vision and Pattern Recognition. 2019: 6868-6877.

**Questions:**

1. In Table I, why not compare the computation complexity of different methods? The speed comparison is unfair by the subjective evaluation
2. What are the differences between the proposed adaptive TNN and SALTS?
3. Please give more details about the differences between the proposed method and TRPCA-COM [1-2], including the model, and theoretical improvements.
4. The authors state the superior computational efficiency of the proposed method compared with the other adaptive TNN, why not compare these methods in the experimental section? e.g., Q-rank, SALTS, etc.

[1] Lu C. Transforms based tensor robust PCA: Corrupted low-rank tensors recovery via convex optimization[C]//Proceedings of the IEEE/CVF International Conference on Computer Vision. 2021: 1145-1152.
[2] Lu C, Peng X, Wei Y. Low-rank tensor completion with a new tensor nuclear norm induced by invertible linear transforms[C]//Proceedings of the IEEE/CVF conference on computer vision and pattern recognition. 2019: 5996-6004.

**Limitations:**

Please see the above Weakness section.

---

> ### Author Rebuttal · Authors · 2023-08-05
>
> We sincerely thank you for your valuable feedback, which is incredibly important to us, and we truly appreciate the time and effort you've taken to review our work. As for your concerns, we give a detailed response which we hope can help you fully understand our work. Please feel free to let us know if you have any further concerns. We will deeply appreciate that you can raise your score if you find our responses resolve your concerns.
>
> **Q1: The proof architecture is regular and simple and has been used in [1,2], thus the theoretical novelty of this paper is rather limited.**
>
> **A1**:  The application of the ATNN norm to the TRPCA and TC models for exact recoverability theory follows a well-established proof framework that is commonly used in tensor nuclear norm(TNN) research. In fact, nearly all theoretical works in the TNN-class domain have employed a proof framework similar to that of this paper. It involves constructing dual variables using the Golf Scheme criterion and verifying that these dual variables satisfy the Karush-Kuhn-Tucker (KKT) conditions of the model, establishing alignment between the optimal solution and the original problem's solution.
>
> For instance, previous works like TNN (Zemin Zhang et al., TSP2016) and TRPCA (Canyi Lu et al., TPAMI2019) have employed the same proof framework in [1,2]. Therefore, in this field, introducing novel theoretical tools can be challenging but also highly significant. Without new theoretical tools, attempting to expand the applicability of TNN or relax assumptions is also an exciting endeavor. ATNN builds upon [1,2] and improves by relaxing the requirement for the transformation matrix to be invertible, instead only necessitating column full rank.
>
> **Q2: The adaptive TNN model has been explored in related works, including linear invertible TNN-based RPCA [1,2], nonlinear transform TNN [3], and NGmeet [4]. Please provided a related works section to outline the distinctions between these methods and ATNN.**
>
> **A2**: We have indeed referenced the journal versions of both [2] and [3] (S2NTNN). As for [1] and [4], we will provide a detailed discussion in our forthcoming versions. Unlike the transformation matrices in [1,2], which are predetermined, ATNN learns the transformation matrix adaptively. In comparison to [3] (S2NTNN) and [4] (NGmeet), ATNN comes with a theoretical recovery guarantee. Additionally, although [4] leverages the low-rank nature of data for acceleration, it does not utilize nuclear norm regularization.
>
> **Q3: The authors only compare the computation-cost methods and neglect the related adaptive TNN methods.**
>
> **A3**: Due to the sluggish runtime of SALTS and its omission from the category of TNN-class methods, coupled with the lack of access to Qrank's code during submission, we did not include a comparison in the experimental section. In the rebuttal PDF, we have included a comparison with Qrank and SALTS. Additionally, we have presented an extended version of the ATNN model, namely the **TCTV*** model, which combines ATNN with TCTV. The following table shows the average restoration results of these methods.
>
> |Indexs|TNN|ATNN|Qrank|SALTS|TCTV|TCTV*|
> |---------|----------|----------|----------|----------|----------|----------|
> |PSNR|26.97|30.26|29.54|32.68|30.64|33.39|
> |Time(s)|404.3|115.3|452.3|3360|1200|146.4|
>
> As seen in the table, ATNN outperforms Qrank with a shorter runtime, thanks to its transformation matrix based on column orthogonal matrices (COM). COM optimally exploits the data's low-rank structure, enhancing computational efficiency. ATNN slightly lags behind SALTS, which employs the Schatten p-norm for transformed tensors, leading to longer processing times and lacking recovery guarantees. Notably, SALTS' runtime is almost 30 times longer than ATNN's. When comparing TCTV and TCTV*, along with TNN and ATNN, it's evident that learning COM from data is effective. Additionally, TCTV* significantly surpasses SALTS with a shorter runtime, underscoring our method's potential showcased in this paper.
>
> **Q4: In Table I, why not compare the computation complexity of different methods? The speed comparison is unfair by the subjective evaluation.**
>
> **A4**: For a given tensor $\mathcal{X}\in \mathbb{R}^{n_1\times n_2\times n_3}$, the sizes of the transformed tensors of all other methods are not smaller than $n_1\times n_2\times n_3$. Only our method results in a small tensor size of $n_1\times n_2\times r_3(r_3\leq n_3)$. Therefore, our method possesses the lowest time complexity, specifically $\mathcal{O}(n_1n_2^2{\color{red}r_3})$. More analysis is provided in the rebuttal PDF file.
>
> **Q5: What are the differences between the proposed adaptive TNN and SALTS?**
>
> **A5**: SALTS and ATNN both learn transformation matrices from data, but differ in regularization. SALTS employs the non-convex Schatten p-norm instead of the tensor nuclear norm. The Schatten p-norm might suit low-rank data better, but its complexity makes solving intricate and time-consuming. Unlike the nuclear norm, it lacks strong theoretical properties. Hence, SALTS is slower and lacks a recovery guarantee like ATNN.
>
> **Q6: Please give more details about the differences between the proposed method and TRPCA-COM [1-2], including the model, and theoretical improvements.**
>
> **A6**: In our model, the integration of the transformation matrix into the optimization framework facilitates customized learning tailored to the specific data. In contrast, TRPCA-COM [1-2] employs a predefined transformation matrix. If this matrix is not well-suited, the results might even lag behind those of the TNN method, as observed in UTNN results in Tables 1-5. Theoretical refinements relax the necessity of matrix invertibility in TRPCA-COM [1-2], indicating that under certain conditions, full column rank is adequate.
>
> **Q7: Provide comparative experiments with Q-rank and SALTS.**
>
> **A7**: We compared SALTS and Qrank, and their numerical results can be found in A3.

---

> > ### Author Response · Authors · 2023-08-16
> >
> > Thank you for your contribution to reviewing this article earlier. We have addressed your previous questions point by point. We are unsure whether your doubts have been cleared, so we kindly ask you to take some time again to review our responses. Your feedback is highly important to us. We will deeply appreciate that you can raise your score if you find our responses resolve your concerns.

---

> > > ### Author Response · Authors · 2023-08-18
> > >
> > > The deadline for the discussion is approaching. Please consider giving the paper another opportunity. If the rebuttal does not address your concerns, we are prepared to provide a more comprehensive response if necessary. If you find that our responses have effectively addressed your concerns, can you give us feedback and raise your score?  Your feedback is crucial to us. Thanks a lot.

---

> > > ### Comment · Reviewer_1VQ8 · 2023-08-18
> > > **The authors have not addressed the main concerns.**
> > >
> > > In this rebuttal, it appears that the authors have not adequately addressed my primary concern, which pertains to the novelty and contributions presented in this paper.
> > >
> > > To commence, the authors concede that the proof's structure adheres closely to the principal proof outline of TNN and the matrix case. Consequently, I concur that following this structure to derive the theoretical outcomes in this paper is not particularly intricate.
> > >
> > > Following this, the authors should verify whether any mistakes have arisen concerning the computational efficiency of SALTS. In the Computational Complexity Section of the original SALTS paper, the authors explicitly stated that the computational complexity of SALTS stands at O(kn^3), with the hyperparameter k playing an analogous role in this paper. Consequently, the computational cost for both SALTS and the present paper should ostensibly align. What arouses my curiosity is the differing runtime of SALTS and this paper. Why, despite sharing the same computational complexity, do these two methodologies necessitate significantly distinct runtimes (with the latter being 30 times slower)?
> > >
> > > Furthermore, the proposed model exhibits negligible divergence from SALTS, aside from the utilization of the Schatten-p norm. Based on prior investigations, the Schatten-p norm is expected to outperform the proposed model. I urge the authors to provide additional elucidation regarding the experimental findings in this regard.
> > >
> > > Broadly, the proposed model does yield certain theoretical results concerning the proposed ATNN. Nevertheless, given the existence of a comparable model in SALTS and the development of analogous results in the comprehensible TNN framework, I will keep my original score.

---

> > > > ### Author Response · Authors · 2023-08-18
> > > >
> > > > Thanks for your feedback. From a theoretical perspective, we have indeed relaxed certain conditions, which is a contribution in itself compared to the TNN series methods. Moreover, this is especially noteworthy when compared to models like SALTS which lack theoretical justifications. As for the computational complexity of SALTS, it's important to note that the original paper did not analyze the time complexity of solving the Schatten-p norm. Thus, even when SALTS is configured to learned transformation matrices with a column size of k, its time complexity is not necessarily O(kn^3). Furthermore, in the released SALTS code, it is recommended to choose k as n_3. In our experiments, we also found that under the Schatten-p norm-setting of SALTS, the best performance is indeed achieved when k is set to n_3.  **Looking at the runtime reported in the original SALTS paper, it is evident that SALTS exhibits the slowest runtime among all comparative methods.**
> > > >
> > > > If the choice of k in SALTS is similar to that of our ATNN and yields poor results, we have previously conducted experiments with such configurations as well. Due to the limitations of our previous response, we did not include this information. Here are the experimental results for this particular setting.
> > > >
> > > > Method | TNN | ATNN  | SALTS^*  | SALTS
> > > > ---- | ---| ---| ---| ---
> > > > PSNR | 26.97| 30.26 | 25.58 | 32.68
> > > > TIme |  404.3|115.3 | 868.3 | 3360
> > > >
> > > > In the table provided, "SALTS^*" represents the results obtained by selecting the same value of k (i.e., the k used in ATNN) for SALTS. As evident from the table, the performance of SALTS in this scenario is even worse compared to TNN. We speculate that the reason behind the above table is that the Schatten-p norm excels in handling relatively small singular values, which is why SALTS performs optimally when k is set to n_3. However, because of the adoption of the Schatten-p norm, SALTS sacrifices recoverability and faces an additional increase in runtime.

---

> > > > > ### Author Response · Authors · 2023-08-19
> > > > >
> > > > > The discussion time is rapidly approaching. It's clear from the review comments that you've dedicated a significant amount of time and effort to carefully evaluate this manuscript, and I sincerely appreciate your dedication. Regardless of the outcome, I kindly hope that you could provide me with feedback to assist me in gauging whether the rebuttal sufficiently addresses your concerns or if there are areas where additional improvements are required. Thank you once again for your invaluable assistance.

---

> > > > > > ### Comment · Reviewer_1VQ8 · 2023-08-20
> > > > > > **Appreciate for the detailed response**
> > > > > >
> > > > > > I am very grateful for the author's detailed response and experimental investigation. The author's response has addressed my question regarding computational time. However, I still believe that for clarity and respect for the original paper, the computational complexity of the SALTS algorithm in the main body of the paper should be denoted as "k" rather than "n^3". The authors can highlight its selection in a comment or elaborate upon it in the experimental section for greater clarity.
> > > > > >
> > > > > > In summary, the authors do make some analysis for the proposed ATNN approach. However, since the ATNN is developed based on SALTS (replacing the tensor Schatten-p norm to TNN), the main contributions lie in the proof of two main theorems.  But the proof has not made improvements compared with previous work. Consequently, the contributions and significance appear to be limited.

---

> > > > > > > ### Author Response · Authors · 2023-08-20
> > > > > > >
> > > > > > > Regarding the reviewers' opinions, I would like to offer a different perspective.
> > > > > > >
> > > > > > > In comparison to SALTS, we have introduced several enhancements that led to **improved speed** and **theoretical guarantee**.
> > > > > > >
> > > > > > > Additionally, from a theoretical standpoint, we have **relaxed the prerequisites necessary** for other theoretical methods, resulting in significantly improved outcomes compared to the previously theoretically assured TNN series approaches.
> > > > > > >
> > > > > > > In fact, our work did not stem from SALTS. On the contrary, SALTS builds upon Qrank by substituting TNN with the Schatten-p norm to achieve enhanced outcomes, albeit at the cost of significantly increased processing time and the loss of theoretical guarantee. I believe the reviewers are well aware of the distinctions between Qrank, SLATS, and our work.
> > > > > > >
> > > > > > > **Therefore, from the perspectives of theory, speed, and effectiveness, our work differentiates itself from prior research and presents improvements**. I hope the reviewers can recognize the contributions of this paper and understand that they are not inconsequential to the field of the tensor nuclear norm. For our paper, at the very least, a strong rejection should not be the primary outcome.

---

> > > > > > > > ### Comment · Reviewer_1VQ8 · 2023-08-20
> > > > > > > > **I still cannot recognize any novelty**
> > > > > > > >
> > > > > > > > I still cannot recognize the novelty and differences between Eq.(5) in this paper and Eq.(15) in SALTS. Since they fully share the same idea. If the authors consist that the proposed ATNN maintains some differences, please give more clarifications. Additionally, the authors have made mistakes: **SALTS is not a simple extension of Q-rank**, since L is not necessarily a square matrix like Q-rank (same as this paper).

---

> > > > > > > > > ### Author Response · Authors · 2023-08-20
> > > > > > > > >
> > > > > > > > > Thank you for the reviewer's feedback. Indeed, SALTS, Qrank, and our proposed ATNN all share a common concept, which involves learning a transformation matrix from the data. While the original SALTS paper suggests using a column orthogonal matrix for the transformation (similar to ATNN in our paper), our experiments have shown that SALTS' transformation matrix, when not constructed as a full-rank matrix, does not guarantee desired results. We have provided these experimental results in a previous rebuttal and are reiterating them here.
> > > > > > > > >
> > > > > > > > > Method | TNN | ATNN | SALTS^*|SALTS
> > > > > > > > > ---- | ---| ---| ---| ---
> > > > > > > > > PSNR | 26.97 | 30.26 | 25.58 | 32.68
> > > > > > > > > Time |  404.3 | 115.3 | 868.3 |3360
> > > > > > > > >
> > > > > > > > > In the table provided, "SALTS^*" represents the results obtained by selecting the same value of k (i.e., the k used in ATNN) for SALTS. The code for SALTS has been made available, making it straightforward to validate this conclusion through experimentation.
> > > > > > > > >
> > > > > > > > > Setting aside the issue of the parameter "k" in SALTS, there are additional differences between ATNN and SALTS:
> > > > > > > > >
> > > > > > > > > **1**: The discrepancy arises from the dissimilar regularization methods employed to measure the low-rank property of transformed tensors: SALTS employs the Scatten-p norm, whereas ATNN utilizes the tensor nuclear norm. Consequently, it can be concluded that SALTS does not fall within the category of TNN-based approaches.
> > > > > > > > >
> > > > > > > > > **2**: ATNN has theoretical analysis for tensor completion tasks.
> > > > > > > > >
> > > > > > > > > **3**: ATNN continues the framework of TNN and can be applied to problems such as Robust Principal Component Analysis (RPCA), whereas SALTS cannot.
> > > > > > > > >
> > > > > > > > > Finally, I would like to emphasize my utmost respect for the work accomplished in SALTS, which undeniably demonstrates exceptional performance. Nevertheless, I kindly hope that the reviewer can also acknowledge the distinctions between ATNN and SALTS.

---

> > > > > > > > > > ### Author Response · Authors · 2023-08-21
> > > > > > > > > >
> > > > > > > > > > Dear reviewer,
> > > > > > > > > >
> > > > > > > > > > The discussion deadline is approaching, and I'm concerned that the reviewers might forget. I kindly request once again for your final feedback and comments. I apologize for any inconvenience caused.

---

### Official Review · Reviewer_Wzur · 2023-07-07

**Soundness:** 2 fair
**Presentation:** 3 good
**Contribution:** 2 fair
**Rating:** 5
**Confidence:** 4

**Summary:**

This paper introduces a new method of learning a transformation that can make the tensor nuclear norm better capture the low-rank structure of tensor data. The method is fast, data-adaptive, and has a theoretical guarantee of being reversible. The paper shows that the proposed method works well in experiments.

**Strengths:**

1. This paper presents a clear and reasonable idea that is well written and easy to understand.

2. The paper proposes an adaptive transformation matrix, which has a better expressiveness than methods with fixed transformations.

3. The paper proves that ATNN has exact recovery capability under certain conditions.

4. The paper conducts extensive experiments to validate the theoretical results and demonstrate the effectiveness of ATNN.


**Weaknesses:**

1. The performance of ATNN is limited by the linear transformation. As shown in Table 3, S2NTNN achieves higher accuracy and faster speed than ATNN. Since the theoretical analysis of deep neural networks is challenging and remains an open problem, the theory of S2NTNN is harder to establish.

2. ATNN has an advantage over TCTV in terms of computational efficiency for color video completion task, but it cannot surpass TCTV in terms of overall model performance.

3. Typos: Eq.(2) $A^{(k)}\rightarrow A^{i}$


**Questions:**

How does ATNN handle the case of a third-order tensor with low-rank properties along all its three modes?

**Limitations:**

See Weaknesses.

---

> ### Author Rebuttal · Authors · 2023-08-05
>
> We greatly appreciate your valuable insights and feedback on our research. Your feedback is highly meaningful to us, and we truly value the effort you've invested in reviewing our work. We've thoroughly addressed your concerns to provide a comprehensive understanding of our study. As for your concerns, we give a detailed response which we hope can help you fully understand our work. If any further uncertainties remain, please feel free to let us know. We will deeply appreciate that you can raise your score if you find our responses resolve your concerns.
>
> **Q1: The performance of ATNN is limited by the linear transformation. As shown in Table 3, S2NTNN achieves higher accuracy and faster speed than ATNN. Since the theoretical analysis of deep neural networks is challenging and remains an open problem, the theory of S2NTNN is harder to establish.**
>
> **A1**: Constrained by linear transformations, our ATNN might not consistently outperform recently proposed methods like S2NTNN (Yisi Luo, TIP 2022) in specific scenarios; however, it maintains comparability on the whole. We have tabulated the experimental values from Tables 3 and 5, revealing that ATNN and S2NTNN achieve PSNR scores of 30.26 and 30.55, respectively, with corresponding runtimes of 115.3 seconds (GPU computation time) and 108.8 seconds (CPU computation time). Besides, ATNN's advantage lies in its ability to harness the prior theory of TNN, enabling straightforward extensions to tensor imputation and tensor RPCA problems. In contrast, S2NTNN faces challenges when extending to tensor RPCA, which involves precise separation of low-rank and sparse tensors.
>
> Although ATNN cannot ensure superiority over S2NTNN across the board, it significantly outperforms TNN-based methods with fixed transformation matrices. Furthermore, ATNN eliminates the need for parameter tuning, unlike S2NTNN which needs carefully tunning the trade-off parameter between modified nuclear norm and data fidelity terms for each dataset. In summary, while ATNN may not completely surpass S2NTNN in terms of performance, our work introduces a more user-friendly tensor nuclear norm model with theoretical support and without parameter tuning.
>
> **Q2: ATNN has an advantage over TCTV in terms of computational efficiency for color video completion tasks, but it cannot surpass TCTV in terms of overall model performance.**
>
> **A2**: Yes, in certain scenarios, ATNN's performance might not surpass that of TCTV, but overall, it is comparable to TCTV and significantly outperforms other TNN methods. This is because TCTV not only leverages the low-rank nature of tensor data but also integrates local smoothness, making it more effective for processing low-rank image-like data with certain spatial smoothness properties. In fact, by extending ATNN to incorporate the TCTV norm—replacing the tensor nuclear norm in Equation 13 with the TCTV norm to develop the **TCTV*** model—we can elevate our performance even further, surpassing the accomplishments of the TCTV method.
>
> The relationships between the above four methods are as follows: By allowing the transformation matrix to be adaptively learned from the data, we transition from TNN to ATNN; incorporating spatial local smoothness into the TNN model results in TCTV;  integrating TCTV and ATNN results in TCTV*.  The following table presents the average restoration results of the four categories of methods across the eight datasets in Tables 3 and 5. The detailed experimental results can be found in the uploaded rebuttal PDF.
>
> |Indexs|TNN|ATNN|TCTV|TCTV*|
> |---------|----------|----------|----------|----------|
> |PSNR|26.97|30.26|30.64|33.39|
> |Time(s)|404.3|115.3|1200|146.4|
>
> Comparing TNN and ATNN, as well as TCTV and TCTV*, it is evident that the introduction of an adaptive transformation matrix yields a significantly substantial improvement. Furthermore, we observed that the TCTV*, which integrates TCTV and ATNN, outperforms TCTV by a considerable margin. This highlights the significant potential of the methodology of learning the change matrix from the data.
>
> **Q3: Typos: Eq.(2) $A^{(k)} \rightarrow A^{(i)}$**
>
> **A3**: Thank you for your thorough review of the entire manuscript. We will carefully examine the paper based on your suggestions.
>
> **Q4: How does ATNN handle the case of a third-order tensor with low-rank properties along all its three modes?**
>
> **A4**: Thank you for your suggestion; it's a great question. There are two potential approaches in this scenario. The first approach involves separately conducting tensor low-rank decomposition for each of the three dimensions, leading to the learning of relatively independent transformation matrices for each dimension. The second approach consists of performing a Tucker decomposition on the three-order tensor, which results in the learning of three transformation matrices for the entire tensor. In either case, due to the linear transformations involved, it's possible to derive their respective recoverability theories. However, it's important to note that both approaches would require more intricate derivation processes. This is one of our goals for future research.

---

> > ### Author Response · Authors · 2023-08-16
> >
> > Thanks again for your constructive comments for this article. We have addressed your previous questions point by point. We are unsure whether your doubts have been cleared, so we kindly ask you to take some time again to review our responses. Your feedback is highly important to us. We will deeply appreciate that you can raise your score if you find our responses resolve your concerns.

---

> > ### Comment · Reviewer_Wzur · 2023-08-16
> >
> > Thank you for your response. I apologize for the delay in replying, as I have been very busy. I have read the reviews of other reviewers and the Rebuttal. I appreciate the novelty and significance of this paper. However, I am not very familiar with the technical details of this field, so I would like to maintain my score.

---

> > > ### Author Response · Authors · 2023-08-18
> > >
> > > Thank you for your response. I appreciate your careful review of my rebuttal and your recognition of the contributions made in this paper.

---

### Official Review · Reviewer_E1Cr · 2023-07-25

**Soundness:** 3 good
**Presentation:** 2 fair
**Contribution:** 2 fair
**Rating:** 4
**Confidence:** 3

**Summary:**

The authors propose a novel Adaptive TNN (ATNN) for tubal rank (t-SVD)-based low-rank tensor recovery and apply it to Tensor Robust PCA and Tensor Complete problems. Theoretical recovery guarantees have been established which is technically sound. However, tubal rank is known as an analog to matrix rank. Compared to CP rank and Tucker rank, the theory of tubal rank is relatively easy to work with. That said, the reported theoretical results are solid but no surprise. In my option, the proposed ATNN is an analog to matrix sketch, although fast, it is not equivalent to the standard t-SVD.I also think 'learnable' is a misleading term in this paper.

**Strengths:**

Solid analysis reslut.

**Weaknesses:**

1. In my option, the proposed ATNN is an analog to matrix sketch, although fast, it is not equivalent to the standard t-SVD.Just like matrix sketch is not equivalent to the stardand SVD. I don't see this discussed in the paper.

2. When a matrix algorithm is learnable, something like
      Deep convolutional robust PCA with application to ultrasound imaging, ICASSP 2019.
      Learned Robust PCA: A Scalable Deep Unfolding Approach for High-Dimensional Outlier Detection, NIPS 2021.
come to my mind. To me, 'learnable' is a bit misleading in the paper. Anyhow, the learnable feature of the ATNN is neither discussed enough nor numerically demonstrated through the paper.

3. The reported numerical results in Tables 3-5, the proposed approach doesn't show sufficient emperical advantage.

4. The notation is really messed up in some part of the paper. For example, in (12), \mathcal{E] should be \mathcal{S} (or the other way around). In (13), \mathcal{S} should be \mathcal{E}.

**Questions:**

See weakness section.

**Limitations:**

No negative societal impact was found.

---

> ### Author Rebuttal · Authors · 2023-08-05
>
> We would like to express our sincere gratitude for your valuable comments on our work. Your feedback is incredibly important to us, and we truly appreciate the time and effort you've taken to review our work. As for your concerns, we give a detailed response which we hope can help you fully understand our work. Please feel free to let us know if you have any further concerns. We will deeply appreciate that you can raise your score if you find our responses resolve your concerns.
>
> **Q1: In my option, the proposed ATNN is an analog to matrix sketch, although fast, it is not equivalent to the standard t-SVD.Just like matrix sketch is not equivalent to the standard SVD. I don't see this discussed in the paper.**
>
> **A1**:  In essence, the core concept of matrix sketching is compressing extensive matrices into smaller approximations to enhance processing efficiency. All low-rank decomposition techniques embrace this notion, breaking down the task of recovering large data into two smaller matrix recovery problems. Yet, effectively restoring these smaller matrices within the matrix sketch (Edo Liberty, ACM SIGKDD2013) or low-rank decomposition framework while establishing corresponding recovery theorems, still remains a challenging and unresolved matter.
>
> Matrix sketching and low-rank decomposition methodologies offer an idea to shift from recovering large data to smaller data. Our paper presents a pragmatic implementation to efficiently recover two smaller matrices within the low-rank decomposition framework, complemented by recoverability theory. As pointed out by the reviewer, a matrix sketch doesn't equate to standard SVD. The fundamental query centers on whether a matrix can be decomposed into two smaller matrices, which is feasible when the matrix is low rank. For instance, when a matrix of size $m\times n$ with rank r undergoes SVD ($A = USV^T$), where $U$ is the orthogonal matrix with dimensions $m\times r$, $V$ is the orthogonal matrix with dimensions $n\times r$, and $S$ is a diagonal matrix, defining $B = US$ enables us to exploit $AA^T = BV^TVB^T = BB^T$. This confirms that precise low-rank decomposition yields the smaller matrix $B$, which can be utilized in matrix sketching through the SVD algorithm. In our revised version, we will add a discussion on the core principles of matrix sketching to provide a more comprehensive context.
>
> **Q2: When a matrix algorithm is learnable, something like Deep convolutional robust PCA with application to ultrasound imaging, ICASSP 2019. Learned Robust PCA: A Scalable Deep Unfolding Approach for High-Dimensional Outlier Detection, NIPS 2021. come to my mind. To me, 'learnable' is a bit misleading in the paper. Anyhow, the learnable feature of the ATNN is neither discussed enough nor numerically demonstrated through the paper.**
>
> **A2**: I apologize for any confusion arising from the terminology used in the paper. The term "Learnable Transformed Tensor Nuclear Norm" doesn't refer to the entire model being learned through deep networks. Instead, it focuses on adaptive learning of transformation matrices within the model, in contrast to previous Tensor Nuclear Norm (TNN) models with fixed matrices. The term "Learnable Transformed Tensor Nuclear Norm" in our paper is also influenced by existing terminology (Tongle Wu, et al., TNNLS 2022).
>
> Given the prevalent use of deep learning, which emphasizes feature extraction, the concept of learning is often linked to this process. However, in practice, any procedure that involves extracting common patterns from data can be regarded as a form of learning, including dictionary learning. Consider a third-order tensor $\mathcal{X}\in \mathbb{R}^{n_1\times n_2\times n_3}$, decomposed into $n_1n_2$ vectors of length $n_3$. The learned transformation matrix $\mathbf{L} \in \mathbb{R}^{r_3 \times n_3}$ acts as a common basis for these vectors. Each original tensor vector can be linearly represented using  $\mathbf{L}$. Thus, the transformation matrix in ATNN effectively captures the principal component information of the original tensor.
>
>
> **Q3: The reported numerical results in Tables 3-5, the proposed approach doesn't show a sufficient empirical advantage.**
>
> **A3**: Tables 3-5 showcase our approach's experimental outcomes across three tasks: hyperspectral data completion, video foreground-background separation, and video completion. Our approach achieves comparable or superior performance compared to competitors while demanding less execution time. Take Table 4, for instance, where ATNN achieves an AUC of 0.9233, outperforming the second-best CTV's AUC of 0.9180. Remarkably, ATNN's runtime is only 2.3 seconds, significantly faster than the compared methods, highlighting its advantage in both performance and speed.
>
> Indeed, the results in Table 3 and Table 5 demonstrate that in certain scenarios, ATNN slightly lags behind TCTV, which further incorporates a local smoothness prior based on TNN, and S2NTNN, which employs deep learning to learn the transformation matrix. However, the performance of our ATNN surpasses other TNN-based methods by a significant margin, highlighting the importance of learning the transformation matrix from the data. Additionally, it's noteworthy that S2NTNN lacks strong theoretical foundations and necessitates dataset-specific hyperparameter tuning, potentially impeding its convenience. In contrast, ATNN is parameter-free and user-friendly.
>
> Furthermore, it's worth noting that S2NTNN was tested on a GPU, while the other methods were tested on a CPU.
>
> **Q4: The notation is really messed up in some part of the paper. For example, in (12), $\mathcal{E}$ should be $\mathcal{S}$ (or the other way around). In (13), $\mathcal{S}$ should be $\mathcal{E}$.**
>
> **A4**: Thank you for your thorough review of the entire manuscript. We will carefully examine the paper based on your suggestions.

---

> > ### Author Response · Authors · 2023-08-16
> >
> > We understand that the reviewing process requires a significant investment of time and effort. We sincerely value your previous insights into the article, which have played a vital role in enhancing its overall quality. We've carefully attended to each of the points you raised; however, we're uncertain whether your concerns have been fully addressed. Your clarification would mean a lot to us. If any uncertainties persist, we kindly hope you can give us an opportunity for further discussion on its specific aspects. We would greatly appreciate your consideration in adjusting your rating if you find that our responses have effectively addressed your concerns. Your support would be highly valued.

---

> > > ### Author Response · Authors · 2023-08-18
> > >
> > > The deadline for the discussion is approaching. Please consider giving the paper another opportunity. If the rebuttal does not address your concerns, we are prepared to provide a more comprehensive response if necessary. If you find that our responses have effectively addressed your concerns, can you give us feedback and raise your score? Your feedback is crucial to us. Thanks a lot.

---

> > ### Comment · Reviewer_E1Cr · 2023-08-19
> >
> > Thank you for the rebuttal. While the rebuttal answers Q2 (and Q4) well, I still hold a negative opinion of the current version of this work.
> > I am familiar with the difference between matrix sketch and SVD. The question is, given the significant numerical advantage of the existing matrix sketch method, the proposed method has very minimum advantages (and disadvantages in some cases) compared to the SVD based approaches. I think the potential of the proposed sketch based method is not explored enough.
> > Therefore, I will keep my current score. That said, I believe the proposed approach still has a lot of potential for future revision.

---

> > > ### Author Response · Authors · 2023-08-19
> > >
> > > Thank you for the reviewer's response. The following replies are based on the literature we have obtained from these two references, [1] and [2]. If the reviewer can provide other references about the matrix sketch, we would appreciate it.  Once again, I would like to express my gratitude to you.
> > >
> > > [1] Tropp J A, Yurtsever A, Udell M, et al. Practical sketching algorithms for low-rank matrix approximation[J]. SIAM Journal on Matrix Analysis and Applications, 2017, 38(4): 1454-1485.
> > >
> > > [2] Woodruff D P. Sketching as a tool for numerical linear algebra[J]. Foundations and Trends® in Theoretical Computer Science, 2014, 10(1–2): 1-157.
> > >
> > > Matrix sketching is a versatile computational technique for accelerating matrix operations. It involves identifying a linear projection operator (i.e., sketch transformation) to project large-scale matrices onto smaller ones to achieve acceleration. **The success of such techniques hinges on finding effective projection operators.** From this perspective, the proposed ATNN can be viewed as a specific refinement algorithm within this category of methods. However, in contrast to general matrix sketching, our approach learns the transformation matrix adaptively from the data, rather than having it predefined. The referenced literature [1] provides several options for choosing sketch transformations, such as FFT transformation, random Gaussian matrices, and orthogonal matrices. To validate the effectiveness of learning sketch transformation matrices, we have compared our approach with various methods in the table. The table includes a total of six categories of comparative methods, primarily differing in their sketch matrix settings. In the following table, "COM" is an abbreviation for Column Orthogonal Matrix, and COM for sketch_1 is obtained by the SVD of clean data.
> > >
> > > Methods | TNN | sketch_1 | sketch_2 | sketch_3(UTNN) | sketch_4(ATNN) | TCTV|sketch_5(TCTV^*)
> > > ---- | ---| ---| ---| ---| ---| ---| ---
> > > sketch form |  FFT |  **COM in clean data** | random COM | random OM| **learned** COM | FFT with local smooth |**learned** COM with local snooth
> > > PSNR | 26.97|30.15|16.54| 20.36|30.26|30.64|33.39
> > > TIME| 404.3|116.6|118.4|197.8|115.3| 1200|146.4
> > >
> > > From the table, it can be observed that only "sketch_1" yields results close to ATNN. Meanwhile, methods involving randomly assigned sketch matrices, such as "sketch_2" to "sketch_4," significantly underperform compared to TNN. However, for practical restoration tasks, our objective is to restore clean data, making it challenging to use clean data for extracting "COM" as the sketch transformation matrix. Therefore, if we consider ATNN as a specific implementation within the broader category of matrix sketching, **it offers an effective approach for finding sketch transformation matrices.**  In fact, in the simulation experiments, we verified that the transformation matrices learned through the ATNN framework and those obtained from clean data yield comparable results.
> > >
> > > Regarding the reviewer's comment that our method doesn't fully exploit the potential of matrix sketching, in the table above, it's evident that our improvements compared to TNN are quite significant. In the race within the realm of TNN-based methods, we are the front-runners in terms of both time and performance. When comparing against matrix sketching methods, it's apparent from the table that our learned matrix sketching approach outperforms the predefined sketch matrix methods. The primary objective of our paper is to demonstrate the effectiveness of learning transformation matrices within TNN-type methods (or the sketch matrices in the matrix sketching framework) from data. In the table, we extend our learned method to compete with the top-performing TCTV approach mentioned in the paper, and the enhancements achieved with TCTV^* are equally remarkable.
> > >
> > > Furthermore, there are notable distinctions between the framework of tensor nuclear norm (TNN) and matrix sketching. TNN investigates the low-rank nature of transformed tensor slices, allowing richer information preservation compared to matrices. In contrast, matrix sketching involves reshaping tensor data into matrices, which may result in information loss. Regardless of the framework, whether TNN or matrix sketching, obtaining optimal transformation matrices is pivotal for achieving effective outcomes in both paradigms.
> > >
> > > Lastly, from a theoretical standpoint, matrix sketching offers an error bound that elucidates the loss incurred in transforming matrices to smaller scales. However, our approach based on the ATNN norm provides an exact recoverability theory. Hence, our theoretical results concerning the ATNN method are superior.
> > >
> > > We hope the reviewer could spare some time to review our response. We are unsure if this reply addresses your concerns adequately. We will take any suggestions you may have seriously. Thanks a lot.

---

> > > > ### Author Response · Authors · 2023-08-19
> > > >
> > > > The discussion time is rapidly approaching. It's clear from the review comments that you've dedicated a significant amount of time and effort to carefully evaluate this manuscript, and I sincerely appreciate your dedication. Regardless of the outcome, I kindly hope that you could provide me with feedback to assist me in gauging whether the rebuttal sufficiently addresses your concerns or if there are areas where additional improvements are required. Thank you once again for your invaluable assistance.

---

> > > > ### Author Response · Authors · 2023-08-20
> > > >
> > > > Dear review,
> > > >
> > > > Since the end of the discussion period is approaching, if you have any further concerns please feel free to let us know and we are pleased to discuss them with you. Thank you very much!

---

> > > > > ### Author Response · Authors · 2023-08-21
> > > > >
> > > > > Dear reviewer,
> > > > >
> > > > > Today marks the final day of discussion. I'm unsure whether my response has addressed your concerns. If all of my responses over these past few days have addressed your concerns, would you be willing to consider raising the score?

---

> > > > > > ### Author Response · Authors · 2023-08-21
> > > > > >
> > > > > > Dear reviewer,
> > > > > >
> > > > > > The discussion deadline is approaching, and I'm concerned that the reviewers might forget. I kindly request once again for your final feedback and comments. I apologize for any inconvenience caused.

---

### Author Rebuttal · Authors · 2023-08-06

Dear AC and reviewers,

Thank you so much for your valuable comments and we truly appreciate the time and effort you've taken to review our work. We are also pleased that the reviewers recognized the value of our work and provided positive feedback. Your feedback is very important to us, and according to the reviewer's comments, we have revised the manuscript carefully and made detailed responses to all your concerns (**please refer to our rebuttal for each reviewer for more detailed responses**).

The focus of this study is to address the importance of adaptively learning an effective transformation matrix from data, a critical aspect within the tensor nuclear norm framework. However, due to the choice of comparison methods in this study, some methods are not purely low-rank, such as the TCTV method that combines local smoothness and low-rank priors. As a result, our proposed approach may not appear to be universally superior among all methods. The selection of the TCTV method serves the purpose of illustrating that even without utilizing local smoothness, our ATNN can achieve comparable performance to both TCTV and the deep learning-based S2NTNN. Additionally, **we have the option of combining the principles of ATNN with TCTV to create a more potent model called TCTV*** . Due to the slow runtime of SALTS and its exclusion from TNN-class methods, coupled with the lack of access to Qrank's code during submission, a comparison of these methods was not included in the paper. To address this, we have provided a **one-page PDF** file showcasing the numerical experimental results of TCTV*, Qrank, and SALTS on the completion problem. The average results of these methods across 8 data samples at a 5% sampling rate are as shown below:

|Indexs|TNN|TNN*(Our ATNN)|S2NTNN|Qrank|SALTS|TCTV|TCTV*(Ours)|
|---------|----------|----------|----------|----------|----------|----------|----------|
|PSNR|26.97|30.26|30.55|29.54|32.68|30.64|33.39|
|Time(s)|404.3|115.3|108.8|452.3|3360|1200|146.4|

When comparing TNN and TNN*, as well as TCTV and TCTV*, it becomes evident that the incorporation of an adaptive transformation matrix results in a notably substantial improvement. Comparing TNN* and Qrank, it can be observed that the performance of the proposed approach, which sets the transformation matrix as column orthogonal matrices, outperforms using invertible matrices. Furthermore, TCTV* achieves optimal performance, further affirming the effectiveness of the method proposed in this paper.

Regarding the theoretical aspect, we have relaxed the requirement from previous works, which needed the transformation matrix to be invertible, to only needing the transformation matrix to be a column full-rank matrix in certain cases.

As for the time complexity mentioned by the reviewers, we have included the time complexity in the rebuttal PDF.  For a given tensor $\mathcal{X}\in \mathbb{R}^{n_1\times n_2\times n_3}$, the sizes of the transformed tensors of all other methods are not smaller than $n_1\times n_2\times n_3$. Only our method results in a small tensor size of $n_1\times n_2\times r_3(r_3\ll n_3)$. Therefore, our method possesses the lowest time complexity. The time complexities of some methods are as shown below:
|Methods|TNN|DCTNN|UTNN|FTNN|S2NTNN|Qrank|SALTS|ATNN(Ours)|
|---------|----------|----------|----------|----------|----------|----------|----------|----------|
|TIme Complexity|$\mathcal{O}(n_1n_2^2n_3)$|$\mathcal{O}(n_1n_2^2n_3)$|$\mathcal{O}(n_1n_2^2n_3)$|$\mathcal{O}(k^3n_1n_2^2n_3)$|$\mathcal{O}(n_1n_2^2n_3)$|$\mathcal{O}(n_1n_2^2n_3)$|$\mathcal{O}(n_1n_2^2n_3)$|$\mathcal{O}(n_1n_2^2r_3)$|

Regarding the crowding of definitions and some typos in the paper, we will further revise them in future versions to enhance the readability of the paper.

Once again, thank you for taking the time to review our work and providing valuable insights, and we will take all the suggestions given by the reviewers into consideration for future improvements. We would also like to have discussions with all the reviewers if you have any further concerns.

---

> ### Author Response · Authors · 2023-08-16
>
> Dear AC and Reviewers,
>
> Thanks again for the constructive comments of all the Reviewers. As the deadline for discussion is approaching, we sincerely look forward to your further feedback. Please feel free to let us know if you have any further concerns or comments.
>
> Thanks. The authors.

---

### Decision · Program_Chairs · 2023-09-21

**Decision:**

Reject

**Comment:**

This paper proposes an Adaptive TNN (ATNN) for tubal rank (t-SVD)-based low-rank tensor recovery and apply it to Tensor Robust PCA and Tensor Complete problems, which is technically sound. The proposed transform is shown to be effective and efficient with theoretical guarantees for the exact recovery.  The authors have successfully addressed some questions and concerns such as computational time by the rebuttal and discussions.  However, several reviewers still have concerns that the novelty of this work is limited, and the theorems and their proofs can be naturally extended from the existing theory, which makes the contribution of this work is incremental.